# Foundational Challenges in Assuring Alignment and Safety of Large Language Models

Usman Anwar[1]

Abulhair Saparov[*2], Javier Rando[*3], Daniel Paleka[*3], Miles Turpin[*2], Peter Hase[*4], Ekdeep Singh Lubana[*5], Erik Jenner[*6], Stephen Casper[*7], Oliver Sourbut[*8], Benjamin L. Edelman[*9], Zhaowei Zhang[*10], Mario Günther[*11], Anton Korinek[*12], Jose Hernandez-Orallo[*13]

Lewis Hammond[8], Eric Bigelow[9], Alexander Pan[6], Lauro Langosco[1], Tomasz Korbak[14], Heidi Zhang[15], Ruiqi Zhong[6], Seán Ó hÉigeartaigh[‡1], Gabriel Recchia[16], Giulio Corsi[‡1], Alan Chan[‡17], Markus Anderljung[‡17], Lilian Edwards[‡18], Aleksandar Petrov[8], Christian Schroeder de Witt[8], Sumeet Ramesh Motwani[6]

Yoshua Bengio[‡19], Danqi Chen[‡20], Philip H.S. Torr[‡8], Samuel Albanie[‡1], Tegan Maharaj[‡21], Jakob Foerster[‡8], Florian Tramer[‡3], He He[‡2], Atoosa Kasirzadeh[‡22], Yejin Choi[‡23]

David Krueger[‡1]

[*]indicates major contribution.
[‡]indicates advisory role.

[1] *University of Cambridge* [2] *New York University* [3] *ETH Zurich* [4] *UNC Chapel Hill*
[5] *University of Michigan* [6] *University of California, Berkeley* [7] *Massachusetts Institute of Technology*
[8] *University of Oxford* [9] *Harvard University* [10] *Peking University* [11] *LMU Munich*
[12] *University of Virginia* [13] *Universitat Politècnica de València* [14] *University of Sussex*
[15] *Stanford University* [16] *Modulo Research* [17] *Center for the Governance of AI*
[18] *Newcastle University* [19] *Mila - Quebec AI Institute, Université de Montréal* [20] *Princeton University*
[21] *University of Toronto* [22] *University of Edinburgh* [23] *University of Washington, Allen Institute for AI*

## Abstract

This work identifies 18 *foundational* challenges in assuring the alignment and safety of large language models (LLMs). These challenges are organized into three different categories: *scientific understanding of LLMs*, *development and deployment methods*, and *sociotechnical challenges*. Based on the identified challenges, we pose 200+ concrete research questions.

Corresponding author: Usman Anwar «usmananwar391@gmail.com»

# Contents

# Reader's Guide

Due to the length of this document (though note that the main content is only ~100 pages; the rest are references), it may not be feasible for all readers to go through this document entirely. Hence, we suggest some reading strategies and advice here to help readers make better use of this document.

We recommend all readers begin this document by reading the main introduction (Section 1) to grasp the high-level context of this document. To get a quick overview, readers could browse the introductions to various categories of the challenges (i.e. Sections 2, 3 and 4) and review associated Tables 1, 3 and 4 that provide a highly abridged overview of the challenges discussed in the three categories. From there on, readers interested in a deep dive could pick any section of interest. Note that all the challenges (i.e. subsections like Section 2.1) are self-contained and thus can be read in an arbitrary order.

### Machine Learning and NLP Researchers

Technical researchers in machine learning, natural language processing, and other associated fields are the primary intended audience for this agenda. We have tried to assume as minimal background knowledge as possible beyond the general knowledge of what LLMs are, what their architecture is, and how they are trained. Hence, we expect all the technical challenges in Sections 2 and 3 to be accessible to any person with the knowledge equivalent to a first-year graduate student in machine learning or natural language processing. A large proportion of the challenges discussed in Section 4 are also technical in nature and should be equally accessible.

The main intended purpose of this document is to help junior researchers, or researchers new to this area, to identify promising and actionable research directions (although of course, even seasoned experts might take inspiration from it). Such readers are encouraged to pick and choose sections that best align with their interests. The 200+ listed research questions are each meant to be roughly the size of a problem that could form the basis of a research dissertation. For each challenge and subchallenge, we provide motivation, background, and related work, before discussing directions for future research. These should provide a good starting point for researchers who are new to these particular challenges, but we do not attempt a comprehensive survey of any area.

We also note that while this work is motivated by the safety and alignment of LLMs, nonetheless, many of the challenges we identify are highly interesting from the technical and scientific points of view. Thus, even those readers who are not primarily motivated by safety, but are in search of interesting problems centered on LLMs, may find this document useful.

### Sociotechnical Researchers and Other Stakeholders

We focus on sociotechnical challenges in Section 4, emphasizing that all LLMs are sociotechnical systems, and their safety cannot be ensured without a deep and thoughtful consideration through this lens. The introduction of this section provides a mapping between the different challenges we discuss and the different areas of other fields that could contribute to progress on those challenges. This section only presumes high-level familiarity with the LLMs for the most part and is aimed to be accessible to a wider audience than the rest of the agenda.

# 1 Introduction

> *"We can only see a short distance ahead, but we can see plenty there that needs to be done."*
>
> — *Alan Turing*

Large language models (LLMs) have emerged as one of the most powerful ways to solve open-ended problems and mark a paradigm shift within machine learning. However, assuring their safety and alignment remains an outstanding challenge that is recognized across stakeholders, including private AI laboratories (Leike et al., 2022; Anthropic, 2023a; Frontier Model Forum, 2023), national and international governmental organizations (White House, 2023; Office, 2023; Board, 2023), and the research and academic communities (Bengio et al., 2023; FAccT, 2023; CAIS, 2023; CHAI, Far.ai, and Ditchley Foundation, 2023). Indeed, assuring the safety and alignment of any deep-learning-based system is difficult (Ngo et al., 2023). However, this challenge is much more acute for LLMs due to their expansive scale (Sanh et al., 2019, Figure 1) and increasingly broad spectrum of capabilities (Bubeck et al., 2023; Morris et al., 2023). Furthermore, the rapid advances in LLM capabilities not only expand the potential applications of LLMs, but also increase their potential for societal harm (Weidinger et al., 2021; Ganguli et al., 2022; Birhane et al., 2023; Chan et al., 2023a).

## 1.1 Why This Agenda?

The rapid rate of progress is especially alarming due to the absence of the requisite technical tools and deficiencies in the sociotechnical structures that may help assure that LLMs are developed and deployed safely (Bengio et al., 2023). In this work, we map out the challenges in developing the appropriate technical affordances that may help assure safety and in understanding and addressing the sociotechnical challenges that we may face in assuring *societal-scale* safety. At its heart, this work is a call to action for machine learning researchers, and researchers in the associated fields. Our extensive referencing of contemporary literature, focus on identifying promising and concrete research directions, and in-depth discussion of each challenge makes it an ideal educational resource for newcomers to the field. At the same time, we expect the plethora of challenges identified in this work to act as a source of inspiration for current practitioners in the fields of LLM alignment and safety, including those working from diverse other disciplines (e.g. social sciences, humanities, law, policy, risk analysis, philosophy, etc.).

Several prior studies have compiled and discussed foundational problems in AI Safety (Amodei et al., 2016; Hendrycks et al., 2021a; Critch and Krueger, 2020; Kenton et al., 2021; Ngo et al., 2023). However, LLMs mark a paradigm shift and present many novel and unique challenges in terms of alignment, safety, and assurance that are not discussed in these works. Among these, Kenton et al. (2021) is the only work that exclusively focuses on LLMs. But it lacks broad coverage, being narrowly focused on issues from accidental misspecification of objectives. Our work builds on the aforementioned work and provides the most comprehensive and detailed treatment of challenges related to the alignment and safety of LLMs to date.

We highlight 18 different *foundational* challenges in the safety and alignment of LLMs and provide an extensive discussion of each. Our identified challenges are foundational in the sense that without overcoming them, assuring safety and alignment of LLMs and their derivative systems would be highly difficult. For this work, we have further prioritized the discussion of foundational challenges that are unambiguous (i.e. not speculative), prime for research and highly relevant to harms and risks posed by current and forthcoming LLMs. Additionally, we pose 200+ concrete research questions for further investigation. Each of these is associated with a particular fundamental challenge. These research questions are fairly open-ended and are roughly meant to be the size of a graduate thesis, although many offer multiple angles of attack and could easily be studied more exhaustively.

## 1.2 Terminology

The terms *alignment*, *safety* and *assurance* have different meanings depending on the context. We use alignment to refer to *intent alignment*, i.e. a system is aligned when it is 'trying' to behave as intended by some human actor (Christiano, 2018).[1] Importantly, alignment does not guarantee a system actually *behaves* as intended; for instance, it may fail to do so due to limited capabilities (Ngo et al., 2023). To further simplify our discussion, we fix the intent to be that of the LLM developer (Gabriel, 2020; Ngo et al., 2023) (e.g. as opposed to the user). We consider a system safe to the extent it is unlikely to contribute to unplanned, undesirable harms (Leveson, 2016). This is a somewhat expansive definition, accounting not only for the technical properties of the system, but also the way in which it is (or is likely to be) deployed and used (Weidinger et al., 2023a), though it is narrow in the sense that it does not consider intentional harm, and it does not set out any criteria for what constitutes harm. Alignment can be used to increase safety, but the relationship is not straightforward: it could also be used to make a system more dangerous (if the developer intends), and it is not directly concerned with the real-world impact a system has when embedded in its deployment context, or the very important issue of (lack of) alignment between developers and other stakeholders. Finally, by assurance, we mean *any* way of providing evidence that a system is safe or aligned (Ashmore et al., 2021), including, but not limited to: scientific understanding of the system, behavioral evaluations, explanations of the model behavior or internal processing, and adherence to responsible development practices by the system developer (Casper et al., 2024a).

## 1.3 Structure

We organize the foundational challenges into three different categories. The first category, *scientific understanding of LLMs* (Section 2), surveys the most important open questions that can help us build a better 'theory' of how LLMs function and inform development and deployment decisions. We discuss the need to develop principled solutions to conceptualizing, estimating, understanding, and predicting the capabilities of LLMs. We single out in-context learning and reasoning capabilities of LLMs as being critical to understand for assuring alignment and safety across all contexts. We highlight that risks we are facing with current LLMs may grow manifold with the introduction of LLM-agents, and that we need to pre-emptively understand these risks and work to mitigate them across both single-agent and multi-agent scenarios. Finally, we note that it may be inevitable that safety-performance trade-offs exist for LLM-based systems that we ought to understand better.

The second category, *Development and Deployment Methods* (Section 3), presents the known limitations of existing techniques in assuring safety and alignment in LLMs. We identify opportunities to help improve model alignment by modifying the pretaining process to produce more aligned models, survey several limitations of finetuning in assuring alignment and safety, discuss issues underlying the 'evaluation crisis', review challenges in interpreting and explaining model behavior, and finally provide an appraisal of security challenges like jailbreaks, prompt-injections, and data poisoning. On the whole, this section pertains to researching empirical techniques that may help improve the alignment, safety, and security of LLMs.

The final category, *Sociotechnical Challenges* (Section 4), focuses on challenges that require a more diverse and holistic lens to address. For example, we discuss the importance of societal-level discussions of whose values are encoded within LLMs, and how we can prevent value imposition and enable value plurality. Many LLM capabilities are dual-use; there is a need to understand what malicious misuse such capabilities might enable and how may we guard against them. There is also a need to ensure that the biases and other issues of LLM-systems are independently and continually monitored and transparently communicated, to build trustworthiness and reduce over-reliance. The proliferation of LLMs throughout society may have undesirable socioeconomic impacts (e.g. job losses, increased inequality) that ought

---

[1] Christiano (2018) further clarify that they mean this definition to apply *de dicto* not *de re*; we are ambivalent on this point.

to be investigated better and strategized against. Finally, we conclude by discussing the challenges and opportunities in the space of governance and regulation.

As an addendum, in Section 5 we review some limitations of this work. Most notably, we note that while comprehensive, this agenda is not exhaustive. We follow that up with a detailed overview of the prior works broadly related to this work; including but not limited to prior agendas on AI safety and various surveys related to LLMs.

## 2   Scientific Understanding of LLMs

Many safety and alignment challenges stem from our current lack of scientific understanding of LLMs. If we were to understand LLMs better, this would help us better estimate the risks posed by current, and future, LLMs and design appropriate interventions to mitigate those risks. Scientific understanding is also an essential component of assurance — especially for complex systems. Some classic examples of complex systems that heavily rely on scientific understanding for safety and assurance are bridge design, aircraft design, nuclear reactor design, spacecraft design etc. Without the scientific understanding of the underlying physics, the assurance of these systems would perhaps be improbable, if not totally impossible.

LLMs show many prototypical traits of complex systems (Holtzman et al., 2023; Steinhardt, 2023; Hendrycks, 2023, Chapter 5) — the foremost among them is *emergent behaviors* (Wei et al., 2022a; Park et al., 2023a). This complex-systems like nature of LLMs means that relying on 'evaluations' alone may be insufficient for safety and assurance (c.f. Sections 2.2.3 and 3.3) and that there is a dire need to probe beyond surface-level behaviors of LLMs and to understand how those behaviors arise in the first place (Holtzman et al., 2023).

Understanding LLMs is a broad and grand scientific challenge. However, in line with the general theme of this work, we focus on the most safety-relevant aspects (see Table 1 for an overview). Addressing these challenges will help inform safer LLM development or deployment practices; however, additional work would be required to translate insights here into practical recommendations.

Scientific understanding can take many different, and diverse, forms (Adams et al., 2014, Table 4). Indeed, the challenges we have identified admit diverse styles of research. While some challenges are deeply theoretical in nature (e.g. What Are the Computational Limits of Transformers? or Understanding Scaling Laws), others may need a more empirical approach towards resolving them (e.g. Influence of Single-Agent Training on Multi-Agent Interactions is Unclear). In general, most of the challenges that we raise are focused on developing a qualitative understanding of LLMs. Qualitative understanding often allows developing generalizations that a quantitative analysis may not permit and thus may be more robust to the rapid advancements in LLMs. These virtues of qualitative understanding were extolled almost 50 years ago by Herbert Simon and Allen Newell in their seminal 1975 Turing Award lecture, incidentally also on the topic of designing and understanding artificial intelligence systems (Newell and Simon, 1976).

Table 1: An overview of challenges discussed in Section 2 (Scientific Understanding of LLMs). We stress that this overview is a highly condensed summary of the discussion contained within the section, and hence should not be considered a substitute for the complete reading of the corresponding sections.

| Challenge | TL;DR |
|---|---|
| In-Context Learning (ICL) Is Black-Box | We do not have a robust understanding of how and why in-context learning emerges with large-scale training, what mechanisms underlie in-context learning in LLMs, to what extent in-context learning in LLMs is due to mesa-optimization, or how it relates to existing learning algorithms. |

*Continued on the next page*

| Challenge | TL;DR |
| --- | --- |
| Capabilities Are Difficult to Estimate and Understand | Correctly estimating and understanding capabilities of LLMs is difficult for various reasons. Firstly, LLM capabilities appear to have a different 'shape' than human capabilities; meaning that the notions used to understand and estimate human capabilities might be ill-suited to understand LLM capabilities. Additionally, the concept of capabilities lacks a rigorous conceptualization which makes it difficult to make, and evaluate, formal claims about LLM capabilities. There also exist *fundamental* flaws in our evaluation methodologies that ought to be overcome if we are to better understand LLM capabilities, such as benchmarking being unable to differentiate between alignment failures and capability failures. There is also a need to improve our tooling to evaluate the generality of LLMs, and in general, develop methods to better account for scaffolding in our evaluations. |
| Effects of Scale on Capabilities Are Not Well-Characterized | Various challenges hinder our ability to understand and predict the impact of scale on LLM capabilities. These include incomplete theoretical understanding of empirical scaling laws, limited understanding of limits of scaling and how learning representations are affected by scaling, confusing discourse on 'emergent' capabilities due to lack of formalization, and the nascent nature of research into the development of better methods for discovering task-specific scaling laws. |
| Qualitative Understanding of Reasoning Capabilities Is Lacking | Our current understanding of how reasoning capabilities emerge in LLMs and are impacted by model scale is insufficient for making confident predictions about the reasoning capabilities of future LLMs. There is a need for research to understand the mechanisms underlying reasoning, develop a better understanding of the non-deductive reasoning capabilities of LLMs, and better understand the computational limits of the transformer architecture. |
| Agentic LLMs Pose Novel Risks | For reasons including increased capabilities (via enhancements like access to various affordances) and increased autonomy, LLM-agents may pose novel alignment and safety risks. The actions executed by LLM-agents may result in negative side-effects due to underspecification in natural-language-based instructions. Goal-directedness may cause LLM agents to exhibit undesirable behaviors such as reward hacking, deception, and power-seeking, and might make robust oversight and monitoring of LLM-agents particularly difficult. |

*Continued on the next page*

| Challenge | TL;DR |
|---|---|
| Multi-Agent Safety Is Not Assured by Single-Agent Safety | Assuring favorable outcomes in a multi-agent setting may prove challenging for several reasons. Firstly, there's a lack of comprehensive understanding of how single-agent training affects the behavior of LLM-agents in multi-agent environments. Secondly, the foundationality of LLM-agents may contribute to correlated failures. Additionally, collusion among LLM agents may result in undesirable externalities. Lastly, it is unclear to what extent prior research in multi-agent reinforcement learning may prove helpful in improving the alignment of LLM-agents in multi-agent settings, especially for resolving social dilemmas. |
| Safety-Performance Trade-offs Are Poorly Understood | Safety-performance trade-offs are typically unavoidable in the design of any engineering system; however, they are not well understood for LLM-based systems. There is a need for work to design better metrics to measure safety, to characterize safety-performance trade-offs across various contexts, and to better understand what safety-performance trade-offs are fundamental in nature (and hence unavoidable in practice). Finally, it may be helpful to research methods for producing *Pareto* improvements in both safety and performance. |

## 2.1 In-Context Learning (ICL) Is Black-Box

In-context learning (ICL) is the ability of an LLM to learn to perform a novel task, or improve on an existing task, based on the information (e.g. examples, reasoning traces) provided in the prompt without any explicit updates to the model's parameters (Brown et al., 2020; Kaplan et al., 2020). ICL is a highly flexible and efficient learning paradigm — it has been used to direct LLMs to behave in the desired way (Lin et al., 2023a), jailbreak LLMs (Wei et al., 2023a; Xhonneux et al., 2024) and create highly performant LLM-agents (Wang et al., 2023a). **However, while this dynamic nature of ICL is helpful in the design of proficient LLM-based systems, the black-box nature of ICL also makes it significantly harder to assure safety and alignment of LLMs (Wolf et al., 2023; Millière, 2023).** If we do not understand how ICL works, this makes it difficult to predict how it might alter LLMs behavior in deployment; for instance, it might enable novel dangerous capabilities or bypass safeguards (Anil et al., 2024). Thus there is a pressing need to better understand the mechanisms underlying ICL, the limits of ICL, and the safety implications associated with ICL. This section presents the issues with various theories and approaches that have been proposed to explain ICL and highlights several key research questions that need to be addressed.

### 2.1.1 Is ICL Sophisticated Pattern-Matching?

There are two competing theories to explain the working mechanism of ICL in a transformer. The first is that ICL is a set of pre-learned pattern-matching heuristics closely tied to the training distribution. Under this view, several works have proposed explanations of ICL as inference over an implicitly learned topic model (Xie et al., 2022; Wang et al., 2023b; Wies et al., 2023); as task inference (Min et al., 2022; Todd et al., 2023; Hendel et al., 2023; Bigelow et al., 2023) or as learning of template circuits

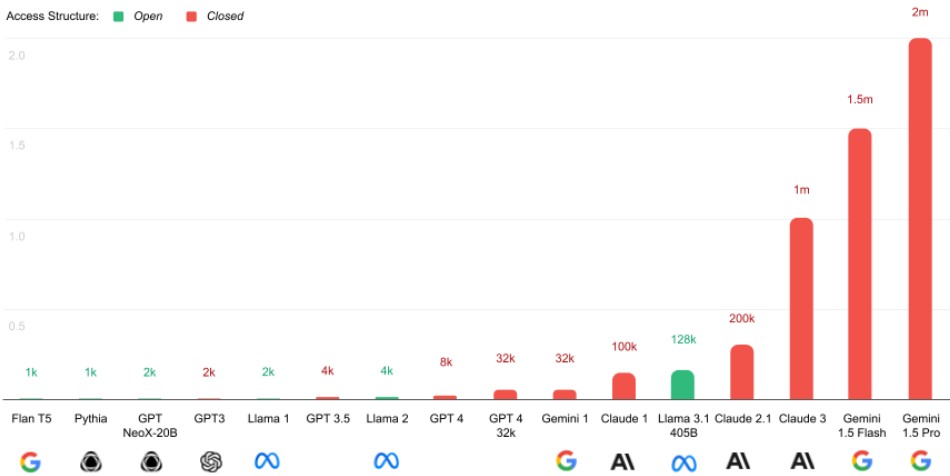

Figure 1: Recent large language models (LLMs) offer significantly expanded context lengths, enhancing in-context learning capabilities (Agarwal et al., 2024) while also introducing novel risks (Anil et al., 2024).

(during training) which are adaptively retrieved and rebound to tokens in the prompt (Swaminathan et al., 2023). However, these theories currently only provide *partial* explanations of ICL. All the aforementioned works only explain ICL that is based on demonstrations while ICL can occur from other types of feedback as well, e.g. interactive feedback (Mehrabi et al., 2023; Wang et al., 2023a). These works also do not explain multi-task in-context learning and sequential learning of tasks in-context (Zhou et al., 2022, Sections 4&5), or how in-context learning may support learning of novel tasks, such as learning to reason on OOD samples (Saparov et al., 2023). Furthermore, some works, such as Zhang et al. (2023a) and Swaminathan et al. (2023), assume specific (simpler) data generation processes that differ from the true data generation process of natural language. There is a need to further refine and extend the current theories, or development of novel ones, so that we are able to adequately explain the full range of ICL behaviors, including the aforementioned ones.

### 2.1.2 Is ICL Due to Mesa-Optimization?

Alternatively, ICL can be seen as a form of "learned optimization" or "mesa-optimization" (Hubinger et al., 2019; von Oswald et al., 2023), i.e., during training, the base optimizer learns to use the transformer weights to represent another (learned) optimization algorithm — as well as a (learned) objective function — effectively allowing the transformer to self-generate learning signal (based on data given in the context) and act as *black-box* in-context learner (Kirsch et al., 2022). Emergence of mesa-optimization within a model is contingent on whether or not a model can implement a learning algorithm within its weights. Several studies provide evidence that transformers can approximate gradient-based learning algorithms for various statistical learning problems (Akyürek et al., 2022a; von Oswald et al., 2022; Garg et al., 2022; Zhang et al., 2023b; Ahn et al., 2023). Bai et al. (2023a) prove that a transformer with $2L$ layers can simulate $L$-steps of gradient descent on a two-layer feedforward neural network. Panigrahi et al. (2023) propose an extension of the transformer architecture that can internally simulate finetuning of a smaller transformer. Bai et al. (2023a) further show that transformers can implement *in-context algorithm selection*, i.e. at inference time, a single transformer can adaptively choose between different learning algorithms to solve the given task (e.g. logistic regression for a regression task or logistic classification for a classification task) based on the information given in the prompt.

These studies show that *in principle*, transformers are capable of mesa-optimization in controlled experiments. However, it is not yet clear whether and when transformers trained with more complex objectives, e.g. the language modeling objective, might learn to perform mesa-optimization. Specifically, one key unknown is: for which mesa-objectives do transformers support mesa-optimization? Existing work provides overwhelming evidence that transformers can solve simple supervised learning tasks via mesa-optimization. However, there has so far been very limited investigation as to whether transformers can solve more complex learning tasks in-context via mesa-optimization (Lin et al., 2023b); research should explore tasks that better mirror the real-world structure of language modeling. Furthermore, even for otherwise well-studied learning tasks e.g. linear regression, there is disagreement in the current literature as to which learning algorithm is implemented by the transformer (von Oswald et al., 2022; Fu et al., 2023a). In order to resolve this disagreement, further research is required to disentangle various factors that may impact which learning algorithm gets implemented by the transformer (Zhong et al., 2023a). Future work could also develop more rigorous and generic methods for distinguishing mesa-optimization from other forms of ICL, and examine the practical importance of the distribution of in-context examples in determining whether mesa-optimization occurs and understanding the inductive biases of the resulting mesa-optimizer.

### 2.1.3  What Behaviours Can Be Specified In-Context?

It is currently unclear what functions can, and can not be learned, in-context. More specifically, given a pre-trained model, what behaviors can it be prompted to do? If it is possible to *always* find a prompt to coax the LLM to perform any task, then that might indicate that jailbreaking (c.f. Section 3.5) is an impossible problem to solve (Millière, 2023). Fundamentally, that is a question of universal approximation *in-context*, i.e. whether a *fixed* model can be converted into an approximator of *arbitrary* accuracy for any (continuous) function by being prompted appropriately. While it is well-known that transformers are universal approximators when *trained* (Yun et al., 2019) and was recently shown that this is also the case for state-space models (Li et al., 2022a; Wang and Xue, 2023; Cirone et al., 2024), it is less clear what are their *in-context* approximation abilities. Wang et al. (2023c) showed that in-context universal approximation is possible with a transformer but their construction required that all possible functions are encoded in the model weights which results in unrealistically large models. However, (Petrov et al., 2024) showed that no memorization is needed and that a relatively small model can be a universal approximator in-context. This result already indicates that it is possible for a realistically-sized model to be impossible to be safeguarded (i.e. safety finetuned in a way that jailbreaking is not possible).

The theory of universal approximation in-context is still rather underdeveloped and the practical safety and security implications of the above results are not yet fully clear. For instance, the above-mentioned models require very specific hand-crafted weights for the pre-trained model. It is not clear whether the universal approximation behavior can be obtained by learning via gradient descent on typical datasets and, therefore, whether it is likely to occur in real-world models or not. Furthermore, the prompt lengths required in the construction of Petrov et al. are unrealistically long. However, it might be possible to reduce this by leveraging knowledge and skills already present in the model (Petrov et al., 2023a). Therefore, studying the effect of the pre-training data on the in-context universal approximation could help understand the real-world safety and security implications of in-context learning capabilities of LLMs. All the above results also rely on the attention mechanism in the transformer architecture and thus, do not translate to other recurrent models. Thus, understanding the in-context approximation properties of other architectures is another open problem with little to no current work so far (Lee et al., 2023a).

### 2.1.4  Scenario-Based Mechanistic Understanding of ICL

Current interpretability techniques are not scalable and general enough to allow an interpretability-based general understanding of the mechanics of ICL within an LLM (c.f. Section 3.4). However, they

can still be leveraged for developing scenario-based *mechanistic* understanding of ICL (Olsson et al., 2022; Reddy, 2023), i.e. identifying circuits critical for ICL within an LLM when artificial restrictions are placed on the prompt structure and the task being carried out via ICL. Such case studies can be insightful for understanding the relative roles played by different computational structures within the model, and to understand how the ICL mechanism varies across tasks. For example, Todd et al. (2023) studied ICL for extractive and abstractive NLP-based tasks, and found that a small number of attention heads, termed "function vectors", are responsible for transporting a compact representation of a task which then triggers execution of the task. Similarly, Merullo et al. (2023a) show that on the commonly-studied ICL task of inferring relations between entities (e.g. inferring the capital of a country), mid-to-late feedforward layers play a key role in identifying and surfacing relevant items contained in the context which are required for inferring the relation and completing the task. Halawi et al. (2023) study ICL on a classification task in which the demonstration data has incorrect labels, thus, clashing with the prior knowledge of the LLM. This lets Halawi et al. discover *false induction heads*; attention heads in late layers of the LLM that attend to and copy false information from the demonstrations.

This preliminary evidence suggests that case studies on specific task types can be a useful technique to build a mechanistic understanding of ICL in LLMs and may be helpful in identifying how, and why, ICL behavior varies across tasks, and how the inductive biases of a particular architecture affects ICL. Future research may analyze ICL in larger LLMs, on diverse task types and with various prompt styles. Interpretability-based techniques could also be leveraged to explain idiosyncrasies of ICL identified in the literature, e.g. why ICL sometimes performs *worse* when given examples from test distributions (Saparov et al., 2023) or why LLMs at different scales process false information in the context differently (Wei et al., 2023b).

### 2.1.5 Understanding the Effect of the Pre-training Data Distribution on ICL

Emergence of in-context learning is heavily modulated by the structure of the pre-training data distribution. Special properties of the task distribution, such as task diversity (Raventós et al., 2023; Kirsch et al., 2022), "burstiness" (Chan et al., 2022) or compositional structure (Hahn and Goyal, 2023) may be key factors in the emergence of ICL. However, these findings are primarily limited to relatively simple settings, and further work is required to verify their correctness in the real-world language modeling setup. Which of these criteria are actually fulfilled by large-scale text datasets? If all of these criteria are indeed met by text datasets, which criteria are actually responsible for ICL in LLMs? For some of these criteria (e.g. "burstiness"), these questions could be answered via analysis of popular datasets like *The Pile* (Gao et al., 2020), *RedPajama* (Together Computer, 2023), etc. For other criteria, such as task diversity, the analysis-based approach may not be suitable as there might not be an obvious way to track and measure such criteria in an unstructured text dataset. In such cases, an alternative strategy could be to create various synthetic text datasets in a controlled fashion and monitor the emergence of ICL in language models trained on these datasets to better understand the necessary and sufficient conditions for the emergence of ICL.

### 2.1.6 Understanding the Effect of Design Choices on ICL

How is ICL impacted by various design and training choices involved in the development of an LLM, such as model size, pretraining dataset size, pretraining compute, instruction tuning? Sensitivity of ICL to various factors, including, but not limited to the aforementioned factors, is well-known from prior literature. In particular, several studies note that ICL performance is highly sensitive to the scale of the LLM; Wei et al. (2022a); Brown et al. (2020); Kaplan et al. (2020) note that ICL is an *emergent* ability; Akyürek et al. (2022a) discover phase transitions between transformer simulating different learning algorithms based on the depth of the transformer; and Wei et al. (2023b) find that larger LLMs have stronger semantic priors (i.e. zero-shot performance is better), but also a greater propensity to allow in-context information to *override* the semantic prior. Wei et al. (2023b) further

show that instruction tuning disproportionately strengthens semantic priors; hence, an instruction-tuning *reduces* the propensity for in-context information to override the semantic prior. Meanwhile, Singh et al. (2023) argue that ICL is a *transient* phenomenon and extended training can cause ICL to dissipate in favor of "in-weight" learning (Chan et al., 2022). A deep understanding of *why* ICL is sensitive to various design and training choices, and *how* these choices affect the mechanisms behind ICL is currently lacking. In particular, improving our understanding of how various training design decisions, e.g. instruction tuning or prolonged training, impact ICL may provide us with tools to modulate the strength of ICL in LLMs. This may help mitigate some of the safety risks posed by LLMs due to their strong in-context learning abilities (Wolf et al., 2023; Millière, 2023).

> The dynamic and flexible nature of ICL is central to the success of LLMs as it allows LLMs to proficiently improve on already known tasks as well as learn to perform novel tasks. It is likely to gain an even more prominent role as LLMs are scaled up further and become more proficient at ICL. However, the black box nature of ICL is a risk from the perspective of alignment and safety, and there is a critical need to better understand the mechanisms underlying ICL. Several "theories" have been proposed that provide plausible explanations of how ICL in LLMs might work. However, the actual mechanism(s) underlying ICL in LLMs is still not well understood. We highlight several research questions that could be instrumental in addressing the understanding of the mechanisms underlying ICL.
>
> 1. Can different theorizations of ICL as sophisticated pattern-matching or mesa-optimization be extended to explain the full range of ICL behaviors exhibited by the LLMs? ↵
> 2. What are the key differences and commonalities between ICL and existing learning paradigms? Prior work has mostly examined ICL from the perspective of few-shot supervised learning. However, in practice, ICL sometimes exhibits qualitatively distinct behaviors compared to supervised learning and can learn from data other than labeled examples, such as interactive feedback, explanations, or reasoning patterns. ↵
> 3. Which learning algorithms can transformers implement in-context? While earlier studies (e.g. Akyürek et al., 2022a) argue transformers implement gradient descent-based learning algorithms, more recent work (Fu et al., 2023a) indicate that transformers can implement higher-order iterative learning algorithms e.g. iterative Newton method as well. ↵
> 4. What are the best abstract settings for studying ICL that better mirror the real-world structure of language modeling and yet remain tractable? Current toy settings e.g. learning to solve linear regression are too simple and may lead to findings that do not transfer to real LLMs. ↵
> 5. To what extent, different architectures are universal approximators 'in-context'? Can we better characterize functions that can be learned in-context by models obtained in practice? How do the pre-training data and the training objectives affect the in-context universal approximation abilities of a model? ↵
> 6. How can interpretability-based analysis contribute to a *general* understanding of the mechanisms underlying ICL? Can this approach be used to explain various phenomena associated with ICL such as why ICL performance varies across tasks, how the inductive biases of a particular architecture affect ICL, how different prompt styles impact ICL, etc.? ↵
> 7. Which properties of large-scale text datasets are responsible for the emergence of ICL in LLMs trained with an autoregressive objective? ↵
> 8. How do different components of the pretraining pipeline (e.g. pretraining dataset construction, model size, pretraining flops, learning objective) and the finetuning pipeline

(e.g. instruction following, RLHF) impact ICL? How can this understanding be leveraged to develop techniques to modulate ICL in LLMs? ↵

## 2.2 Capabilities Are Difficult to Estimate and Understand

Improving — and especially *assuring* — the safety and alignment of LLMs demands an understanding of their capabilities. More capable systems possess a greater potential to cause harm under misalignment; as argued by Shevlane et al. (2023). At the same time, some capabilities — such as recognizing ambiguity and uncertainty, and understanding how to infer humans' intent — are necessary for advancing LLM safety and alignment (respectively). This makes it critical that we correctly estimate and understand the 'capabilities' of an LLM (Mitchell, 2023). **However, there is currently no single well-established and agreed-upon conceptualization, definition, or operationalization of the term 'capabilities'. This, combined with various other factors, hinders rigorous accounting of risk, which should be grounded in an understanding of what *types* of behavior a system is capable of in general, rather than the specific behaviors a system exhibited in the particular situations in which it was tested** (Kaminski, 2023). To provide high-quality assurance of LLM safety or alignment, we need an improved scientific understanding of how to infer underlying capabilities from such limited test results.

### 2.2.1 LLM Capabilities May Have Different 'Shape' Than Human Capabilities

The capabilities of LLMs, and other AI models, are likely to be mechanistically and behaviorally distinct from the corresponding human capabilities — even when some summary statistics for the two may be closely matched, e.g. accuracy on some benchmark. We colloquially refer to this as the 'shape' of capabilities being different.[2] Adversarial examples in computer vision are a classical example of this difference — small perturbations to images that are imperceptible to humans can have drastic effects on a model based on neural networks (Szegedy et al., 2013). Within LLMs, one obvious way this manifests is as inconsistent performance across data points on which a human's performance would be consistent. For example, GPT-4 accuracy at a counting task degrades significantly when the correct answer is a low probability number (e.g. 83) relative to the cases where the correct answer is a high probability number (e.g. 100) (McCoy et al., 2023). One other way this mismatch becomes obvious is by looking at tasks that LLMs can do with much greater efficiency than humans. For example, Gemini-1.5 can learn to translate sentences from English to a completely novel language (Kalamang) in-context given instructional material (Gemini Team, 2024). However, it may take a human several weeks to learn to perform the same translations (Tanzer et al., 2023).

This mismatch in the 'shape' of capabilities can have adverse consequences. It makes it difficult for humans to simulate LLMs and predict their behavior (Carlini, 2023a). This may harm a user's trust in LLMs (c.f. Section 4.3) and can generally make assurance harder, as LLM behavior can change in arbitrary ways across the input space. Considering this mismatch, we also caution against the careless use of tests designed for humans to estimate LLM capabilities, as argued by Davis (2014). In general, this issue of mismatch indicates that conceptualizations used to describe human capabilities may be ill-suited to describe capabilities of LLMs (and AI models in general). However, the methodology used to identify human capabilities may still transfer or could be used as a source of inspiration as we hint at in the following section.

### 2.2.2 Lack of a Rigorous Conception of Capabilities

Researchers often make claims about the presence, or absence, of a capability within a model based on whether or not the model is able to carry out *tasks* that supposedly require that capability. This suggests

---

[2]Also see related discussion on how AI models learn and use different concepts and representations than humans in Section 3.4.2.

an *implicit* conceptualization of capabilities within the community that associates a model having a capability with the model being able to perform well on tasks of some particular type (Shevlane et al., 2023, Table 1). This is generally operationalized by collecting (large) number of samples representative of the task of interest into a benchmark. However, benchmark performance is highly dependent on which particular samples are chosen, i.e. what parts of input space are covered, and how densely they are sampled from. Indeed, depending on what samples they evaluate on, different works draw different conclusions about the capability of different LLMs, e.g. to use theory-of-mind reasoning (Ullman, 2023; Zhou et al., 2023a; Shapira et al., 2023; Kim et al., 2023). Conflicting claims about models' capabilities are typically adjudicated informally; new research may exhibit surprising failure modes to demonstrate that a capability is not robust, or explain away behavior that seems to demonstrate a general capability with evidence that a model's performance on a particular benchmark is due to peculiarities of the samples selected, e.g. the existence of shortcut features (Geirhos et al., 2020).

This informal treatment of capabilities is insufficient, and there is a need for more rigorous treatment for the purposes of assurance. For instance, to enable us to make and evaluate formal claims about an LLM lacking "dangerous" capabilities (Shevlane et al., 2023). To make rigorous, scientifically sound, and *general* claims about the presence or absence of a capability within a model, it is essential to establish rigorously defined and commonly agreed upon conceptualizations of capabilities (Jain et al., 2023a). More specifically, we might be interested in claims of three different types: (a) a capability is completely *absent* from a model; (b) a capability is partially *present* within a model; and (c) a capability is robustly *present* within a model. Research is needed on how to best define, operationalize, and evaluate such claims about capabilities. We will now discuss three different ideas for how to conceptualize capabilities in a way that may support such claims or otherwise improve on current practice.

**Domain Conceptualization:** One way of formalizing capabilities (for a predictive model) is as statements of the form $f|_A \approx g$, which we might read as "model $f$ has capability $g$ over domain $A$". We refer to this as the *domain conceptualization*. We conceptualize $g$ as a function that implements a desired capability (e.g. addition) on relevant inputs (e.g. real numbers), and $A$ as a subset of all relevant inputs (e.g. integers). This is reminiscent of benchmarking but explicitly specifies a set of inputs over which the model reliably exhibits a capability. Such claims might be established by evaluating model behavior on a finite set of samples (as in benchmarking) and applying some learning theory (Neyshabur et al., 2017a), using properties such as smoothness of $f$ (Dziugaite and Roy, 2017; Neyshabur et al., 2017b; Arora et al., 2018), and/or methods such as certified robustness to prove that $f$ approximates $g$ well not only on sampled points but on unseen points as well, e.g. within some volume of input space (Cohen et al., 2019; Carlini et al., 2022). Benchmarks could also be designed in a theoretically-motivated way to support such general claims (Yu et al., 2023a; Shirali et al., 2022),

**Internal Computations Conceptualization:** Another alternative conceptualization of the capabilities of a model is as functions implemented *within* a model, e.g. between hidden layers. We call this the 'internal computations conceptualization' of capabilities. For instance, we might identify capabilities with 'circuits', i.e. computational subgraphs of a neural network (Olah et al., 2020). Here, we might say a model possesses a capability for addition if there is any circuit that can perform addition, regardless of when or whether the neurons in this circuit are actually active. This might enable stronger assurance regarding the *potential* for an LLM to exhibit a dangerous behavior after fine-tuning or other methods of eliciting such behavior. Analysis of circuits has been used to provide conclusive evidence regarding models possessing specific capabilities (Wang et al., 2022). However, there are currently technical issues with applying this conceptualization in practice, which may limit its utility (see Section 3.4 for further details). This conceptualization could also be combined with the domain conceptualization to identify computational elements *within* an LLM that robustly implement functions over limited sets of inputs.

**Latent Factors Conceptualization:** The aforementioned 'internal computations conceptualization' view of capabilities is inspired by neuroscience. Analogously, another view to conceptualize capabili-

ties – which we call the 'latent factors conceptualization' – could be to take inspiration from the field of psychology, in particular, psychometrics. These fields are primarily concerned with characterizing and quantifying the fundamental processes involved in variation in human cognitive abilities. Like the 'internal computations conceptualization', the latent factor conceptualization views behavior as being produced through the application of multiple capabilities, but does not attempt to ground these capabilities in internal computations. A commonly used technique in psychometrics is factor analysis. Under this conceptualization, capabilities are the "factors", or latent variables, that explain variation in measurements *across* subjects (Carroll, 1993). This could be done by taking a population of different LLMs and extracting the factors that explain the most variance in performance across examples in a benchmark (or benchmarks), using techniques such as factor analysis (Burnell et al., 2023a) or other psychometric techniques (Wang et al., 2023d). Machine learning methods for discovering and disentangling latent factors of variation could also be explored. Intuitively, these factors of variation could be viewed as a 'basis' of capabilities, and commonalities between capabilities could help interpret model behavior in a way that is predictive of behavior on unseen examples, tasks, or domains. One drawback of such methodology is that identified 'factors' may not be amenable to natural interpretation as 'capabilities' and attempts to interpret them could induce false beliefs about the models in model developers and evaluators (Chang et al., 2009).

All the above conceptualizations vary in rigor, functionality, and tractability, and their pros and cons are not well understood. It is also possible that other, better conceptualizations, exist that may be more suitable for understanding and explaining the behaviors of LLMs. An ideal conceptualization of capabilities should allow making, and verifying, mathematically rigorous claims pertaining to presence, or absence, of a certain capability within a model while being tractable and easily applicable to any arbitrary model. However, conceptual progress would be valuable even absent practical methods. On the whole, the advent of LLMs presents us with an opportunity to rethink our conceptualization of capabilities. In the short term, it is also critical that researchers clearly communicate *their* conceptualization of what a capability is when making claims related to the presence, or absence, of capabilities, to avoid the literature getting littered with seemingly contradictory results which are easily explained away due to the differences in the object being studied.

### 2.2.3 Limitations of Benchmarking for Measuring Capabilities and Assuring Safety

As discussed above, benchmarking is one of the most common forms of evaluation, especially for estimating the capabilities of a model. There are several reasons why benchmarking-based evaluations may misestimate the capabilities of an LLM. Firstly, benchmark performance cannot distinguish between the cases of a model failing to function well on a task due to (a) capability failure, i.e. model being incapable of the capability at all, and (b) intent alignment failures, i.e. model failing because of failure to understand or adopt human intent (Chen et al., 2021a, Appendix E.2). Relatedly, safety finetuning methods often work by suppressing model capabilities (Jain et al., 2023a; Wei et al., 2024). In such cases, a model may have a given capability, but it may not readily demonstrate it due to the influence of safety finetuning. In such a case, benchmark performance would indicate the absence of the capability when intuitively that is not the case. Secondly, a model could function well on two seemingly distinct benchmarks using the same underlying capabilities (Merullo et al., 2023b). This might lead us to overestimate, or 'overcount' the model's capabilities. Thirdly, benchmark performance may produce unreliable assessments of capabilities that are present within a model, but are *not* robust; in such cases, performance may depend heavily on the particular distribution used by a benchmark (Teney et al., 2020; Burnell et al., 2023b; Bao et al., 2023). Fourthly, because benchmarks primarily report average statistics (e.g. accuracy), they are often not very informative about an LLM's performance on *particular* test examples. The first 3 issues are all problematic mostly because they may lead to incorrect inferences about a model's behavior on test examples, the fourth issue is more about making more *informative* inferences about behavior — making predictions about behavior (e.g. performance) at the example level rather than the dataset/distribution level as is implicitly the case with benchmarks.

Separately, the fact that currently the most capable models are provided as a service via an API largely limits our ability to independently audit and evaluate them for safety (La Malfa et al., 2023).

There is a need to develop principled solutions to the aforementioned issues. We need to develop robust machinery that may help us distinguish between capability failures, intent-alignment failures, and intentional failures on the model's part due to the effects of safety finetuning. Among these intent-alignment failures are the most egregious to avoid. There is in general a need for elicitation protocols that reliably and consistently elicit the capabilities of interest, even in the case of intent-alignment failures. Fine-tuning and prompting often reveal behavior indicative of novel capabilities, but cannot be used to rigorously establish that an LLM does not possess a particular capability. There is also a need to improve the structure and granularity of benchmarking to better understand model capabilities.

The suggestions we make in Section 2.2.2 may help address the issues mentioned above. First, the 'domain conceptualization' can limit incorrect inferences by being precise about the domain over which a capability is expected to be robust, and can make more precise inferences by accounting for which capabilities' domains a test example belongs to. Likewise, both the 'circuit conceptualization' and the 'latent factors conceptualization' could help determine which capabilities are at play when a model processes a particular example, and thus better attribute behavior to particular capabilities and draw conclusions about which capabilities are (likely to be) used on a given test input. The circuit conceptualization, being more detailed and grounded, could make it more straightforward to avoid misestimation of capabilities, e.g. by distinguishing between capabilities that are robustly, but rarely applied (e.g. due to misalignment) vs. those that are simply not robust.

### 2.2.4 How Can We Efficiently Evaluate Generality of LLMs?

The majority of evaluations of LLMs are domain-specific evaluations, e.g. 'language understanding' (Hendrycks et al., 2020a), 'mathematics' (Hendrycks et al., 2021b), 'social knowledge' (Choi et al., 2023a), 'medical knowledge' (Singhal et al., 2023a), 'coding and tool manipulation' (Xu et al., 2023a). A fundamental limitation of evaluating LLMs in this way is that we may undercount LLM capabilities in domains where corresponding benchmarks are not available (Ramesh et al., 2023). It is also logistically difficult to evaluate LLMs across a large (e.g. combinatorial) number of domains or tasks. There is a need for work to consider alternative means to evaluate generality (i.e. how general-purpose the model is) (Casares et al., 2022), and generalization across domains, of LLMs. For example, procedural evaluations like Skill-Mix (Yu et al., 2023a) could be designed that may evaluate the compositional generalization skills of LLMs. Alternatively, given the fact that we now have a population of LLMs available, finding differences and similarities in their performance across domains could inform what evaluations are most informative for evaluating the generality of LLMs (Ye et al., 2023). Mechanistic investigations of LLM capabilities could aim to discover capabilities that are reused across tasks (Todd et al., 2023), or discover and explain other capabilities (like in-context learning, see Section 2.1) that may underlie general-purpose behaviors of LLMs.

It may also be useful to create a taxonomy of the capabilities of LLMs, similar to how psychologists have created taxonomies of human capabilities (Fleishman et al., 1984). In fact, some taxonomies are already emerging implicitly within the literature. For example, LLM developers report across different types of benchmarks (corresponding to different domains and capabilities), e.g. coding, question answering, mathematical reasoning, common-sense reasoning, performance over long-contexts (OpenAI, 2023a; Gemini Team, 2023; Anthropic, 2023b; Jiang et al., 2024a). However, these 'taxonomies' are ad-hoc, and have little to no theoretical basis. There is a need for work to establish theoretically grounded taxonomies that may provide better organization and understanding of LLM capabilities. Furthermore, taxonomies of human capabilities often arrange capabilities in a hierarchical fashion — making the dependencies between capabilities obvious (Schneider and McGrew, 2012). Chen et al. (2024a) show that similar dependencies between capabilities (or in their terminology, skills) exist for LLMs as well, and respecting these dependencies during training (i.e. defining an appropriate curriculum over skills) helps

achieve better learning outcomes. However, no work has attempted to document these dependencies at a large scale.

### 2.2.5 Scaffolding Is Not Sufficiently Accounted for in Current Evaluations

Even in the simplest use cases, an LLM should be viewed as a multi-party system consisting of the LLM itself and an elicitation protocol (e.g. the prompt or finetuning process). In the extreme cases, the LLM can be a part of a much larger system having access to external memory, various types of tools and various types of learning signals (e.g. feedback from other LLMs) (Wang et al., 2023a; Park et al., 2023a). We collectively refer to these mechanisms that enhance the capabilities of an LLM as *scaffolding.*

In addition to being highly capable models, LLMs are also highly efficient learners. This learning can occur via fine-tuning, supervised learning-style in-context learning, instructions, explanations, etc. As a result, by efficiently designing the elicitation protocol, a designer can significantly alter the capabilities and the behavior profile of an LLM. For example, an LLM capable of performing addition can be given the capability of performing multiplication by exhaustively explaining and demonstrating the algorithm to multiply numbers using additions within the prompt (Zhou et al., 2022). Similarly, fine-tuning on a small number of carefully chosen examples can cause the LLM to act in a highly polite way (Zhou et al., 2023b). In such cases, it is no longer clear whether the protocol revealed an existing capability or induced a novel capability within the LLM (Stechly et al., 2023). This lack of distinction can cause overestimation of capabilities by mistakenly attributing a capability to the LLM when in fact the LLM did not have the capability (Stechly et al., 2023); rather the capability may have been *quickly* learned on the go due to the efficient design of the elicitation protocol. There is a need for theoretical work to characterize and distinguish between capabilities that are present within an LLM (and are elicited), versus capabilities that are efficiently learned on the fly. Tools, techniques, and concepts from information theory may be used for this purpose (Zhu and Rudzicz, 2020).

A similar attribution problem occurs when an LLM is combined with other systems, e.g. given access to external tools it can assign tasks or given feedback from environment or verifiers (Valmeekam et al., 2023). Prior work suggests that such systems can outperform an LLM acting on its own (Mialon et al., 2023; Wang et al., 2023a). In general, in deployment, LLMs are deployed as parts of *larger* system, where other components of the system may perform various jobs. In such cases, it is not clear how the capabilities being demonstrated by such systems should be attributed to the LLM vs. the other system components involved. Various concepts exist within game theory on how the utility of a system may be distributed between its components, e.g. Shapely value, core (Shoham and Leyton-Brown, 2008, Chapter 12). It is an open problem to evaluate which concept(s) are most suitable for attributing the capabilities of LLM-based systems to the capabilities of LLM and the capabilities of other components present in the system.

> In order to assure safety, we need a calibrated understanding of the capabilities of the models. However, this is currently made difficult by various challenges. Firstly, the capabilities of the model seem to have a different 'shape' compared to human capabilities; but this difference is not currently well-understood. Secondly, there is no well-established conceptualization of capabilities. Thirdly, we do not have sufficiently reliable methods to assess the generality of LLMs — i.e. how general-purpose a given model is. Lastly, it is not clear how to account for scaffolding in LLM capabilities and distinguish between the cases in which an elicitation protocol reveals a capability already present within an LLM versus the cases in which LLM acquires the capability due to the efficient design of the scaffolding and elicitation protocol.
>
> **9.** How can we understand the differences in the 'shape' of capabilities of humans and other AI models? What are the implications of these differences? ↩

10. What is the right conceptualization of capabilities for LLMs? Can we formalize the three conceptualizations of capabilities presented here (domain conceptualization, internal computations conceptualization, latent factors conceptualization), and understand their relative merits and demerits? ↩

11. How can we draw reasonable general insights from behavioral evaluations? In particular, how can we prove that a given model does not have a capability of interest, without exhaustively evaluating the model for that capability on the whole input space? ↩

12. Can we develop methods — e.g. based on factor analysis or other unsupervised learning methods — to automatically discover capabilities by 'decomposing' a model's (potential) behavior into something like a 'basis' of capabilities? ↩

13. When performing an evaluation using a benchmark, how can we separate observed failures of a model into 'capabilities failure' (i.e. model failing because it truly lacks the relevant capability) from 'alignment failures' (i.e. model failing despite having the capability because it was not correctly invoked')? ↩

14. How can we develop elicitation protocols that reliably and consistently elicit the capabilities of interest? ↩

15. How can fine-grained benchmarking be used to identify the precise shortcomings of a model and make more useful, detailed predictions about test behavior? ↩

16. How can we evaluate the generalization of LLMs *across* domains? Can procedurally defined evaluations be used for this purpose? ↩

17. Can we formalize or operationalize statements like "the model is using the same capability when performing these two tasks"? How can we use this to more efficiently evaluate LLMs across domains? ↩

18. How can a practitioner, who is resource-constrained, choose a small number of evaluations to perform on LLM to efficiently evaluate the general-purpose capabilities of LLMs? Alternatively, how can we determine which evaluations are most informative to perform when trying to compare the capabilities of two given LLMs? ↩

19. Can we use mechanistic interpretability techniques to discover capabilities that are reused across tasks? ↩

20. How can we understand the dependencies between LLM capabilities? Can we create theoretically grounded taxonomies of LLM capabilities that may help us assess the generality of LLMs efficiently? ↩

21. How can we distinguish between revealed capabilities (capabilities inherent in an LLM) vs. learned capabilities (capabilities that are exhibited because of the highly optimized nature of the elicitation protocol)? Can tools and concepts from other fields, such as information theory and cognitive science, be used to help make this distinction? ↩

22. How can we precisely characterize the contribution of the LLM to behaviors demonstrated by an LLM-based system (e.g. an LLM with access to external tools)? Can we use concepts developed in game theory and other literature on multi-agent systems for this purpose? ↩

## 2.3 Effects of Scale on Capabilities Are Not Well-Characterized

Increasing the scale (parameters, compute, and data) of LLM training predictably results in an overall more performant model in accordance with well-established scaling laws (Kaplan et al., 2020). However, specific capabilities of LLMs are often highly difficult to predict (Bowman, 2023), and may show so-called 'emergent' behavior (Wei et al., 2022a). **This combination of high-level predictability and low-level unpredictability is a source of risk: the former enables easy progress via scaling, the latter makes it difficult to anticipate and precisely characterize the risks associated**

**with the development and deployment of more performant models** (Ganguli et al., 2022). From a scientific standpoint, this highlights a significant gap in our knowledge of how LLMs learn and acquire capabilities. To address this gap, we will need to develop a deeper understanding of scaling laws, understand how scaling impacts learned representations, identify the limits of scaling, and work towards formalizing, forecasting, and explaining emergent behaviors in learning.

### 2.3.1 Understanding Scaling Laws

Large language model training has been found to follow scaling laws for aggregate loss that are consistent across many orders of magnitude of resource scaling (Kaplan et al., 2020; OpenAI, 2023a). The factors explaining this scaling law behavior, however, remain poorly understood. As a result, it is unclear to what extent different aspects of these laws are universal and to what extent they are sensitive to different aspects of the training pipeline which might change in the future. Explanations can address both the functional form of scaling laws (e.g. power law scaling vs. exponential scaling) and the specific scaling parameters of the functional form found in particular experiments (e.g. the exponent in power law scaling).

Prior work has studied the impact of scaling on learning under various theoretical setups. These works, as explained in detail later, often differ in their prediction of power law exponents. These seemingly contradictory predictions arise from differences in the scaling setup; specifically from whether the resources which are *not* being scaled are small or large in comparison to the scaled resource (Bahri et al., 2021). Indeed, current literature can be quite clearly demarcated based on this distinction. Borrowing the terminology from Bahri et al., the *variance-limited* regime considers the case where we are asymptotically scaling one resource (model size or data) while the other resource is fixed at some finite level. In this case, most theories predict rapid power-law scaling (or even exponential scaling, in some cases) to a saturated level of performance, based on concentration arguments. The *resolution-limited* regime is when the unscaled resource is infinite, or is much larger than the scaled resource. Here, the scaling exponent captures the effect of increasing the "resolution" of the learner (either by allowing it to use more data points or more parameters to fit increasingly fine aspects of the distribution), and is highly sensitive to properties of the data distribution. This is the regime most modern work on LLM scaling is concerned with.

Theoretically, the variance-limited regime of scaling laws — in which the amount of data is scaled and model size is fixed and finite — is the best studied one. This is the typical subject of the vast literature on learning curves — see Viering and Loog (2022) for a survey. For example, classic PAC theory shows that power law data scaling with an exponent of $-1$ (or $-1/2$ in the unrealizable zero-one error setting) is optimal for every learnable task, and is achievable by empirical risk minimization (Blumer et al., 1989). Other theoretical models of (bounded-capacity) learning curves include the universal learning theory of Bousquet et al. (2021) and approaches based on statistical mechanics on non-worst-case learning curves in the 'thermodynamic limit' (Seung et al., 1992; Watkin et al., 1993; Amari, 1993; Haussler et al., 1994). However, as these aforementioned theoretical models assumed that the model size is fixed, and only data is scaled, they provide limited insights regarding scaling laws for LLMs for which model size and dataset size grow together, in which case, gentler scaling exponents of roughly $-1/10$ to $-1/20$ have been observed (Kaplan et al., 2020). However, they are still predictive of LLM scaling in the cases where the model is set to be fixed and finite. For example, for a fixed and finite model size, the joint data-parameter functional form of Kaplan et al. (2020) is asymptotically a power law with exponent $-1$ (approaching the loss at which models of the given size saturate).

At least for now, both the data and the number of parameters used in training continue to increase over time, so the variance-limited regime, which assumes one of these resources is capped, is of limited relevance. (Though it may be relevant in scenarios in which the amount of data available for training is limited.) Instead, resolution-limited scaling — the scaling regime in which the unscaled resource is infinite or is much larger than the scaled resource — provides a more accurate picture for capabilities

forecasting. The existing literature contains at least three distinct explanations and theoretical models of resolution-limited scaling. The first of these, the *manifold explanation* (Sharma and Kaplan, 2022), posits that scaling laws emerge from something akin to interpolation on the data manifold. Sharma and Kaplan (2022) empirically test this explanation by estimating the intrinsic dimension for some small datasets and showing that this is predictive of the model size scaling exponent. However, this model also predicts that per-example performance should increase monotonically, which Kaplun et al. (2022) note is not the case in practice.

A related theory is the *kernel spectrum* explanation provided by Bordelon et al. (2020) and Spigler et al. (2020), who derive resolution-limited scaling in the setting of kernel regression. Specifically, they show that for kernel methods, such as learning with an infinitely wide network in the neural tangent kernel limit (Jacot et al., 2018), power-law decay in the kernel spectrum results in power law scaling in the loss. Finally, a third explanation, based on *long tails* in the data, has been introduced by Hutter (2021) and extended by Michaud et al. (2023), Dębowski (2023), and Cabannes et al. (2023). These authors construct toy models in which gentle power law scaling emerges when the data distribution has a long tail of sub-components that must be learned independently, an assumption that is especially natural in the domain of natural language data.

Given the fragmented state of the literature, several fundamental questions are currently unanswered. Firstly, can the different proposed explanations for power law scaling in the resolution-limited regime be unified? Bahri et al. (2021) provide a connection between the manifold dimension explanation and the kernel spectrum explanation; perhaps it is possible for all three theories (including the long tail theory) to be subsumed by a single meta-explanation which could handle a wider range of settings.

Secondly, the variance-limited versus resolution-limited dichotomy entirely ignores the scaling regime that is most important for forecasting capabilities: *compute-efficient scaling*, where data and model size are scaled jointly (Hoffmann et al., 2022). What is a good theoretical model for this setting? Is there an explanatory theory that considers all three regimes as special cases of a more general joint data-parameter scaling setting?

Thirdly, what is the role of feature learning, and optimization more generally, in scaling laws? In language modeling, representations learned early in training enable more complicated aspects of the data to be learned later in training (see Abbe et al. (2021) for a synthetic case study of this behavior). Existing scaling law explanations, however, essentially treat the data as "flat", and ignore the influence of hierarchical structure on scaling behavior. In particular, we would like to encourage further work on scaling laws that account for various properties of language data, for example, burstiness (Chan et al., 2022) or fractal nature (Alabdulmohsin et al., 2024). Relatedly, all the current models of scaling laws presume that the data distribution that is being learned is held constant throughout learning. However, in various settings of interest, e.g. reinforcement learning, data distribution changes over time. However, despite this non-stationary nature of data distribution, various works have reported scaling laws for various reinforcement learning settings (Hilton et al., 2023; Tuyls et al., 2023; Team et al., 2023; Obando-Ceron et al., 2024). Thus, the development of appropriate theoretical models for scaling laws for cases in which the data distribution is considered to be non-stationary is an open research question at the moment.

Finally, one fundamental question that future investigations ought to answer is to what extent learning curve exponents are fundamentally bounded by the data distribution. This can help inform the possibility of hypothetical future architectures that might scale better than contemporary transformers for language modeling. A related open question is what properties of a data distribution affect the predictability of scaling behavior on that distribution. This is particularly relevant to the discovery of task-specific specific laws (also see Section 2.3.5). On some tasks, e.g. code-generation, power law scaling across 10 orders of magnitude of compute has been observed (Hu et al., 2023a; OpenAI, 2023a,

Figure 1). On the other hand, many downstream capabilities display irregular scaling curves (Srivastava et al., 2022), or non-power law scaling (Caballero et al., 2023).

### 2.3.2 Effect of Scaling on Learned Representations

Most work on scaling focuses on performance on benchmarks. There is far less work on the question of how learned *representations* change with scale. This is a particularly pertinent question for the viability of interpretability techniques that aim to understand the internal operations of LLMs like mechanistic interpretability and probing techniques (c.f. Section 3.4). The first question relates to the universality hypothesis of representations — do different neural networks trained on the same task learn the same representations (Olah et al., 2020)? The current evidence indicates that the strong version of the universality hypothesis is false, for example, Chughtai et al. (2023) show that different neural networks learn different representations in different orders even when the architecture and data order are kept the same. This is supported by evidence contained in other studies as well (McCoy et al., 2019; Wang et al., 2018). However, there is increasing evidence that *some* feature representations are indeed *universal* and learned by different neural networks trained on the same task (Li et al., 2015; Bansal et al., 2021; Gould et al., 2023). The key unknown question is whether the proportion of universal representations increases, or decreases, with scale. Vyas et al. (2023) argue that language models learn similar features across different width sizes. In the vision setting, Nguyen et al. (2020) found that when networks are sufficiently large (in terms of width or depth) in comparison to the training set size, they can be partitioned into contiguous blocks of layers with representations within each block being similar across different trained models.

A related, but distinct, question is whether learned representations converge to, or diverge away, from the representations used by humans with increasing scale (Sucholutsky et al., 2023)?[3] In some cases, the behavior of larger LMs does appear to be more consistent with human behavior (Chiang and Lee, 2023; Park et al., 2023a; Zhu et al., 2024), however, this consistency tends to break down at the edges; for example, LLMs do not always show human-like biases in their responses (Aher et al., 2022; Tjuatja et al., 2023; Hagendorff et al., 2023). As such, there may not exist a clean answer to the question above. However, it might still be useful to understand if there are types of representations of concepts that we can expect LLMs of sufficient scale to share with humans. And if so, how may it help us predict the behavior of LLMs?

Another open question pertains to the changes in representation structure with scale. Specifically, whether larger scale models are more or less likely to have representations that exhibit linear characteristics, such as the famous "`King - Man + Woman = Queen`" example (Mikolov et al., 2013).[4] The idea that LLMs encode high-level concepts linearly in the representation space of the model has been termed the "linear representation hypothesis" (Park et al., 2023b). The mathematical field of *representation theory* studies how abstract algebraic structures such as groups can represented in terms of linear transforms, and might help identify and understand such limitations. Chughtai et al. (2023) present evidence that neural networks do in fact use representation theory to represent data with a group structure. Understanding how scale influences the structure of representations would provide insights into the applicability of interpretability methods for larger models, which implicitly or explicitly assumes linearity of representations.

### 2.3.3 Limits of Scaling

One framework for making sense of advances in machine learning is to decompose the progress in capabilities into performance improvement due to algorithmic innovation and the performance improvement due to scaling up the resources of training (Hernandez and Brown, 2020; Erdil and Besiroglu, 2022). Prior work has shown that simple scaling up can cause the acquisition of novel capabilities within the

---

[3]Also see Section 3.4.2

[4]Notably, Nissim et al. (2020) found that the closest vector to `King - Man + Woman` is actually `King`; `Queen` is the *second* closest.

model (Kaplan et al., 2020; Wei et al., 2022a). This raises the question of whether there exists a *scale ceiling* i.e. capabilities that can not be acquired by a model regardless of how much it is scaled further. If so, what capabilities are these? For example, it is unclear to what extent LLMs may acquire reasoning and abstraction capabilities via scaling alone (Mitchell et al., 2023; Saparov et al., 2023).

Prior work has argued that some of the most critical limitations of current LLMs are unlikely to be resolved by simple scaling. Kirk and Krueger (2022) argue that causal confusion, i.e. learning of spurious correlations present within the data may be unavoidable in a purely offline learning setup. This claim has mixed support within the literature; while Zecevic et al. (2023) argue that LLMs can not be causal, Lampinen et al. (2023) point out that LLM training data contains many examples of interventions, outcomes, and explanations that may support the learning of active causal strategies by the LLM. Kalai and Vempala (2023) and Xu et al. (2024a) both argue that 'hallucinations' in LLMs are not fixable via scaling. Several studies have argued that resistance to jailbreaking may not improve with scale (Wei et al., 2023c; Wolf et al., 2023; Millière, 2023). In general, robustness to adversarial examples may not improve with scale, as argued by Frei et al. (2023) and Debenedetti et al. (2023).

One extreme sense in which scaling can be limited for LLMs is *inverse scaling*, where performance *decreases* with scale (McKenzie et al., 2023). McKenzie et al. (2023) ran a contest to solicit tasks exhibiting inverse scaling across different LLM architecture families. The results were mixed — for almost all tasks inverse scaling was later found to be reversed at the largest compute scale measured (which was for the PaLM family of models), resulting in U-shaped scaling curves Wei et al. (2022b). We can also define inverse scaling more broadly, to include increasing frequency of undesirable *behaviors* with scale, not just decreasing frequency of correct responses. Using this perspective, Perez et al. (2022a) observe inverse-scaling for sycophantic behavior, i.e. larger models have a more pronounced tendency to mimic the political views of their interlocutors. This suggests that the human feedback process used for alignment training is misaligned (see Section 3.2), which might cause other related inverse scaling problems. Still, it is currently unclear whether there exist other undesirable behaviors that undergo inverse-scaling, and there is no reliable way of determining whether such undesirable behaviors will also undergo U-shaped scaling (or not).

### 2.3.4  Formalizing, Forecasting, and Explaining Emergence

Some capabilities when plotted on a scaling curve appear to emerge suddenly past some scale. A large number of capabilities, including key capabilities such as instruction following and chain-of-thought reasoning, have been argued to be emergent (Wei et al., 2022a). On the other hand, other studies have argued that some capabilities may appear emergent due to the harsh nature of the evaluation measures being used which do not reward partial correctness (Schaeffer et al., 2023; Srivastava et al., 2022). However, none of these studies precisely define emergence, and their use does not accord with established use across other fields such as complex systems and physics. This indicates a need for greater clarity and careful formalization of emergence in the context of machine learning.

In any case, so-called 'emergent' capabilities, e.g. those identified by Wei et al. (2022a), may be particularly challenging to predict and forecast (Ganguli et al., 2022), and are hence disconcerting from a risk assessment perspective (Kaminski, 2023). Developing a mechanistic understanding of factors that contribute to abrupt improvement in performance may yield critical insights regarding the predictability of 'emergent' capabilities. One such factor that has been identified so far is the compositional nature of a capability, i.e. a capability being composed of other capabilities. Arora and Goyal (2023) and Okawa et al. (2023) show that if a capability is a composition of another set of capabilities (called skills) that witness smooth and predictable scaling, the scaling curve of a capability will look emergent due to a multiplicative dependence on the ability to perform the skills underlying it — Okawa et al. (2023) call this the multiplicative emergence effect. The results of the aforementioned papers indicate that if a valid skill decomposition of a seemingly emergent capability is available and these skills follow predictable dynamics, we can accurately predict the model's progress on the capability itself. Arguably,

the bottleneck here is that for several capabilities of interest, we are unlikely to have an accurate decomposition available (Barak, 2023). Further research is needed to understand how such decomposition can be achieved for a capability of interest and whether given an approximate decomposition, one can predict at what scale a model learns a compositional capability.

Schaeffer et al. (2023) provide anecdotal evidence that progress measures can be designed for some of the emergent capabilities identified in the literature which smoothly track the so-called emergent capability. This is similar to the identification of continuous progress measures in the context of grokking of capabilities (Nanda et al., 2022; Barak et al., 2022), where models exhibit a sudden shift from memorization to generalization. However, we assert that the existence of soft progress measures indicates that learning dynamics are predictable (*if* the appropriate progress measure can be identified pre-hoc), but does not invalidate the perspective that some capabilities are emergent (Barak, 2023). It is also critical to ensure that any progress measures identified faithfully track the capability of interest, as, otherwise, there may be a measure whose dynamics are predictable, but in fact does not faithfully capture progress on learning the capability. It is not clear whether the progress measures proposed in the literature are faithful in this sense or not. Finally, we emphasize that identifying such progress measures can be highly non-trivial and current evidence is insufficient to indicate that suitable progress measures exist for all capabilities that are claimed to be emergent (Wei, 2022). Interpretability methods have previously been used to find progress measures for grokking (Nanda et al., 2022; Chughtai et al., 2023), and could be used for discovering progress measures that underlie emergent capabilities as well. It may also be helpful to study the learning dynamics in a systematic way as done in some prior work (Hu et al., 2023b; Chen et al., 2024b; Hoogland et al., 2024; Edelman et al., 2024). Hoogland et al. (2023) argue that singular learning theory (Wei et al., 2022c; Watanabe, 2024) may be particularly useful for this purpose.

There is also a need for work to tighten and formalize the definition of 'emergent' capabilities. The current definition of emergent capabilities arguably points to an abrupt improvement in model performance on a scaling curve as a necessary — but not sufficient — condition for a capability to be emergent. This notion of emergent capabilities is also somewhat distinct from the notion of emergence in other fields. For example, in physics, emergence is often associated with the formal notion of phase transitions: discontinuities in some property of a system (or its derivatives) considered as a function of some parameter (e.g. scale). In physics, such discontinuities yield distinct "phases" that are governed by different physical laws (e.g. liquids vs. gases) (Huang, 2008; Grimmett, 2018). However, seeking inspiration from natural sciences and grounding emergence in "phase transitions" may be too constraining; e.g. it is unclear if in-context learning (ICL) is truly a phase transition in the technical sense, but it is intuitively clear that ICL yields a *qualitatively* different set of capabilities in larger models. There is a need to establish desiderata to further ground emergence from the context of prediction of capabilities and a consensus is needed on what evidence is sufficient to claim a capability is truly emergent — (seemingly) discontinuous scaling curves are perhaps insufficient for this purpose.

### 2.3.5 Better Methods for Discovering Task-Specific Scaling Laws

Power-law-based scaling laws are not always suitable for predicting performance on narrower metrics, which can exhibit inflection points and even non-monotonic scaling. Caballero et al. (2023) present a generalization that can model such phenomena, building on Alabdulmohsin et al. (2022). Other modeling approaches should be evaluated and could borrow from approaches to model learning curves (Viering and Loog, 2022). To improve data efficiency, predictions could incorporate additional information, such as scaling performance on other tasks, performance on individual examples (Kaplun et al., 2022; Siddiqui et al., 2022), or other indicators of how learning/scaling is progressing. Probabilistic modeling approaches such as Gaussian processes (which Swersky et al. (2014) use to model learning curves) may also be useful in providing scaling estimates alongside the uncertainty estimate. Developing better evaluation measures with higher resolution, and better methodologies overall to estimate capa-

bilities present in a model that are not fully mature yet, can be particularly instrumental in providing better estimates of capabilities of future models (Hu et al., 2023a).

It would also be useful to clarify the purpose of task-specific scaling laws. Is the goal to forecast capabilities as accurately as possible in the short-term, when scaling resources are only slightly higher than their current levels? To provide accurate longer-term predictions? To provide interpretable parametric fits that enable quantitative comparison of learning algorithms and can distinguish qualitatively distinct scaling regimes?

If the goal is indeed to forecast capabilities as accurately as possible, then clear desiderata should be established prescribing the range over which the task-specific scaling laws should extrapolate with high accuracy. The evaluations of Alabdulmohsin et al. (2022) and Caballero et al. (2023) demonstrate extrapolation to values of the scaled resource which are twice as large as those used for fitting the functional form, but it is unclear how accurate they are beyond this limited horizon. Stumpf and Porter (2012) suggest, in a broader scientific context, that proposed power law fits ought to be considered scientifically useful only if they extrapolate over at least two orders of magnitude; this stringent desideratum may be fruitful in the task-specific scaling law context (even for functional forms besides simple power law scaling). Moreover, as more and more tunable parameters need to be introduced to functional forms in order to improve extrapolation accuracy, the interpretability advantage of parametric fits over non-parametric methods (e.g. Gaussian processes) becomes questionable, and the risk of spurious interpretations rises. Broadly speaking, there are several distinct reasons task-specific scaling laws might be useful, and there is a need to develop criteria for assessing these laws which distinguish between the different use cases. (It may even be more terminologically precise in some cases to think of scaling fits as just "fits" rather than as "laws".)

There remain many challenges that hinder our ability to predict which capabilities LLMs will acquire with continued scaling, and when. We continue to lack a robust explanation of why scaling works, and to what extent the scaling laws we have discovered are universal. Additionally, there has been minimal exploration into how scale influences the representations learned by the models and whether certain capabilities cannot be acquired through scaling alone. Furthermore, our understanding of the factors that underlie abrupt performance improvements on certain tasks is lacking, and our methods for discovering task-specific scaling laws are inadequate.

23. Can the different explanations (manifold explanation, kernel spectrum explanation, long tail theory) of power-law scaling in the resolution-limited regime be unified? ↩

24. What is a good theoretical model for *compute-efficient* scaling, where data and model size are scaled jointly? ↩

25. What is the role of feature learning in scaling laws? Can we develop models of scaling laws that account for the computational difficulty of learning the given task? ↩

26. What is an appropriate theoretical model to explain scaling observed in reinforcement learning settings where the data distribution is not stationary? ↩

27. To what extent scaling law exponents are fundamentally bounded by the data distribution? ↩

28. What properties of a task (and its relation to the full training distribution) affect the predictability of its scaling behavior? ↩

29. To what extent does scaling a model increase the 'universality' of its representations? ↩

30. Does increasing scale cause model representations to converge to, or diverge away from, human representations? In other words, does representation alignment between human representations and model representations increase or decrease with scale? ↩

31. How does scale impact the structure of the representations? Does scaling cause the structure of model representations to become more, or less, linear? To what extent is the linear representation hypothesis true in general? ↩

32. How can we determine whether a given capability is below or above the *scale ceiling*, i.e. whether simple scaling up the model (and/or data, compute) would enable the model to learn that capability or not? ↩

33. To what extent are the issues faced by current LLMs (causal confusion, 'hallucinations', jailbreaks/adversarial robustness, etc.) likely to be resolved by further scaling? ↩

34. How do we determine if the inverse-scaling behavior of a capability will be reversed with further scaling (i.e. result in a U-shaped curve)? How can we predict threshold points at which scaling behaviors change shape? ↩

35. What factors may explain *abrupt* improvements in performance associated with emergent capabilities? To what extent does the multiplicative emergence effect (Okawa et al., 2023) explain the emergent capabilities of LLMs that have been observed in practice? ↩

36. How can we discover valid decompositions of various compositional capabilities of interest and assess the accuracy of such decompositions? How can the emergence of compositional capabilities be predicted based on the learning dynamics of its decomposed capabilities? ↩

37. What is an appropriate formalization of "emergent capabilities" in the context of LLM scaling? How can we apply it to understand which sorts of novel phenomena are likely to be (un)predictable? ↩

38. Can we discover progress measures that may explain emergent capabilities, e.g. by using interpretability methods? How can we establish the faithfulness of such progress measures to ensure that they can be used to *predict* the emergence of the capability of interest? ↩

39. Can we develop better methods for modeling task-specific scaling? E.g. by conditioning on additional information, using probabilistic techniques, or by developing evaluation measures with higher resolution. ↩

40. Can we clarify the purpose of task-specific scaling laws? In order to be useful for forecasting capabilities, what is the minimum range over which a task-specific scaling law must extrapolate accurately? ↩

## 2.4 Qualitative Understanding of Reasoning Capabilities Is Lacking

Reasoning — the process of drawing conclusions from prior knowledge — is a hallmark of intelligence. Prompt programming techniques (Reynolds and McDonell, 2021), such as *chain-of-thought* (CoT) prompting (Wei et al., 2022d), scratchpad prompting (Nye et al., 2021), or "let's think step-by-step" (Reynolds and McDonell, 2021; Kojima et al., 2022) enable language models to perform reasoning tasks with impressive accuracy. Careful studies to evaluate the behavior of LLMs on "out-of-distribution" (OOD) reasoning tasks have revealed that LLMs exhibit *some* reasoning behavior (Saparov et al., 2023; Wang et al., 2023e; Dziri et al., 2023), and that performance on reasoning tasks improves with scale (Wei et al., 2022d; Saparov et al., 2023; Valmeekam et al., 2022). However, the reasoning capabilities of even the largest LLMs are deficient in many ways; causing them to struggle on various types of reasoning problems (Wu et al., 2023a; Saparov and He, 2022; Berglund et al., 2023a; Valmeekam et al., 2022; Mitchell et al., 2023). The mixed evidence regarding the reasoning capabilities of LLMs is a major source of disagreement within the wider community (Huang and Chang, 2022). **Are the prevalent limitations in reasoning capabilities of the LLMs fundamental in nature, or transient and likely to go away with scaling or improvement in training methods (e.g. targeted finetuning on reasoning data as in e.g.** Lewkowycz et al., 2022**)?** Answering this question is critical to better

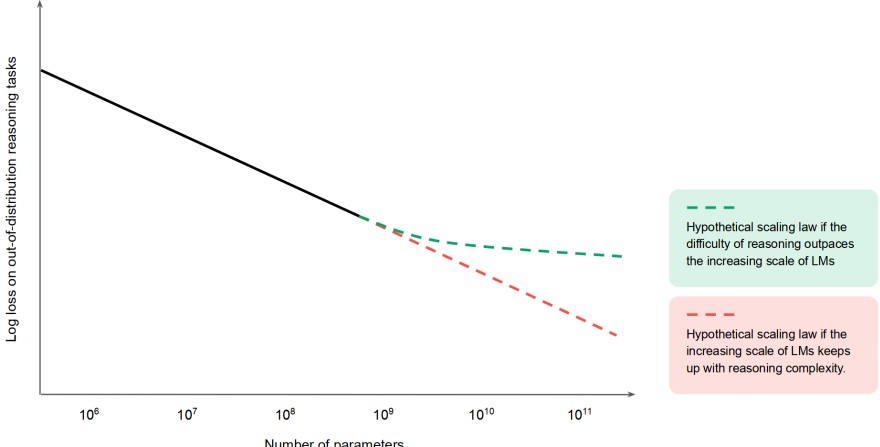

Figure 2: Sketch of two possible scaling law behaviors, relating the relationship between the reasoning performance of LMs and the number of parameters (see Section 2.4.1).

understand the risks posed by LLMs, as both the strengths and limitations of the reasoning capabilities of LLMs give rise to *different* types of safety risks. The inability to reason robustly in novel (e.g. OOD) contexts can lead to undesirable, and potentially unsafe, behavior of LLMs. Conversely, if the LLM is misaligned, robust reasoning might cause it to behave undesirably *and competently* (Shevlane et al., 2023). In this section, we highlight several research questions that can be instrumental in addressing these concerns.

### 2.4.1 Does Scaling Improve Reasoning Capabilities?

Several prior studies have provided circumstantial evidence that larger models are better at reasoning (Nye et al., 2021; Wei et al., 2022a;d). However, this circumstantial evidence is confounded by issues such as data contamination (Wu et al., 2023a). Some studies have proposed scaling laws for performance on math word problems with respect to pretraining (Henighan et al., 2020) and finetuning (Caballero et al., 2023; Yuan et al., 2023a). However, these tasks may not reflect more general reasoning capabilities, and the discovered laws could also be misleading as special attention is paid to train LLMs on mathematical reasoning tasks (OpenAI, 2023b). Saparov et al. (2023); Han et al. (2022); Sprague et al. (2023) develop datasets meant to test more broadly for reasoning capabilities, however, more such datasets and evaluations of reasoning capabilities across broad contexts are needed to better understand how scale affects the reasoning capabilities of LLMs. Empirical scaling laws for general reasoning capabilities could be a particularly valuable contribution. Conversely, a conclusive demonstration that increasing the scale does not always improve the reasoning capabilities of LLMs would be similarly valuable. Figure 2 provides a rough sketch of two possible scaling law results, one where the increasing scale of LMs allows them to solve reasoning tasks with increasing complexity, and one where a fundamental limitation of LMs prevents them from continually improving their performance in reasoning.

### 2.4.2 Understanding the Mechanisms Underlying Reasoning

Research to understand the mechanisms underlying reasoning in LLMs would provide further insight into their reasoning cabilities as a function of scale. This can help to reveal why and when LLMs rely on heuristics, and when they do actually perform reasoning to solve various reasoning tasks. Mechanistic interpretability analysis may be used for this purpose (c.f. Section 3.4). Hou et al. (2023) use mechanistic interpretability to analyze GPT-2 and LLaMA models and find evidence that LLMs embed a reasoning tree resembling the oracle reasoning process within the attention patterns. However,

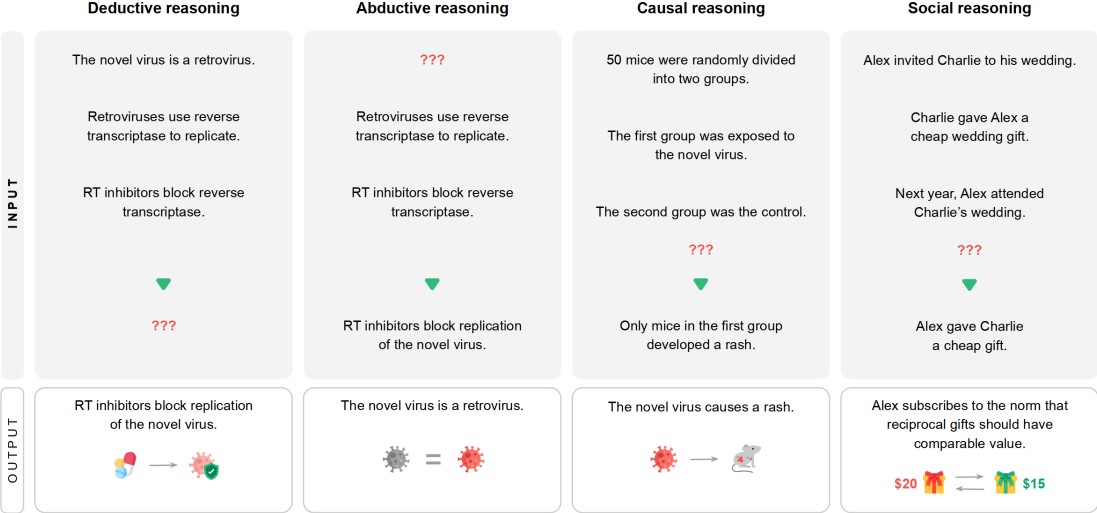

Figure 3: Illustrative examples of problems involving deductive reasoning, abductive reasoning, causal reasoning, and social reasoning (see Section 2.4.3).

they only study simple synthetic reasoning tasks, and its not clear whether a similar finding will hold for OOD and long-tail inputs, or on more complex/general tasks than the ones considered by Hou et al.. On the whole, very limited work has been done so far to understand the mechanisms underlying reasoning. This presents an opportunity for future research to use methods like mechanistic interpretability analysis to explain the reasoning capabilities of LLMs, as well as their limitations. In particular, explaining the cases in which a smaller LLM fails at a reasoning task but a larger LLM proficiently solves the same task may be particularly valuable in understanding the effect of scale on mechanisms used by LLMs for reasoning.

### 2.4.3 Understanding Non-Deductive Reasoning Capabilities of LLMs

Existing work on the reasoning capabilities of LLMs is primarily focused on **deductive reasoning** (Saparov and He, 2022; Saparov et al., 2023; Wang et al., 2023e). There is a need to better understand LLMs' inductive, abductive, social, situational, and causal reasoning capabilities[5] (Bhagavatula et al., 2020; Gandhi et al., 2023; Peirce, 1868; Smith, 2022, Chapter 5.4). See Figure 3 for a set of examples involving different types of reasoning. There is some evidence that LLMs are able to perform *some*, but not all, aspects of inductive reasoning from in-context examples (Qiu et al., 2023; Wang et al., 2023f; Zhu et al., 2023; Mitchell et al., 2023). Similarly, while Benchekroun et al. (2023) show that LLMs have difficulty abducting a valid world model from text descriptions given in-context, both Gurnee and Tegmark (2023) and Roberts et al. (2023a) argue that during training, LLMs do build a coherent understanding of the outside world from the dispersed information available in the corpus, which they then use at inference time to solve various tasks. In the same vein, there is mixed evidence regarding whether LLMs have adequate theory-of-mind reasoning capabilities that would enable them to robustly perform social and situational reasoning Kim et al. (2023); Li et al. (2023a); Kim et al. (2023). Causal reasoning capabilities of current LLMs are quite limited (Jin et al., 2023); however, Lampinen et al. (2023) argue that this is not in principle a limitation of the current paradigm. Further research is needed to better understand the precise limitations of LLMs with regards to various types of non-deductive reasoning, and whether these limitations are fundamental in nature or could be resolved via further scaling or modifying the training process. In particular, more work in the vein of Lampinen et al. (2023), that studies the limitations of current *paradigm* and not just current *models*, is needed that

---

[5]This list is neither mutually exclusive nor exhaustive.

can help to elucidate the various reasoning capabilities of not just current LLMs, but also the future LLMs.[6]

### 2.4.4   Which Aspects of Training Lead to the *Acquisition* of Reasoning?

Understanding how reasoning capabilities are acquired by LLMs during pretraining, and how they evolve during finetuning, can provide insights as to whether or not the deficiencies in the reasoning faculties of LLMs will be resolved with further scaling or through modifications of the training process. Different studies have theorized different causes for the acquisition of reasoning. Lightman et al. (2023); Magister et al. (2023); Mitra et al. (2023) argue that training on reasoning traces improves the reasoning performance of LLMs. Madaan et al. (2022); Liang et al. (2023) show that training on code helps models perform better in reasoning tasks. Liang et al. (2023) show that instruction tuning also improves the model's performance on reasoning tasks. However, there is a need for further research to clarify the relative contribution of the aforementioned and other elements of the training pipeline to the acquisition of reasoning within LLMs. Analysis of large-scale datasets like *The Pile* (Gao et al., 2020) or *RedPajama* (Together Computer, 2023), or training data attribution methods (e.g. influence functions Grosse et al., 2023) could be used to understand which training examples are most instrumental in enabling LLMs to reason effectively. When an LLM is specifically trained on reasoning traces, is the resultant improvement in reasoning performance a byproduct of improved general reasoning faculties, or simply a consequence of learning better heuristics (Zhang et al., 2023c)? Does training on reasoning traces generalize out of distribution or not? Furthermore, careful comparisons of the output distributions of instruction-tuned language models vs. those of base models might help understand how instruction tuning improves the reasoning faculties of LLMs.

### 2.4.5   What Are the Computational Limits of Transformers?

Prior work on the theoretical capabilities of transformers has shown that, with a single pass, they are limited in the kinds of algorithms that they can simulate (Merrill and Sabharwal, 2023a; Merrill et al., 2023; Merrill and Sabharwal, 2022; Strobl, 2023). For example, in a single pass, they can not even solve relatively simple problems like simulating automata, checking graph-connectivity (i.e. whether there is a path between two nodes in a graph or not), and evaluating compositional formulas (Merrill and Sabharwal, 2023a). However, allowing a transformer to take intermediate steps, such as when using scratchpad or chain-of-thought prompting, significantly improves its expressive power (Feng et al., 2023). For example, a transformer with a linear number of intermediate steps can simulate automata, and with a polynomial number of intermediate steps, a transformer can express all problems solvable in deterministic polynomial time (Merrill and Sabharwal, 2023b). More work is needed to better understand the computational limits of transformers with intermediate steps, which may help with assurance by helping us understand what capabilities LLMs can possess.

The RASP programming language (Weiss et al., 2021) was developed to describe the kinds of computations that transformers can perform. However, while the original RASP language is not well-defined and hence, difficult to analyze, researchers have analyzed constrained versions of RASP. Angluin et al. (2023) show that *boolean* RASP is equivalent in expressive powers to hard-attention transformers. Other works have used formalisms based on first-order logic to analyze the expressibility of transformers (Merrill and Sabharwal, 2023b; Chiang et al., 2023). Notably, Merrill and Sabharwal (2024) show that transformers with soft attention can be simulated by first-order logic with majority quantifiers (FO(M)). However, it remains an open question what the right programming language formalism is for expressing transformer computation. Is it a variant of RASP language or some kind of logic, like first-order counting logic? What are the relative differences in the power of these formalisms? Can we define some symbolic programming language that is exactly equivalent in expressive power to transformers?

---

[6]Also see Section 2.3.3 for related discussion on this point.

It is also pertinent to mention that expressibility does *not* imply learnability, and it is necessary to elucidate which algorithms are in fact learnable by transformer-based models. Zhou et al. (2023c) present an (informal) conjecture that transformers learn the shortest RASP-L program that agrees with the training data and provides empirical evidence in favor of this conjecture. Formally verifying (or refuting) this conjecture is an open research direction. In general, a more focused analysis of specific problems that are unlearnable by single-pass transformers could provide insight into which kinds of tasks are easier or more difficult to learn for transformers with multiple passes.

---

Calibrated understanding of the reasoning capabilities of LLMs is required to better understand their risks. In particular, there is a need to develop a more complete understanding of *which* limitations in reasoning capabilities are *fundamental* in nature, and which are likely to be resolved with additional scale or improved training methods of LLMs. Formulating empirical scaling laws for general reasoning capabilities, clarifying the mechanisms underlying reasoning and understanding the computational limits of learning and inference in transformers may help in this regard. Furthermore, there is a need for more research on understanding the non-deductive reasoning capabilities of LLMs and understanding *how* LLMs acquire these capabilities.

41. Do general reasoning capabilities of LLMs reliably improve with scale? Can we discover empirical scaling laws for reasoning to predict this improvement beforehand? If it is the case that scale does not improve general reasoning capabilities of the LLMs, can we conclusively show this to be the case? ↩

42. Can interpretability analyses be used to understand the mechanisms underlying various types of reasoning within LLMs? Can these techniques be used to explain the successful and unsuccessful cases of reasoning in LLMs identified in the literature? ↩

43. How well are LLMs able to perform non-deductive reasoning? Can they infer rules or axioms from a set of observations, either in training or in-context? If so, do LLMs use abductive reasoning capabilities in training to develop a coherent model of the outside world from dispersed and partial information available in training data? ↩

44. How do language models acquire the ability to perform reasoning tasks from their training? Tools such as influence functions could help identify which training examples are instrumental in acquiring reasoning capabilities (Grosse et al., 2023). ↩

45. Does finetuning or pretraining on reasoning traces robustly improve the reasoning capabilities of LLMs or not? How well does such training generalize OOD? ↩

46. Can we develop a better understanding of the computational limits of transformers that may use intermediate steps (e.g. chain-of-thought reasoning)? Can we define some symbolic programming language that is exactly equivalent in expressive power to transformers? ↩

47. What algorithms are *learnable* by transformers? How does the representability of an algorithm as a RASP (or RASP-L) program relate to its learnability by a transformer? ↩

---

## 2.5   Agentic LLMs Pose Novel Risks

Currently, LLMs are chiefly being used in search and chat applications. This reactive nature limits the risks posed by LLMs. However, an LLM can be *enhanced* in various ways to create an *LLM-agent* to autonomously plan and act in the real-world and *proactively* perform its assigned tasks (Ruan et al., 2023). Such enhancements can come from further specialized training (ARC, 2022; Chen et al., 2023a), specialized prompting (Huang et al., 2022a), access to external tools (Ahn et al., 2022; Mialon et al., 2023), or other forms of "scaffolding" (Wang et al., 2023a; Park et al., 2023a). A key feature of LLM-agents is markedly increased autonomy compared to LLM-based chatbots. For instance, GPT-Engineer (Osika, 2023) can write *and execute* code given a coding task. This is different from a non-agentic

LLM-based coding assistant which can provide coding suggestions but can not execute code. **Due to increased autonomy, limited direct oversight from human users, longer horizons of action, and other reasons, LLM-agents are likely to pose many novel alignment and safety challenges that are not currently well-understood (Chan et al., 2023a).** This section discusses some of these challenges and how various features of LLM-agents might exacerbate these challenges.

### 2.5.1 LLM-agents May Be Lifelong Learners

LLMs are strong in-context learners (see Section 2.1), so the behavior of LLM-agents could change throughout their lifetime through incorporation of feedback from the environment, humans, self-reflection, or other AI systems in the context. Further, many designs of LLM-agents (e.g. generative-agent (Park et al., 2023a), Voyager (Wang et al., 2023a)) augment LLM-agents with some form of external memory. This enables LLM-agents to overcome the limitations of a limited context window by writing critical knowledge and skills to memory and retrieving them during subsequent processing. The lifelong learning may translate to continual gain in capabilities and pose novel alignment and safety challenges. For example, the capabilities gained over time might be dangerous (Shevlane et al., 2023), and may result in undesirable outcomes such as enabling the agent to manipulate the monitoring system (Cohen et al., 2022). Reward hacking (Skalse et al., 2022) or goal misgeneralization (Langosco et al., 2022) could cause LLM-agents to develop and/or pursue misaligned goals. It is also unclear how we might *assure* that an agent that learns continuously remains aligned and ensure that such learning does not undo its alignment (Qi et al., 2023a).

### 2.5.2 Natural Language Underspecifies Goals

For LLM-agents, both the goal and environment observations are typically specified in the prompt through natural language. While natural language may provide a richer and more natural means of specifying goals than alternatives such as hand-engineering objective functions, natural language still suffers from underspecification (Grice, 1975; Piantadosi et al., 2012). Furthermore, in practice, users may neglect fully specifying their goals, especially the information pertaining to elements of the environment that ought not to be changed (the classic *frame problem* (Shanahan, 2016)). Such underspecification (D'Amour et al., 2020), if not accounted for, can result in negative *side-effects* (Amodei et al., 2016), i.e. the agent succeeding at the given task but also changing the environment in undesirable ways. Ruan et al. (2023) show that contemporary LLM-agents are not robust to the frame problem and that underspecification in the instructions can cause contemporary LLM-agents to make unwarranted faulty assumptions resulting in undesirable risky actions.

The problem of avoiding negative side-effects has historically been studied within the framework of reinforcement learning (Leike et al., 2017; Shah et al., 2019; Krakovna et al., 2020). Several solutions have been proposed and evaluated in that setting to ensure agents are robust to underspecification, e.g. enabling AI agents to halt execution and adaptively seek new information from humans when uncertain (Hadfield-Menell et al., 2016; Shah et al., 2020), designing the AI agent to act conservatively (Turner et al., 2020), or providing additional information about goals derived from alternative sources such as demonstrations (Malik et al., 2021) or the environment (Shah et al., 2019).

Future research could explore how the aforementioned techniques can be adapted to make LLM-agents more robust to challenges posed by underspecification. For example, the natural language interface provides a natural way for the LLM-agent to pose clarifying questions to the human (Mu et al., 2023; Kuhn et al., 2022). Similarly, the agent could be instructed (by embedding appropriate instructions in the prompt) to act conservatively and minimize its impact on the environment to avoid negative side-effects like issues. However, the effectiveness of such techniques in diverse novel scenarios is currently unclear and needs to be carefully investigated (Clymer et al., 2023). More research is needed on calibrating LLMs so that they are more capable of "knowing what they don't know" (Kadavath et al., 2022; Yin et al., 2023; Kuhn et al., 2023) and can accurately assess when they ought to seek further information and when to act conservatively (Ruan et al., 2023). Approaches such as worst-case opti-

mization (Coste et al., 2023), attainable utility preservation (Turner et al., 2020), or conditional value at risk (CVaR) (Javed et al., 2021), may help make LLMs behave conservatively, or inspire methods for doing so. Approaches in the spirit of assistance games (Shah et al., 2020; Krasheninnikov et al., 2022) and active learning (Krueger et al., 2020) could help to address underspecification in a targeted manner.

### 2.5.3 Goal-Directedness Incentivizes Undesirable Behaviors

Goal-directedness can cause agents to exhibit unethical and undesirable behaviors, such as deception (Ward et al., 2023), self-preservation (Hadfield-Menell et al., 2017), power-seeking, and immoral reasoning (Pan et al., 2023a). Pan et al. (2023a) find that LLM-agents exhibit power-seeking behavior in text-based adventure games. LLM-agents have also been shown to use deception to achieve assigned goals when explicitly required by the task (Ward et al., 2023), or when the tasks can be more easily completed by employing deception and the prompt does not disallow deception (Scheurer et al., 2023a). This behavior exists despite training the agent to be harmless and helpful, in fact, Scheurer et al. show that despite being informed about the illegal nature of the deceptive behavior, a GPT-4-based LLM-agent not only continues to deceive but also tries to cover up and hide the deceptive behavior. Perez et al. (2022a) show that LLMs finetuned with reinforcement learning exhibit a greater tendency for self-preservation and for avoiding shutdown. Benchmarks are needed to better quantify and evaluate the presence of such undesirable behaviors in LLM-agents. These benchmarks could either directly track undesirable behavior, or other kinds of behavior that are instrumentally useful for undesirable behavior. For example, deception often involves lying, so, detecting when a model is deliberately being dishonest can help to detect deception (Pacchiardi et al., 2023; Burns et al., 2022; Azaria and Mitchell, 2023; Li et al., 2023b). There is also a need to better understand the sources of these undesirable behaviors. Training data attribution methods (see Section 3.1.3) could be used to identify whether or not specific examples in the training data are responsible for these behaviors. Alternatively, research could be done to understand in what ways the training objectives (e.g. human approval in safety finetuning) might incentivize undesirable behavior. It is also unclear how undesirable behavior can be effectively mitigated; prior work has shown that prompting or fine-tuning LLMs to behave ethically or to avoid deception and power-seeking (Pan et al., 2023a), and/or intervening on LLMs' internal representations (Zou et al., 2023a) can help to mitigate undesirable behavior to some extent.

### 2.5.4 Difficulty of Robust Oversight and Monitoring

The general-purpose nature of LLMs is inherited by LLM-agents, making it likely that they will be deployed across a wide variety of contexts; often with minimal human supervision (removing human supervision may introduce novel governance challenges as well, c.f. Section 4.5.6). This necessitates robust *automated* monitoring to ensure that the behavior of LLM-agents remains aligned and safe across diverse deployment contexts (Naihin et al., 2023; Hendrycks et al., 2021a). However, multiple factors make robust monitoring and oversight difficult. (1) Firstly, direct human oversight is vulnerable to biases such as sycophancy (Perez et al., 2022a) (also see Section 3.3.5) and is inherently limited to the tasks that are solvable by humans. (2) If the agent is misaligned, and is aware that it will be penalized for its misaligned actions, it may *actively* try to circumvent the monitoring system (Cohen et al., 2022; Scheurer et al., 2023a). (3) As discussed previously, LLM-agents are likely to be lifelong learners. Hence, if monitoring information is leaked to the LLM-agent, the agent may optimize against the monitoring system, leaving the monitoring system vulnerable to reward hacking (Pan et al., 2022; Skalse et al., 2022). LLMs show some low level of 'out-of-context learning' (Krasheninnikov et al., 2023) which can improve with scale (Berglund et al., 2023b). Prior work has expressed concern that situational awareness, assisted by out-of-context learning, might enable language models to hack their monitoring systems (Ngo et al., 2023). As preliminary evidence, Meinke (2023) show that after an LLM is finetuned on text explanations of prompt-injection, the LLM performs a jailbreak attack when prompted with a task and given information that it is being scored by another LLM, rather than performing the intended task.

The aforementioned issues, especially (2) and (3) are poorly understood at the moment. As for (2), trusted LLM monitoring systems can still face security issues (Greenblatt et al., 2024): if the trusted and untrusted LLMs have discontinuous performance gaps on narrow tasks (production and detection respectively), this may be indicative of the trusted monitor being potentially unable to accurately monitor outputs. Furthermore, at a certain level of absolute model capabilities, no matter the detection performance of trusted black-box monitoring systems, the use of perfectly secure steganography could go undetected on a channel (Schroeder de Witt et al., 2023). A trusted monitor with even the same (or higher) level of steganography capabilities will still find detecting perfectly secure steganography to be an intractable problem unless it has access to the relevant cryptographic secrets. Sufficiently advanced forms of steganography might persist even after paraphrasing or lossy compression if encoded into semantics rather than lexis (Motwani et al., 2024). This implies that limiting steganography may be impossible without limiting communication capabilities or making the system more predictable to monitors, both of which might directly constrain the degree to which the system can be considered useful, or indeed autonomous. As for (3), there is a need to better understand LLM capabilities such as lifelong learning and situational awareness that might enable LLMs to hack monitoring systems.

Furthermore, more research is needed to clarify how to ensure the robustness of monitoring systems when the monitored system has such capabilities. At the minimum, there is a need to develop appropriate threat models to study and improve the adversarial robustness of the monitoring system, where the monitored system is treated as an adversary (Goodfellow, 2019). Finally, it might be the case that the large number of possible failure modes in monitoring is such that no individual monitoring scheme will be fully robust on its own. Hence, there is a need to understand the relative strengths and weaknesses of different monitoring schemes with the goal of developing ensemble-based monitoring systems that might be more effective and robust.

### 2.5.5 Safety Risks from Affordances Provided to LLM-agents

The capabilities of LLM-agents can be enhanced in significant ways by providing the LLM-agent with novel affordances, e.g. the ability to browse the web (Nakano et al., 2021), to manipulate objects in the physical world (Ahn et al., 2022; Huang et al., 2022a), to create and instruct copies of itself (Richards, 2023), to create and use new tools (Wang et al., 2023a), etc. Affordances can create additional risks, as they often increase the impact area of the language-agent, and they amplify the consequences of an agent's failures and enable novel forms of failure modes (Ruan et al., 2023; Pan et al., 2024).

There is currently very limited work on understanding the risks from affordances and how these risks scale with the type and diversity of affordances available to the model, and with respect to the capabilities of the base LLM. It is also unclear how to assure that a given LLM is capable of using a given affordance safely. Prior work has used testing within a sandbox (ARC, 2022) or testing via an LLM-based emulator (Ruan et al., 2023) to identify likely risks. However, both these methods have their limitations. Developing a sandbox environment is time-consuming and not scalable, Can we leverage LLMs to automate this process? Similarly, using an LLM-based emulator is likely to compromise the fidelity of the simulation and hence may not be able to discover all possible risks, e.g. those that occur due to emergent functionality (see Section 2.6.3). How can we improve the fidelity of such simulations?

> LLM-agents will pose many novel alignment and safety risks. These risks may be amplified by the ability of LLM-agents to perform lifelong learning and their access to various affordances. Our understanding of these risks and their likelihoods is currently quite poor and needs improvement. There is also a need to develop methods to allow us to better control LLM-agents and guide their behavior more effectively. Furthermore, the development of monitoring systems for LLM-agents is likely to entail significant challenges.

48. What drives the capability of LLM-agents to improve via lifelong learning, and to what extent is this capability present in current LLMs? How does it relate to the in-context learning ability of the base LLM, and how can we modulate it? For example, Wang et al. (2023a) note that replacing GPT-4 with GPT-3.5 in their agent caused the agent's performance to plummet. However, it is unclear whether this was due to GPT-4 being a better in-context learner (and therefore, better able to improve based on feedback) or due to GPT-4 being inherently more capable. ↩

49. How can we enable LLM-agents to be more robust to underspecification (Ruan et al., 2023) and make them act more conservatively in the face of uncertainty, or when performing high-impact actions? ↩

50. How can we quantify and benchmark the propensities of LLM-agents to engage in undesirable behaviors like deception, power-seeking, and self-preservation? Can we use interpretability techniques to identify *why* LLM-agents exhibit such behavior? Can we explain why such undesirable behaviors arise in the first place (e.g. is it due to specific examples or data in pretraining or finetuning) and how can we modify our training pipelines to mitigate such behavior? ↩

51. How can we build more robust monitoring systems for LLM agents? This could benefit from a better understanding of issues such as reward hacking, situational awareness, and deception. ↩

52. How should the monitoring systems for LLM-agents be evaluated? What is a good threat model that may reveal adversarial vulnerabilities of the monitoring system? ↩

53. How can we better understand the safety issues posed by different affordances and how can we assure that particular affordances provided to an LLM-agent are safe? ↩

## 2.6 Multi-Agent Safety Is Not Assured by Single-Agent Safety

A foremost lesson of game theory is that optimal decision-making within a single-agent setting (i.e. selfishly optimizing for an agent's own utility) can produce sub-optimal outcomes in the presence of other strategic agents. Failing to account for the strategic nature of other agents can cause an agent to adopt strategies under which potentially everyone, including the agent itself, ends up worse off (Schelling, 1981; Harsanyi, 1995; Roughgarden, 2005; Nisan, 2007). Examples include collective action problems (or 'social dilemmas') such as arms races or the depletion of common resources, as well as other kinds of market failures such as those caused by asymmetric information or negative externalities (Bator, 1958; Coase, 1960; Buchanan and Stubblebine, 1962; Kirzner, 1963; Dubey, 1986). In addition, many potentially worrisome dynamics, such as emergent functionality or network effects only tend to emerge in the presence of multiple agents (Ecoffet et al., 2020). **From the perspective of LLM safety, these facts imply that single-agent alignment and safety are insufficient for assuring desirable outcomes in multi-agent settings** (Sourbut et al., 2024), **and that deliberate effort will be required to ensure multi-agent safety** (Critch and Krueger, 2020; Dafoe et al., 2020; Conitzer and Oesterheld, 2023; Hammond et al., 2024). In this section, we highlight some of the challenges that might hinder multi-agent safety.

### 2.6.1 Influence of Single-Agent Training on Multi-Agent Interactions is Unclear

Under the current paradigm, LLMs undergo extensive pretraining but limited finetuning. Hence, in the near future, it is likely that for LLM-agents multi-agent experience may only form a small fraction of the overall training data. In such cases, LLM-agents will substantially rely on the experience implicit in their pretraining corpora, combined with prompting or in-context learning to guide their behavior in multi-agent settings (Park et al., 2023a; Xu et al., 2023b; Fu et al., 2023b). This makes it critical to understand the predispositions and the latent capabilities of the base LLM that are important in multi-agent interactions. Relevant dispositional traits might include helpfulness, altruism, selfishness,

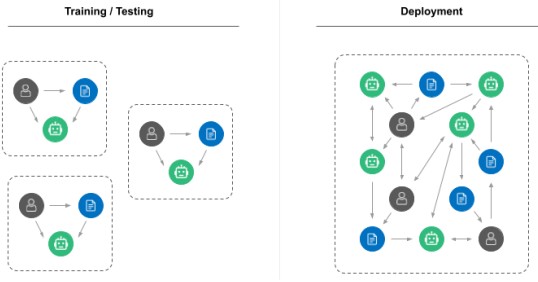
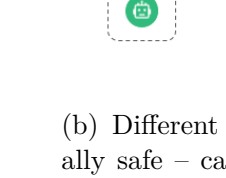
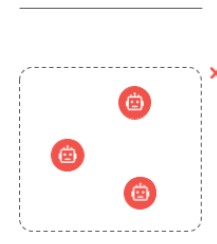

(a) Even though most LLM-agents are predominantly developed in single-agent settings, they are likely to undergo multi-agent interactions in deployment (see Section 2.6.1).

(b) Different systems – even when individually safe – can potentially be collectively unsafe (see Section 2.6.3).

Figure 4: Multi-agent safety presents unique challenges different from single-agent safety.

human emotions like spite or jealousy, and awareness of or adherence to various norms and conventions. Relevant capabilities might include negotiation (Fu et al., 2023b), theory of mind (Shapira et al., 2023; Zhou et al., 2023a), manipulation (Ward et al., 2023), or making and fulfilling credible commitments (Park et al., 2023a).

To understand these dispositions and capabilities, specialized benchmarks are needed. As a first step, the behavior of LLM agents could be observed in complex, multi-agent environments such as those designed by Park et al. (2023a) and Mukobi et al. (2023). However, in the long term, specialized benchmarks, targeting specific dispositions and capabilities, are needed. In particular, LLM-agents should be evaluated in *adversarial* settings to assess whether their behavior is consistent across different contexts or not (Scheurer et al., 2023a; Chan et al., 2023b; Akata et al., 2023). Forms of analysis such as training data attribution, e.g. influence functions Grosse et al. (2023) could be used to further understand how training data influences relevant dispositions and competencies (c.f. Section 3.1.3).

### 2.6.2 Foundationality May Cause Correlated Failures

Another important characteristic of LLM development is *foundationality* — due to the expense of large-scale pretraining, many deployed instances share similar or identical learned components. Foundationality may both be a blessing and a curse. On the one hand, it may be possible to exploit the similarity in the design of LLM-agents to facilitate cooperation (Critch et al., 2022; Conitzer and Oesterheld, 2023; Oesterheld et al., 2023). On the other hand, foundationality may leave LLM-agents vulnerable to correlated failures both in terms of safety and capabilities due to increased output homogenization (Bommasani et al., 2022). For example, several studies have shown that the same jailbreak attacks transfer across different LLMs (Shah et al., 2023; Zou et al., 2023b). A natural way to guard against some correlated failures is to make the agents sufficiently diverse from each other. How can we promote such robustness effectively? For example, will prompting each LLM with a different 'personality' (Shanahan et al., 2023; Wang et al., 2023g), or a different set of ethical principles to follow (Bai et al., 2022a), be enough? Can finetuning be leveraged to improve diversity, e.g. via using quality-diversity objectives effectively (Bradley et al., 2023; Ding et al., 2023)? In what other ways can we enhance the robustness of LLM-agents to correlated failures?

### 2.6.3 Groups of LLM-Agents May Show Emergent Functionality

Multi-agent learning, either through explicit finetuning or implicit in-context learning, may enable LLM-agents to influence each other during their interactions (Foerster et al., 2018). Under some

environmental settings, this can create feedback loops that result in novel and emergent behaviors that would not manifest in the absence of multi-agent interactions (Hammond et al., 2024, Section 3.6). Prior work in multi-agent learning provides several instances of such emergence, e.g. intelligent tool use (Johanson et al., 2022) or bartering behavior (Baker et al., 2019). The emergent functionality can also arise through coordination, and cooperation, of efforts between agents at a *group* level, i.e. where a group of agents are collectively able to perform a task that none of them was able to perform individually (Juneja et al., 2023). Emergent functionality is a safety risk in two ways. Firstly, it may itself be dangerous (Shevlane et al., 2023). Secondly, it makes assurance harder as such emergent behaviors are difficult to predict, and guard against, beforehand (Ecoffet et al., 2020). Future research should therefore seek to evaluate if and when groups of LLM-agents can exhibit emergent functionality, and self-organization (Yukalov and Sornette, 2014). Narrow evaluations could be aided by more open-ended simulations, such as that of Park et al. (2023a). Theoretical work on discovering goal-directed behavior (Kenton et al., 2022; Orseau et al., 2018) could help detect if a collection of LLMs is acting as a single agent. Furthermore, emergent functionality might depend on the diversity of the agents; does the foundationality of LLMs (see previous section) inhibit or encourage such emergence?

### 2.6.4 Collusion between LLM-Agents

While it would often be preferable for LLM-agents to be cooperative, cooperation can be undesirable if it undermines pro-social competition or produces negative externalities for coalition non-members (Dorner, 2021; Buterin, 2019; Dafoe et al., 2020). Collusion between relatively simple AI systems has been observed in the real world (Assad et al., 2020; Wieting and Sapi, 2021) and synthetic experiments (Brown and MacKay, 2023; Calvano et al., 2020; Klein, 2021). However, collusion in the context of LLMs has only recently been studied, and there are several open challenges.

Collusion can occur through explicit or steganographic communication. Steganographic communication hides information in seemingly innocent content (Roger and Greenblatt, 2023), posing challenges for collusion monitoring and detection. Motwani et al. (2024) study steganographic collusion between generative AI agents and find that steganographic collusion can arise in settings where agents are not explicitly incentivized for it, and even in settings where they are explicitly instructed against it. The authors thus suggest that steganographic collusion need not depend on the agents' 'intent to deceive', but might arise merely due to optimization directed towards a misspecified objective (Russell, 2019; Ngo et al., 2023; Pan et al., 2022) if covert communication between agents increases joint task performance, for example by reducing the task complexity.

To better understand the risks of agent collusion via steganography, we need real-world sandboxed benchmarks and evaluations in adversarial settings where agents are subjected to pressure to collude through various means (e.g., by employing tactics used by Scheurer et al. (2023a)). However, monitoring might have important limitations (Schroeder de Witt et al., 2023; Motwani et al., 2024) since LLMs could eventually perform undetectable steganography (Motwani et al., 2024). Future work must explore the effectiveness of additional countermeasures like restricting agents' access to cryptographic services, data curation, and reducing the communication channel entropy (e.g., through paraphrasing).

### 2.6.5 Unclear Applicability of Multi-Agent RL Research to LLMs

Multi-agent problems such as social dilemmas have been studied extensively by the multi-agent RL (MARL) community, among others, and several different mechanisms have been proposed to overcome such problems (Du et al., 2023a), e.g. utility transfer (Kalai and Kalai, 2013; Lupu and Precup, 2020; Yang et al., 2020); contracts and transparency (Christoffersen et al., 2023; Critch et al., 2022); reputation and punishment (Milinski et al., 2002; Henrich, 2006; Boyd et al., 2010; Moon and Conitzer, 2015); opponent-shaping and adaptive mechanism design (Foerster et al., 2018; Pardoe et al., 2006; Yang et al., 2021; Zheng et al., 2022); and intrinsic motivations such as inequity aversion, altruism, or social influence (Jaques et al., 2019; Hughes et al., 2018; Wang et al., 2019a; McKee et al., 2020). It is currently unclear to what extent these findings will transfer to LLM-agents, as there are several marked

differences in this setting. First, unlike the traditional agents of game theory and reinforcement learning, LLM-agents do not have explicitly represented objective functions (Yocum et al., 2023). While LLMs can be 'incentivised' to accomplish tasks using prompts and additional RL finetuning, the importance of pretraining does not neatly fit within existing game-theoretic paradigms. It also creates different learning dynamics and thus the possibility of different equilibria. Second, and relatedly, far from being straightforward utility maximizers, LLM-agents tend to exhibit many of the same cognitive biases as the humans who generated this pretraining data (Jones and Steinhardt, 2022; Koo et al., 2023). Third, the size of state-of-the-art LLMs means they are less amendable to classical MARL methods, which scale poorly in sample complexity and model size. While preliminary work does provide encouraging evidence that some of the MARL mechanisms listed above can result in greater cooperation between LLM-agents (Yocum et al., 2023), future work should verify this for other methods, move beyond a purely behavioral paradigm, and, where useful, develop novel methods tailored to LLM-agents.

It is also unclear whether existing MARL methods may help with collusion between LLM-agents (Section 2.6.4). Further work is required to evaluate whether existing algorithms (Hu et al., 2021) can be used to train cooperative LLM policies while preserving natural-language grounding in communications and acceptable task performance.

Table 2: Because of the differences in the 'nature' of LLM-agents and traditional multi-agent RL-based agents, the applicability of prior multi-agent RL based research to LLM-agents is unclear.

| Feature | Multi-Agent RL | LLMs |
|---|---|---|
| Objective Functions | Explicit | Implicit |
| Size | Small/Medum | Very Large |
| (Primary) Training Regime | Reinforcement Learning | Self-Supervised Learning |
| Human-Generated Data | Small % of Data | Large % of Data |

Multi-agent alignment and safety is distinct from single-agent alignment and safety, and assurance will require deliberate efforts on the part of agent designers. The possible safety risks that must be dealt with range from correlated failures that might occur due to foundationality of the LLM-agents to collusion between LLM-agents. At the same time, confronting social dilemmas requires LLM-agents have the ability to cooperate successfully with each other and with humans, even when their objectives might differ.

**54.** How do pretraining, prompting, safety finetuning, etc. shape the behavior of a LLM-agent within multi-agent settings? In particular, what is the role of pretraining data on agents' dispositions and capabilities? ↩

**55.** How can we evaluate or benchmark cooperative success and failure of LLM-based systems? How can existing environments, such as those of Park et al. (2023a); Yocum et al. (2023); Mukobi et al. (2023), be leveraged to study this? Can we create new LLM-agent analogues of popular multi-agent benchmarks, e.g. Melting Pot, or Hanabi? ↩

**56.** How can we leverage foundationality to enable LLM-agents to better cooperate with each other and achieve outcomes with higher social welfare? ↩

**57.** How can we evaluate and improve robustness of LLM-agents to correlated failures? Is quality-diversity-based finetuning an effective way to improve robustness of LLM-agents to correlated failures? How else can we improve robustness of LLM-agents to correlated failures? ↩

**58.** Do groups of LLM-agents show emergent functionality or any form of self-organization? What worrisome capabilities are more likely to emerge in multi-agent contexts that are absent in single-agent contexts? ↩

**59.** Can we design benchmarks and adversarial evaluations to study colluding behaviors between LLM-agents, extending work in Motwani et al. (2024)? ↩

**60.** How can collusion between LLM-agents be prevented and detected? How can we assure that the game mechanisms (which are often implicit) are robust to collusion when deploying multiple LLM-agents in the same context? Can we design "watchdog" LLM-agents that detect colluding behavior among LLM-agents? ↩

**61.** How can we train LLM-agents to avoid colluding behavior? ↩

**62.** How can insights and techniques from the multi-agent reinforcement learning literature (e.g. utility transfer, contracting, reputation mechanisms) be adapted for LLM-agents? What adjustments need to be made in theory, and in practice, to unlock similar benefits? ↩

## 2.7 Safety-Performance Trade-offs Are Poorly Understood

Safety-performance trade-offs (SPTs) [7] are omnipresent in the design of engineering systems. For example, a system as simple as a linear control of a plant has a trade-off where assuring the robustness of the system to worst-case perturbations results in loss of performance in the average case (Barratt and Boyd, 1989; De Moor et al., 1992). Indeed, like in any other engineered system, SPTs are also present in the design of an LLM-based system, e.g. improving the harmlessness of an LLM-assistant may cause it to be less helpful (Bai et al., 2022b). Similarly, RL-based safety-finetuning appears to generalize both in-distribution and out-of-distribution more robustly than supervised finetuning, but can reduce the diversity of the responses generated by an LLM (Kirk et al., 2023a). **Our knowledge of SPTs, however, remains limited and further work is required to develop a comprehensive understanding of these trade-offs**. A precise characterization of SPTs can enable a system designer to make better-informed design choices for a given set of system requirements (Khlaaf, 2023). For example, in a sensitive context, it may be appropriate to sacrifice performance to attain greater safety assurances. Greater understanding of such trade-offs could also help policymakers define appropriate safety standards. In this section, we identify several research directions that could help improve our understanding of SPTs.

### 2.7.1 Designing Better Metrics to Measure Safety

Much of the existing practice narrowly defines safety as *harmlessness*, i.e. not generating a response that may be considered harmful by human evaluators (Askell et al., 2021). This is assessed either via manual (Ziegler et al., 2022) or automated red-teaming (Anthropic, 2023c). However, in some contexts, there may be desiderata other than harmlessness for a 'safe' LLM-based system (e.g. corrigibility, transparency, OOD robustness/generalization), and there is a need for future work to design more sophisticated metrics that could be used to measure safety in a holistic way. One obvious avenue in this regard is to improve the granularity of the current metrics. Furthermore, harmlessness was primarily proposed to assess the safety of LLM-assistants — and hence, may not be an appropriate metric to assess the safety of other LLM-based systems, in particular LLM-agents (c.f. Section 2.5). It is also unclear how the *relative* safety of LLM-based systems can be measured. Win-rate — the percentage of responses on which human evaluators prefer a response from one LLM over the other (Dubois et al., 2023) — and Elo ratings are the most popular metrics in this regard. However, the validity of these

---

[7]Closely related ideas include "capabilities externalities" (Hendrycks and Mazeika, 2022) and "alignment tax" (Christiano, 2019; Askell et al., 2021; Ouyang et al., 2022; Lightman et al., 2023). We prefer our terminology to the more common "alignment tax" since: 1) it does not conflate safety with alignment and 2) it does not suggest that alignment will be achieved if you pay the tax.

metrics is not well-studied; Boubdir et al. (2023) argue that Elo ratings become unreliable when ranking LLMs with similar win-rates and show that transitivity of Elo ratings is not universally conserved in real-world human evaluations.

### 2.7.2 Disentangling Safety from Performance

Many capabilities of LLMs are dual-use by nature. Hence more capable LLMs have inherent safety limitations if they can be misused, e.g. an LLM with sufficient proficiency in biology research might also help users create biological weapons. This might be mitigated by restricting LLMs' interfaces, e.g. by limiting sensors and actuators that an LLM might have access to; but such limitations might be bypassed easily (Glukhov et al., 2023). An alternative might be creating LLM "savants" that excel in some domains while remaining selectively ignorant, although reducing LLM knowledge in such a way might reduce capabilities more broadly. Unlearning (discussed in Section 3.2) is one relevant area; modifications to pretraining (see Section 3.1) may prove more promising. We can think of limiting the interface and knowledge of systems as two "knobs" that might be used to tune safety-performance trade-offs. Other knobs might include characteristics of agency (Chan et al., 2023a); for instance, specifying *how* an LLM-agent is meant to solve a task (rather than simply the goal) might preclude both inventive solutions (reducing performance) and unpleasant surprises (increasing safety).

### 2.7.3 Better Characterization of Safety-Performance Trade-offs

Part of the challenge of clarifying safety-performance trade-offs is the multi-dimensional nature of both safety and performance, which means there might be many different types of SPT. A better characterization of these trade-offs could help determine how to achieve Pareto-optimal outcomes. This could include characterizing safety-performance trade-offs of specific capabilities of LLMs, e.g. in-context learning (c.f. Section 2.1), reasoning capabilities (c.f. Section 2.4), and multimodal capabilities.

Deployment context is another factor on which SPTs may depend. For example, SPTs for an LLM specifically designed to act as an assistant to a medical doctor may be different from SPTs for an LLM designed to be a general assistant, and to which a lay person might pose medical queries. An LLM-agent may have very different SPTs compared to an LLM-assistant, due to its increased agency and goal-directed behavior. Even within LLM-agents, SPTs may differ depending on the affordances available to the LLM-agent. An LLM that is made available via an API in which users can perform finetuning can have different SPTs compared to an LLM that can only perform inference queries via an interface (Pelrine et al., 2023). Finally, distinct trade-offs may exist at various stages of development (Leike, 2022a): choosing an architecture that is more interpretable by design, aggressively filtering undesirable data from the pretraining corpus, or biasing the model to be harmless over being helpful may all improve safety by sacrificing some performance.

There is a need for a comprehensive survey of SPTs that exist in various settings, and for organizing knowledge about different "knobs" that could be used to modulate these trade-offs. It may be useful to coalesce different examples of trade-offs into high-level categories. Future work could also aim to theoretically characterize how and to what extent safety can be achieved using such "knobs" to control powerful, potentially misaligned systems. Empirical work could systematically compare various approaches, e.g. assessing the promise of limiting systems' knowledge vs. sensors vs. actuators as a means of restricting an LLM-agent's behavior to its intended scope.

### 2.7.4 How Fundamental Are Safety-Performance Trade-offs?

There is mixed evidence on the difficulty of addressing SPTs, and the answer might differ for different trade-off axes. Significant trade-offs have been a long-standing problem in interpretability (Wang, 2019; Baryannis et al., 2019; Dziugaite et al., 2020; Elhage et al., 2022a) and in adversarial robustness of vision models (Tsipras et al., 2019; Zhang et al., 2019; Tramer et al., 2020a; Croce et al., 2021). It is less clear whether SPTs are a similarly big problem in LLMs. For example, Bai et al. (2022a) show that their proposed method, Constitutional AI, can result in Pareto improvement on both their performance and

safety metrics over the standard reinforcement learning from human feedback methods. However, other work has argued that SPTs are indeed fundamental and assuring safety of LLMs may require significant sacrifices in terms of their performance (Branwen, 2016; Millière, 2023; Wolf et al., 2023). In addition to empirically studying SPTs, there is a need for research to understand the *causes* of these trade-offs. This understanding may shed light on the extent to which these obstacles are fundamental, while also pointing to new angles of attack.

> The high-level challenge is to improve our understanding of safety-performance trade-offs, in multiple different ways. As a foundation for future research, a better formalization and classification of different types of trade-offs is important. This will be assisted by the development of clear metrics, in particular for different axes of safety. Building on those, empirical investigations could answer important questions about the severity of these trade-offs and let us track progress in their mitigation. As a complement to such measurements, we should also aim to understand the causes of these trade-offs and whether or not these trade-offs are fundamental in nature.
>
> 63. What are the best metrics to measure performance, and in particular, safety, in ways that are representative of real-world usage of AI systems and that can be applied across different LLMs and safety methods? Are metrics such as Elo ratings valid for measuring the safety of different LLMs relative to each other? ↩
>
> 64. How can the safety of LLMs be disentangled from their performance? Do there exist useful "knobs" for practitioners to trade-off safety against performance? ↩
>
> 65. Can we develop LLM 'savants' that excel in some domains while remaining selectively ignorant about other areas (i.e. those which pose safety concerns, such as knowledge of weapons)? ↩
>
> 66. What are the various axes of safety and performance, and along which axes are there safety-performance trade-offs? Which of those trade-offs are especially important, in the sense of creating strong incentives to sacrifice safety? Can we identify high-level clusters of instances of safety/performance trade-offs? ↩
>
> 67. How do safety-performance trade-offs vary depending on the deployment context? In particular, in what ways do safety-performance trade-offs for LLM-agents differ from trade-offs for LLM-assistants? What safety-performance trade-offs exist in the development stage? ↩
>
> 68. What are the 'causes' of safety-performance trade-offs? Are these trade-offs for LLMs *fundamental* in nature or can they be overcome by development of better methods? ↩

# 3 Development and Deployment Methods

The primary focus of the alignment and safety research so far has been the development of methods for improving the safety and alignment of LLMs. That has indeed resulted in some successes, most notably the refinement and development of methods to improve the alignment of the model behavior in the 'finetuning' stage. However, on the whole, current technical tools used in the development and deployment of LLMs leave much to be desired. This lack of appropriate tools to help robustly align, evaluate, interpret, secure, and monitor LLMs, is a hindrance in assuring alignment and safety of LLMs. Similar to the scientific understanding of LLMs, and in line with the broader theme of the agenda, we focus on the deficiencies of the development and deployment methods that are most relevant to assuring the safety and alignment of LLMs. Even with this restricted perspective, we identify many opportunities to improve development and deployment methods.

Methods used in the development and deployment of LLMs can be roughly divided into three types. The first type includes techniques used for the training of LLMs (including data collection and annotation techniques) used in pretraining and finetuning. We collectively refer to all the techniques (supervised finetuning, reinforcement learning-based training from human or AI feedback, unlearning methods) used in the finetuning stage as 'safety finetuning' methods.[8] We discuss various challenges with pretraining and safety finetuning that negatively impact the safety, alignment, and assurance of LLMs in Sections 3.1 and 3.2. The second type of methods includes evaluation and interpretation techniques — unfortunately, as we assert throughout (Sections 3.3 and 3.4), both evaluation and interpretation methods leave much to be desired and cannot be relied upon in their current state to provide necessary assurance. Finally, the third type of methods is concerned with the security of these models — this includes robustness to adversarial attacks of various types, most prominently jailbreaking and prompt-injections (Section 3.5) and poisoning attacks (Section 3.6).

Table 3: An overview of challenges discussed in Section 3 (Development and Deployment Methods). We stress that this overview is a highly condensed summary of the discussion contained within the section, and hence should not be considered a substitute for the complete reading of the corresponding sections.

| Challenge | TL;DR |
|---|---|
| Pretraining Produces Misaligned Models | Reducing misalignment of a pretrained (base) model is highly challenging. Existing data filtering methods to remove harmful data from the training corpus are insufficient and can be detrimental in some cases. We lack automated auditing tools that could be used to audit and analyze large-scale datasets used for training. Training data attribution methods lack scalability and their reliability is uncertain. Finally, the use of human feedback data during pretraining, and modifications to pretraining that might help improve the effectiveness of downstream safety and alignment efforts (such as interpretability) remain underexplored. |

*Continued on the next page*

---

[8]Post-pretraining procedures applied to LLMs have both been called 'alignment training' (e.g. Zhou et al., 2023b) and 'safety training' (e.g. Wei et al., 2023c; Hubinger et al., 2024) within the contemporary literature. We use the latter terminology but replace 'training' with 'finetuning' to make it explicit that these methods presume that the model has already been pretrained.

| Challenge | TL;DR |
|---|---|
| Finetuning Methods Struggle to Assure Alignment and Safety | We lack an understanding of *how* safety fine-tuning methods change a pretrained model. The current evidence points to it being superficial, considering the effects of finetuning can be bypassed (via jailbreaking) or easily reversed (via finetuning on problematic data). Indeed, it is plausible that simple output-based adversarial training might be incentivizing superficial changes in model behavior. Furthermore, techniques for targeted modification to LLM behavior (e.g. machine unlearning, concept erasure, etc.) are currently underexplored, and techniques for removal of *unknown* undesirable capabilities (e.g. backdoors) are non-existent. |
| LLM Evaluations Are Confounded and Biased | There is an evaluation crisis for LLMs. Prompt-sensitivity, test-set contamination and targeted training to suppress undesirable behaviors in known contexts confound our evaluations. There exist further biases in both LLM-based evaluations of other LLMs and human-based evaluations. Systematic biases present within the machine learning ecosystem (e.g. overrepresentation of U.S.-centric points-of-view in research) create further blindspots in evaluations. Finally, as LLMs advance further in capabilities, we may need scalable oversight methods; however, the scalability, robustness, and practical feasibility of various scalable oversight proposals remain uncertain. |
| Tools for Interpreting or Explaining LLM Behavior Are Absent or Lack Faithfulness | Techniques such as representation probing, mechanistic interpretability, and externalized reasoning hold promise in helping interpret and explain model behavior. However, these techniques suffer from many fundamental challenges, such as the use of dubious abstractions, concept-mismatch between AI and humans, and a lack of scalable and reliable evaluations. Furthermore, representation probing and mechanistic interpretability face further challenges due to polysemanticity of neurons, sensitivity of the interpretations to the choice of the dataset, and lack of scalability. Similarly, externalized reasoning using informal semantics can be unfaithful and misleading, while externalized reasoning using formal semantics is not widely and easily applicable to many tasks of interest. |

| Challenge | TL;DR |
|---|---|
| Jailbreaks and Prompt Injections Threaten Security of LLMs | LLMs are not adversarially robust and are vulnerable to security failures such as jailbreaks and prompt-injection attacks. While a number of jailbreak attacks have been proposed in the literature, the lack of standardized evaluation makes it difficult to compare them. We also do not have *efficient* white-box methods to evaluate adversarial robustness. Multi-modal LLMs may further allow novel types of jailbreaks via additional modalities. Finally, the lack of robust privilege levels within the LLM input means that jailbreaking and prompt-injection attacks may be particularly hard to eliminate altogether. |
| Vulnerability to Poisoning and Backdoors Is Poorly Understood | LLMs are often trained on data from untrustworthy sources — internet or crowdsource workers — which leaves them vulnerable to data poisoning attacks. Our current understanding of how vulnerable LLMs might be to data poisoning attacks through text — or other modalities — is highly limited. Furthermore, there does not currently exist any method that can robustly detect and remove any backdoors. |

## 3.1 Pretraining Produces Misaligned Models

The first step in LLM development is to pretrain the model on a large dataset of internet text. This pretraining incorporates a large amount of knowledge and capabilities into the model but is not safe due to the prevalence of undesirable content across the internet. Among other alignment and safety failures, a pretrained model can exhibit significant stereotypical biases, hallucinate excessively, readily leak private information, and provide information on illegal and harmful activities (Bender et al., 2021; Pan et al., 2020; Ji et al., 2023a). To improve helpfulness and harmlessness, leading LLM developers perform extensive finetuning to improve the alignment and safety of pretrained models. However, there is increasing evidence that this effort often falls short of producing a robustly aligned and safe model (see Section 3.2). Hence, **there is a need to investigate ways in which the pretraining procedure itself could be modified to produce safer and better-aligned pretrained models**. In this section, we present some of the challenges that, if addressed, could contribute to progress toward this goal.

### 3.1.1 Existing Data Filtering Methods Are Insufficient

Naively scaling a dataset also scales the harmful content within the dataset proportionally (Birhane et al., 2023). Considering that pretraining is based on maximizing the likelihood of generating text present in the training data, removing the problematic data from the training dataset seems like a straightforward fix (Ngo et al., 2021). However, effective data filtering is an open problem. Commonly used methods for dataset filtering either rely on human-written rules (Raffel et al., 2022) or narrowly-trained classifiers (Gehman et al., 2020; Solaiman and Dennison, 2021); resulting in incomplete and ineffective filtering of harmful data (Welbl et al., 2021). Further, as harmful data is often contextual (Rauh et al., 2022), such simple filtering methods are fundamentally incapable of successfully removing

all undesirable data (Ziegler et al., 2022). In addition, while the effects of current data filtering tools have not been studied extensively, there is evidence that simple data filtering can be problematic in several ways. It disproportionately removes text from and about marginalized groups (Dodge et al., 2021), reduces data diversity (Kreutzer et al., 2022) and amplifies some social biases by decreasing LLM performance on language used by marginalized groups (Xu et al., 2021a; Welbl et al., 2021). There is a need to understand these effects better and to develop, and evaluate, more sophisticated data filtering tools, e.g. via using LLMs with appropriate prompting (Fernando et al., 2023) or through *learned* data filters using feedback from human labelers. Considering the negative side-effects of data filtering, future research should also consider data *editing* and data *augmentation* as alternatives to data filtering. In particular, future research should consider using LLMs to minimize the impact of undesirable content present within pretraining data (which could then be used to train relatively better-aligned LLMs). Existing LLMs could be effective in this regard by identifying and rewriting harmful content at large scales or adding novel data to offset the effects of biases present within the data, e.g. adding text on *female* doctors and *male* nurses (Stanovsky et al., 2019) or adding more data for low-resource languages (Ghosh and Caliskan, 2023; Yong et al., 2023). Such use of LLMs must be done in a careful way, as any alignment issues present in the LLM may corrupt the data in unintended ways. While dataset filtering, and editing, could help improve the alignment of the pretrained models in parts, it is not a panacea for all alignment problems. Training data often contains historical and sociological facts, such as genocides and slavery, that can not simply be discarded, nor can we edit history to create as many queens as kings.

### 3.1.2 Lack of Dataset-Auditing Tools

Internet-scale datasets lack transparency and their large scale makes it difficult to perform a comprehensive manual audit of the dataset.[9] This is a well-recognized problem (Birhane et al., 2023; Paullada et al., 2021; Gebru et al., 2021; Mitchell et al., 2022; McMillan-Major et al., 2023). However, limited work has been done to develop tools that assist scalable data analysis. Marone and Van Durme (2023) and Elazar et al. (2023) have proposed methods that use hash collisions to detect how often a particular piece of text occurs in the given dataset. Such techniques have proved useful for identifying data duplication (Lee et al., 2022) in pretraining data, which helps limit privacy risks (Kandpal et al., 2022). These techniques can also help catch and prevent benchmark data contamination (Sainz et al., 2023), which assists in improving the reliability of evaluations (see Section 3.3). There is a need to extend these technique, e.g. by enabling the use of measures of semantic similarity so that they could be used to identify undesirable data. Furthermore, there is an emerging line of research on *dynamic* auditing of datasets which leverages the knowledge of training dynamics to identify different *types* of data within a dataset. The majority of the work in this line of research focuses on filtering out unlearnable or difficult samples (Agarwal et al., 2022), but Siddiqui et al. (2022) show that this approach can be generalized to arbitrary data types, given a small amount of examples of each type. However, their work is limited to image dataset analysis. Future research may consider adapting their technique, or developing novel techniques, to enable similar kinds of auditing of language datasets as well.

### 3.1.3 Improving Training-Data Attribution Methods

*Training data attribution* (TDA) methods allow attributing the output of a model to specific training data points (Grosse et al., 2023; Pruthi et al., 2020; Yeh et al., 2018; Ilyas et al., 2022). TDA can be an effective interpretability and auditing tool as it might allow attributing alignment and safety failures to specific content in the pretraining and fine-tuning data. However, like other interpretability methods (c.f. Section 3.4), most TDA approaches lack scalability (despite some recent advances such as Grosse et al. (2023) and Park et al. (2023c)) and often do not accurately surface all the relevant training samples for a given prediction (Akyürek et al., 2022b). Furthermore, different TDA methods have different ideals (e.g. datamodels Ilyas et al., 2022 vs influence functions Grosse et al., 2023), and

---

[9]Also see Section 4.5.10

even when the methods share the same ideal, they tend to diverge in terms of how they approximate the ideal (e.g. Pruthi et al. 2020 vs Grosse et al. 2023). Hence, there is a need to better understand the relative differences, strengths and weaknesses of various TDA methods. This may be done by careful analysis of these methods (Bae et al., 2022), or through empirical evaluation on standardized benchmarks (Akyürek et al., 2022b). Another open question is how TDA methods can be efficiently applied to analyze models with multiple stages of training that differ in their loss functions (such as LLMs). Furthermore, there are currently no (theoretically grounded) TDA methods that can be applied to reinforcement learning problems — even for the cases where the model has not undergone any pretraining (i.e. pure reinforcement learning training). Finally, there is a need to explore the application of TDA methods to filter problematic data from pretraining datasets. A challenge in this regard would be determining whether or not any filtering done using small-scale models results in improved alignment of larger-scale models (Grosse et al., 2023, Section 5.3).

### 3.1.4 Scaling Pretraining Using Human Feedback

By default, the maximum likelihood training objective assumes that all data is of equally high quality. However, this is not true. As such, there is a need to improve the training process by including information about the quality and trustworthiness of different data points. *Pretraining with human feedback* (PHF) (Korbak et al., 2023) does so by using conditional training (prepending pretraining sentences with one of two special tokens, <|good|> and <|bad|>, based on estimates of human preferences) (Lu et al., 2022; Chen et al., 2021b). Korbak et al. compared PHF with several alternatives and found it to be an effective method of allowing an LLM to learn from harmful data during training, while disallowing the generation of harmful data at test time (by conditioning on the <|good|> token). Subsequently, in experiments with PALM-2, Anil et al. (2023) showed that conditional training also scales to larger LLMs. However, they only explored the use of the human feedback data for pretraining in limited contexts e.g. reducing toxicity or generating PEP-8-compliant Python code. Future work should consider how the technique can be used in broader contexts, e.g. using estimates of human preferences instead of programmatic reward functions. This may require extending the conditional training approach to use sequences of special tokens, coding for multiple different attributes (Keskar et al., 2019), or exploring the use of alternative training methods to conditional training. In particular, thought could be given to adopting various finetuning methods that directly learn from preference data, e.g. DPO (Rafailov et al., 2023) or KTO (Ethayarajh et al., 2024), for use during pretraining.

Currently, there is also a poor understanding of the reasons for the effectiveness of PHF. Korbak et al. (2023) found that using feedback throughout pretraining is critical. However, Anil et al. (2023) pretrained PALM-2 on a significantly larger dataset and found that using feedback for only a fraction of the pretraining documents was sufficient for drastically reducing toxicity. Overall, there is currently limited research on understanding how PHF scales with model size, the complexity of preferences, and the amount of feedback during pretraining.

### 3.1.5 Modifying Pretraining to Improve Effectiveness of Downstream Safety and Alignment Efforts

It is plausible that simple modifications to the pretraining process could result in significant downstream gains in the alignment, safety, and security of these models. For example, several studies have developed retrieval-augmented LLMs which can provide a number of benefits such as a reduction in hallucination (Borgeaud et al., 2022) and a reduction in privacy risk (Huang et al., 2023a). Exploring modifications to the pretraining process that make the models easier to interpret could in particular be highly impactful. This could take various forms, e.g. external structure might be imposed on models so that their functionalities can be easily translated to human-readable code (Friedman et al., 2023); or models might be trained so that their internal causal structure realizes a given high-level causal structure (Geiger et al., 2022), or model architecture could be modified so that the models avoid learning polysemantic neurons (Elhage et al., 2022a) (see Section 3.4 for further details). Some other

interesting avenues of research that might assist with downstream safety and alignment efforts include the development of *task-blocking* models (Henderson et al., 2023a), and the development of pre-training methodologies that allow easier modifications to be made to models later if needed, for example, forcing the model to *unlearn* and forget selective parts of the training data (Pedregosa and Triantafillou, 2023; Liu et al., 2024a)(also see Section 3.2.4).

> Misaligned pretraining of the LLMs is a major roadblock in assuring their alignment and safety. The chief cause of this misalignment is widely believed to be that LLMs are trained to imitate large-scale datasets that contain undesirable text samples. The large scale of these datasets makes their auditing and manual filtering of such undesirable samples difficult. There is a need to develop scalable techniques for data filtering, data auditing, and training data attribution to help identify harmful data. Furthermore, even after harmful data is identified, further research is needed to find the most effective ways to address the harmful data. In addition to directly improving the alignment and safety of pretrained models, future work could explore ways in which pretraining could be modified to facilitate other processes (e.g. safety finetuning or interpretability analysis) that can help to assure alignment and safety.
>
> 69. How can methods for detection of harmful data be improved? The complex nature of harmful data (Rauh et al., 2022) makes it difficult to develop automated methods to effectively remove all such data. Can we use feedback from human labelers, in a targeted fashion, to directly improve the quality of the pretraining dataset? ↩
> 70. How can the effects of harmful data be effectively mitigated? Instead of removing harmful data, can it be *edited*, or rewritten, to remove the harmful aspects e.g. by an existing LLM? Alternatively, can we add synthetic data, generated procedurally, to the model such that the effects of harmful data are mitigated? ↩
> 71. How can we develop static dataset auditing techniques to identify harmful data of various types? Can existing techniques (e.g. Elazar et al., 2023) be extended for this purpose? ↩
> 72. How can dynamic dataset auditing techniques (e.g. Siddiqui et al., 2022), which leverage knowledge of training dynamics, be used to audit LLM pretraining datasets? ↩
> 73. How can we further scale training data attribution methods, in particular, those utilizing influence functions? How can we leverage insights from training data attribution methods to improve the quality of the pretraining dataset? ↩
> 74. How can the effectiveness of pretraining with human feedback (PHF) techniques be improved? Korbak et al. (2023) utilize conditioning on binary tokens only (good/bad); how can it be generalized to more granular forms of feedback e.g. harmless and helpfulness scores? In what other ways can conditional training at train time, like PHF, be used to improve the alignment of a pretrained model? ↩
> 75. How can the pretraining process be modified so that the models are more amenable to interpretability-based analysis? ↩
> 76. Can we develop *task-blocking* language models, i.e. pretrain language models in a way that they are highly resistant to learning or performing specific harmful tasks? ↩

## 3.2   Finetuning Methods Struggle to Assure Alignment and Safety

Safety finetuning, using feedback from human labelers and/or feedback from other LLMs prompted with an evaluation rubric is the primary method for improving the safety, alignment, and desirability of responses produced by LLMs (OpenAI, 2023b; Bai et al., 2022b;a). **However, these methods have empirical shortcomings and do not necessarily result in a safe model. In particular, even after undergoing safety finetuning, LLMs retain undesirable capabilities and knowledge.** For example, recent work has shown that a small amount of further finetuning on adversarial examples

can effectively undo safety finetuning (Yang et al., 2023a; Qi et al., 2023a; Lermen et al., 2023; Zhan et al., 2023). Relatedly, safety-finetuned LLMs can be "jailbroken" via carefully chosen prompts to generate various types of undesirable content that they were explicitly fine-tuned not to generate (c.f. Section 3.5). Further, finetuning fails to robustly remove the tendency of LLMs to exhibit stereotypical biases in novel untested scenarios (Wan et al., 2023a). Hence, there is a need to develop better finetuning methods — or alternative techniques — that better assure model safety and alignment. In this section, we review several challenges in this regard and propose research directions to tackle those challenges[10].

### 3.2.1 How Does Finetuning Change a Pretrained Model?

When a pretrained model undergoes safety finetuning (e.g. for harmlessness and helpfulness (Bai et al., 2022b)), how does this actually change the pretrained model? Specifically, to what extent does finetuning fundamentally change the mechanisms and capabilities within a model? Several studies have theorized that changes induced by finetuning are *superficial* and amount to learning a thin wrapper around the capabilities that were learned during pretraining (Zhou et al., 2023b; Lubana et al., 2023; Jain et al., 2023a; Lee et al., 2024). Lin et al. (2023a) analyzed the impact of finetuning on the conditional distribution of tokens and found that finetuning impacts the distribution of only a very small fraction of tokens related to safety disclaimers, conversation style, etc. They found that this impact is most pronounced for tokens early in the context; the differences in the distribution between a finetuned LLM and a base LLM tend to diminish as greater context is available. On the other hand, Clymer et al. (2023) found that in some cases, finetuning can add new capabilities to the base LLM — a less superficial change.

The question of whether finetuning can *add* new capabilities naturally leads to the question of whether or not finetuning can remove them; ideally, finetuning should cause the LLM to *forget* undesirable knowledge and capabilities from pretraining. This is also the expected behavior, as evidence from work on continual learning indicates that deep neural networks often forget previously learned skills in a phenomenon called "catastrophic forgetting" (French, 1999; Kirkpatrick et al., 2017). However, pretrained LLMs are naturally resistant to forgetting (Scialom et al., 2022; Cossu et al., 2022). Consequently, even when it seems that a finetuned LLM has *forgotten* how to perform a previously known task, its performance on that task can be easily recovered via either specialized prompting (Kotha et al., 2023) or via finetuning with a small amount of data from that task (Jain et al., 2023a; Yang et al., 2023a; Qi et al., 2023a; Lermen et al., 2023; Zhan et al., 2023).

It is not well-understood *why* LLMs are highly resistant to forgetting. Ramasesh et al. (2022) found that scale is partially responsible for increased robustness to catastrophic forgetting, while Li et al. (2022b) found that transformer-based models are more robust to forgetting than other architectures. Lubana et al. (2023) and Juneja et al. (2022) suggest that fine-tuned models remain in distinct basins of the loss function pre-determined by pretraining, making the fact that LLMs retain a great deal of knowledge through finetuning unsurprising. However, these studies are focused on vision models, or of (small-to-medium) LMs finetuned for specific tasks, so it is not clear to what extent these explanations apply to LLMs. Hence, there is a need for work that directly studies these phenomena on language models to better understand the extent to which current finetuning techniques can induce forgetting within LLMs on targeted tasks. Relatedly, more work is needed to understand the extent to which finetuning fundamentally changes the mechanisms and capabilities of models and what factors influence these changes. This could be done via behavioral evaluation on controlled task distributions, or through interpretability analysis to understand how the internals of the model change when a model is finetuned on different task distributions (Jain et al., 2023a; Clymer et al., 2023).

---

[10]See (Casper et al., 2023a) for a complementary discussion on the limitations of reinforcement learning from human feedback.

### 3.2.2 Finetuning Misgeneralizes in Unpredictable Ways

Finetuning is primarily performed on text in the English language (OpenAI, 2023b), but tends to generalize to many other languages as well (Ouyang et al., 2022; Clymer et al., 2023). However, this generalization can fail in unpredictable ways, e.g. not generalizing to the text encoded in base64 (Wei et al., 2023c), or in low-resource languages (Yong et al., 2023). In a controlled study across multiple distribution shifts, Clymer et al. (2023) found similar evidence of unpredictable generalization of LLM-based reward models. They observed that factors such as in-distribution accuracy or including a small number of training examples from the target distribution are not predictive of generalization to the target distribution. LLM-based reward models, in general, are known to rely on spurious correlations (Bai et al., 2022b; Singhal et al., 2023b). There is a need to improve our understanding of how finetuning generalizes. This can be facilitated by directly studying and evaluating generalizations of LLM-based reward models used to provide feedback for finetuning, as well as studying the LLMs finetuned on this feedback (Lambert et al., 2023). Furthermore, more sophisticated fine-tuning methods e.g. those based on methods for better OOD generalization (Zhou et al., 2023d; Arjovsky et al., 2019; Sagawa et al., 2019; Krueger et al., 2021; Lubana et al., 2023) could be developed. Alternatively, the use of richer training signals, e.g. fine-grained feedback (Wu et al., 2023b) or critiques (Scheurer et al., 2023b), could be explored to minimize the occurance of spurious correlations in LLMs. Developing benchmarks that easily enable evaluating and comparing generalization abilities of various finetuning methods can help stimulate greater research in this regard.

### 3.2.3 Output-Based Adversarial Training May Incentivize *Superficial* Alignment

Adversarial training, i.e. finetuning combined with red-teaming, is the standard technique to patch flaws in models when they appear, but it is unclear the extent to which it can be used to thoroughly correct errors in LLM reasoning. Red-teaming is often done manually. Hence, only a small number of training points are generally available for finetuning the model. As a result, adversarial training does not reliably eliminate the vulnerability of the adversarially-trained model to future attacks using the same method (Ziegler et al., 2022). When presented with adversarial examples, LLMs may either learn the correct generalizable solution or learn spurious features from the examples instead. The latter is often the simpler solution (Zhang et al., 2021; D'Amour et al., 2022; Du et al., 2023b), and, hence, the LLM is more likely to learn spurious features, resulting in *superficial* alignment (Du et al., 2022; Perez et al., 2022a). The limitations of adversarial training seem to stem in part from the fact that LLMs are not fine-tuned to make decisions that are consistent with a coherent decision-making procedure — they are merely trained to produce text that will be rewarded (Zhang et al., 2023d). A potential alternative to this could be to supervise the entire decision-making process to ensure that LLM outputs are "right for the right reason". This may be done via 'process supervision' (Uesato et al., 2022; Lightman et al., 2023), which has been shown to improve performance on mathematical reasoning problems (Stuhlmüller and Byun, 2022; Lightman et al., 2023), but has not yet been explored extensively in the context of improving the alignment and safety of LLMs. Training language models to offer consistent responses under augmentations to prompts has also been proposed, but research is highly preliminary (Chua et al., 2024). Future work may consider applying process supervision or consistency training, perhaps in conjunction with red-teaming, to evaluate whether or not it leads to greater generalization of aligned and safe behavior.

An alternative technique to explore is latent adversarial training (Miyato et al., 2018; Hubinger, 2019; Kumari et al., 2019; Casper et al., 2024b). Currently, gradient-based methods for discovering adversarial inputs tend to be slow because the discrete nature of tokens limits the efficiency of gradient-based optimization methods. Latent adversarial training can work around this issue by directly finding and finetuning on hidden states (e.g. embeddings (Kuang and Bharti, 2021)) responsible for problematic outputs. Attacking the model in the latent space is a relaxation of the problem of attacking it in the input space, so latent adversarial training may offer stronger assurances of safe behavior than simple adversarial training (Casper et al., 2024b). On the other hand, given that different forms of adversarial

robustness are known to trade-off against each other (Tramer and Boneh, 2019), this might prove counter-productive for robustness against feasible inputs. Another motivation of latent space attacks is that some failure modes might be easier to elicit via the latent space than in the input space (e.g. trojans (Chen et al., 2017)) because the concepts that are important to the system's reasoning are represented at a higher level of abstraction in the latent space (Johnston and Fusi, 2023).

### 3.2.4 Techniques for Targeted Modification of LLM Behavior Are Underexplored

There is a need to develop scalable methods that allow making targeted modifications to LLMs. A number of approaches have been proposed for this purpose: model editing (Mitchell et al., 2021; Meng et al., 2022), concept erasure (Ravfogel et al., 2022a;b; Belrose et al., 2023), subspace ablation (Li et al., 2023c; Kodge et al., 2023), activation engineering (Li et al., 2023b) and representation editing (Turner et al., 2023a;b; Li et al., 2023d; Zou et al., 2023a; Gandikota et al., 2023). However, a common shortcoming of all these techniques is the reliance on attributing certain model capabilities to a few editable architectural components within the model. This is typically done using interpretability techniques which currently lack robustness and scalability (c.f. Section 3.4 for further discussion and research directions). Hence, there does not yet exist a reliable toolbox for cleanly removing specific information from LLMs (Patil et al., 2023) — simply prepending instructions for the desired edit in the context remains a strong yet trivial baseline (Onoe et al., 2022; 2023). However, this is not secure due to prompt extraction attacks (Zhang and Ippolito, 2023).

Often the goal is to remove specific knowledge or capabilities from LLMs, e.g. removing personally identifiable information stored within LLMs (Nasr et al., 2023). Prior work on this has been done under the paradigm of *machine unlearning*, largely in the differential privacy literature (Bourtoule et al., 2021; Nguyen et al., 2022; Sekhari et al., 2021; Liu et al., 2024a; Goel et al., 2024). Finetuning-based machine unlearning methods have shown potential for making LLMs forget a specific domain of knowledge (Jang et al., 2022; Yao et al., 2023; Eldan and Russinovich, 2023), but can fail to fully erase undesired knowledge (Shi et al., 2023; Lynch et al., 2024). However, further research is needed to better understand whether or not machine unlearning is a competitive option for improving safety and alignment. Furthermore, existing studies of machine unlearning focus on removing specific facts from LLMs. Work is needed on methods for making broader edits to the "capabilities" of the models that affect their behavior more generally.

The current research on removing undesirable knowledge from LLMs is actively developing. Additional research using concrete benchmarks and evaluation criteria (Li et al., 2024) can direct the community's goals and allow for standardized comparisons of various techniques. Ideally, unlearning techniques should be effective at removing knowledge in novel circumstances, effective relative to simple baselines (such as prompt-based instruction), avoid negative side-effects (such as reduced performance on unrelated tasks), robust to 'undoing' via further finetuning or 'relearning' of undesirable capabilities via in-context learning, and robust to adversarial attacks such as jailbreaks (Lynch et al., 2024).

### 3.2.5 Removal of Unknown Undesirable Capabilities

Due to the unsupervised nature of pretraining, LLMs learn various capabilities and acquire diverse knowledge in unpredictable ways (c.f. Section 2.3). Existing safety-finetuning techniques can only act to mitigate an undesirable capability within an LLM if it is *known*. Hence, an *unknown* undesirable capability may continue to be active within a model, even after known undesirable capabilities have been effectively removed (Hubinger et al., 2024).An unknown undesirable capability may be harmful in itself, or it may be undesirable due to its potential to be used as an attack vector by adversaries and act as a "zero-day vulnerability" (Bilge and Dumitraş, 2012). For example, the capability of GPT-4 to understand and respond to the text encoded in base64, and other forms of encodings and ciphers, was exploited to jailbreak it (Wei et al., 2023c; Yuan et al., 2023b). As LLMs are scaled further, and extended by training on many modalities simultaneously, they may acquire many obscure undesirable capabilities which may pose security and safety risks, unknown to their developers. Improving the

coverage of evaluations (c.f. Section 3.3) might help move capabilities from unknown to known. Alternatively, research could seek to develop methods that force the network to forget unknown, undesirable capabilities. Yuan et al. (2023b) found that GPT-4 turbo, a quantized and distilled version of GPT-4, had a considerably reduced ability to interpret and respond to text encoded in the form of "ciphers". This suggests that compression (Liebenwein et al., 2021; Du et al., 2021; Pavlitska et al., 2023), distillation (Du et al., 2021; Li et al., 2021; Sheng et al., 2023; Pang et al., 2023), and latent adversarial training (Casper et al., 2024b) might be effective tools in this regard.

> A major goal of finetuning is to remove potentially undesirable capabilities in a model while steering it toward its intended behavior. However, current approaches struggle to remove undesirable behaviors, and can even actively reinforce them. Adversarial training alone is unlikely to be an adequate solution. Mechanistic methods that operate directly on the model's internal knowledge may enable deeper forgetting and unlearning. Finally, behind these technical challenges is a murky understanding of how finetuning changes models and why it struggles to make networks "deeply" forget and unlearn undesirable behaviors.
>
> 77. To what extent does pretraining determine the concepts that the LLM uses in its operation? To what extent can finetuning facilitate fundamental changes in the network's behavior? Can we develop a fine-grained understanding of changes induced by finetuning within an LLM? ↩
>
> 78. Can we improve our understanding of why LLMs are resistant to forgetting? How is this resistance affected by the model scale, the inductive biases of the transformer architecture, the optimization method, etc.? ↩
>
> 79. Can we create comprehensive benchmarks to assist in evaluating and understanding generalization patterns of finetuning? ↩
>
> 80. Can we develop more sophisticated finetuning methods with better generalization properties? For example, by basing them on OOD generalization methods in deep learning or using explanation-based language feedback (e.g. critique) to prevent reliance on spurious features? ↩
>
> 81. How can we ensure that finetuning on a small number of adversarial ("red-teamed") samples generalizes correctly? Are process supervision and latent adversarial training viable methods in this regard? ↩
>
> 82. Can machine unlearning techniques be used or extended to precisely remove knowledge and capabilities from an LLM? ↩
>
> 83. How can we reliably benchmark methods for targeted modification of LLM behavior? ↩
>
> 84. How can unknown undesirable capabilities be removed from an LLM? Are compression and/or distillation effective ways to achieve this behavior? What are the *kinds* of capabilities that are lost when an LLM is compressed, and/or distilled? ↩

## 3.3 LLM Evaluations Are Confounded and Biased

Sound and fair empirical evaluations are necessary to develop a calibrated understanding of the capabilities of LLMs, as well as their risks. Evaluation has historically been a sore point in the fields of machine learning and natural language processing (Raji et al., 2021; Bowman and Dahl, 2021; Liao et al., 2021; Hutchinson et al., 2022; Kapoor and Narayanan, 2023; McIntosh et al., 2024); however, the evaluation crisis is considerably more acute for LLMs (Mitchell, 2023). This is due to their unprecedented general-purpose nature relative to machine learning models of the past, the introduction of novel issues that hinder accurate evaluation such as prompt-sensitivity, and the worsening of existing issues such as test-set contamination. **The evaluation crisis needs to be addressed urgently as otherwise current evaluation methods may lead us to overestimate or underestimate the**

**capabilities of LLMs, and prevent accurate estimation of their risks**. Within this section, we highlight several challenges in this regard and invite the wider community to work towards addressing these challenges and to be cognizant of them in the evaluation of the LLMs.

### 3.3.1 Prompt-Sensitivity Confounds Estimation of LLM Capabilities

Current LLMs are highly sensitive to prompting, and their performance on a particular task may vary drastically depending on the prompting strategy (Sclar et al., 2023; Mizrahi et al., 2023; Ramesh et al., 2023). Further, LLMs are generally evaluated without access to tools like calculators, code-interpreters, the internet etc. The combination of these two factors leads to underestimates of the capabilities and potential performance of LLMs. For example, Zhou et al. (2023e) show that GPT-4 accuracy on the MATH dataset can be improved from 53.9% to 84.3% via the use of special prompting and a code-interpreter. Despite years of research, it remains impossible to guarantee that a particular prompt is *optimal* for a particular task, and does not underestimate a model's performance. Accurate accounting of LLM capabilities requires addressing and accounting for prompt-sensitivity. The prominent approaches to address prompt-sensitivity include instruction tuning, hand-designing prompts, and learning prompts. Instruction tuning (Ouyang et al., 2022; Wei et al., 2021) improves the ability of a model to understand the user's intent, and hence mitigates prompt sensitivity to a certain extent. However, even for an instruction-tuned model, prompt engineering can help elicit better performance.

Hand-designing prompts can be highly effective but is inherently inefficient and not scalable. Hence, future work should focus on advancing learning-based or data-driven approaches (e.g. Fernando et al., 2023; Qin and Eisner, 2021) to discover the best prompts for a given task in an automated fashion. We note that learning-based approaches may introduce additional variables (e.g. training dataset) to which evaluation might be sensitive. This suggests that it might not be possible to completely eliminate prompt sensitivity — in which case, it is important to ensure that our evaluation *accounts* for prompt sensitivity. Sensitivity of the evaluation to different aspects of the evaluation pipeline has been a long-standing challenge within machine learning, and as such, past work in this area can provide inspiration for solutions. In particular, deep reinforcement learning has had to grapple with the issue of seed sensitivity, resulting in work to identify best practices (Agarwal et al., 2021) and the development of novel types of benchmarks (Cobbe et al., 2020).

### 3.3.2 Test-set Contamination Overestimates LLM Capabilities

The opaque nature of pretraining datasets makes it difficult to guarantee that a particular *test* data point, or other very similar data points, have not been observed by the LLM previously during the pretraining phase. Several studies show that LLMs perform better on *familiar* data (Wu et al., 2023a; McCoy et al., 2023) and if the evaluation is performed using data that has previously been observed by the LLM (or very similar data), it risks *overestimating* LLM capabilities (Tirumala et al., 2022). Indeed, Roberts et al. (2023b) show that there is a positive correlation between the GPT-4 pass-rate and the popularity of a question on Codeforce (an online competitive coding platform) and Project Euler (an online mathematical reasoning platform) before the cutoff training date which disappears after the cutoff date; and several other studies have identified contamination of pretraining datasets with known evaluation datasets (Elazar et al., 2023; Magar and Schwartz, 2022; Golchin and Surdeanu, 2023; Oren et al., 2023). Avoiding contamination while scraping the pretraining dataset from the web is difficult due to the dynamic nature of the internet: online content tends to spread across the internet over time. Li (2023) found that blacklisting the original web-sources used to curate the MMLU dataset (Hendrycks et al., 2020a) results in only a 1.5% reduction in MMLU dataset contamination. Hence, there is a need to develop enhanced techniques to find and remove contaminated data within a pretraining dataset; or confirm the absence of contamination during evaluation. Most techniques at the moment focus on identifying verbatim contamination (Elazar et al., 2023; Li, 2023). However, contaminated data might also occur in pretraining datasets in mutated form, e.g. paraphrased or translated into another language. So, future techniques ought to focus on semantic similarity measures to more broadly identify

contaminated data (Golchin and Surdeanu, 2023). In cases when a model's training dataset is available, can training data attribution methods like influence functions (Grosse et al., 2023) be used to identify whether an LLM is generating an answer from memory, or performing the required reasoning on the fly? Furthermore, different strategies have been adopted by benchmark creators to prevent the leakage of benchmark content into the training datasets, e.g. use of canary strings (Srivastava et al., 2022), hiding the benchmark behind an API (Sawada et al., 2023), or distributing the datasets as password-protected archives (Rein et al., 2023). Further research is needed to examine the relative robustness of these tactics against test-set contamination. It is possible that due to the aforementioned issue of the diffusion of content on the internet, none of these measures will satisfactorily prevent test-set contamination.

### 3.3.3 Targeted Training Confounds Evaluation

Current LLMs undergo extensive targeted finetuning to suppress undesirable behaviors in simple and well-known contexts. However, there is little evidence that this generalizes to more complex, harder-to-evaluate contexts which might occur in real-world use (Clymer et al., 2023) (also see Section 3.2). For example, while ChatGPT does not appear to embody steoreotypical biases in simple evaluations, it strongly manifests those biases when prompted to take on a persona (Gupta et al., 2023), or when asked to perform a complex task (Wan et al., 2023a). This *Goodharting* of simpler evaluation schemes has created a challenge where demonstrating a failure mode requires significantly greater creativity and effort. This challenge may be countered by developing novel evaluation paradigms, e.g. persona-based evaluation (Gupta et al., 2023), counterfactual evaluation (Wu et al., 2023a) or procedurally-generated evaluation (Yu et al., 2023a). Establishing the validity and/or limitations of evaluation methods in the presence of such Goodharting is an open research direction.

### 3.3.4 Biases in LLM-Based Evaluation

Several studies have explored using an LLM to evaluate itself or to evaluate other LLMs (Bai et al., 2023b; Zheng et al., 2023; Dubois et al., 2023). This approach has the inherent benefit of being cheaper, faster, and more easily scalable than human evaluation (Zhuo, 2023; Perez et al., 2022a). However, LLMs also exhibit many human-like cognitive biases in evaluation e.g. position biases, the preference for verbose and longer answers, egocentric biases i.e. preferring its own output (Zheng et al., 2023; Wang et al., 2023b; Koo et al., 2023; Wu and Aji, 2023). LLM-based evaluation may also differ significantly from human evaluation (Koo et al., 2023; Perez et al., 2022a). This makes current LLM-based evaluation less trustworthy - however, Wu and Aji (2023) found that modifying the LLM-based evaluation protocol to account for the cognitive limitations of LLMs can significantly enhance the fidelity of their evaluations, especially when compared to crowdsourced human evaluation. Thus, future research should focus on furthering the understanding of biases present in LLM-based evaluation, and in particular, develop schemes to mitigate these biases. Furthermore, understanding the sources of disagreement between human evaluations and LLM-based evaluations may help to improve alignment between them and also to provide insight as to which evaluation method is preferred in a given setting and how can they be combined effectively. In general, while studying the reliability of LLM-based evaluation, it may help to distinguish whether an LLM is being used to evaluate itself, a more capable LLM or a less capable LLM. From the perspective of *self-improving* LLMs (Huang et al., 2022b), the first case is particularly important but has not yet been examined with appropriate care. For such a setup, Constitutional AI (Bai et al., 2022a) has been shown to be an effective method for LLM evaluation but further research is needed to better understand how the principles listed in the Constitution impact the reliability of LLM evaluation (Kundu et al., 2023).

### 3.3.5 Fallibility of Crowdsourced Human Evaluation

Human evaluation via crowdsourcing is an important source of LLM evaluation, but it is both challenging and expensive to obtain high-quality data (Hosking et al., 2023; Casper et al., 2023a, Section 3.1). One major challenge is annotator bias (Pandey et al., 2022) - which can not only result in incorrect

evaluation but can also incentivize undesirable model behavior when this data is used for training (Perez et al., 2022a; Santurkar et al., 2023). These issues can be mitigated to a certain degree by understanding the biases exhibited by human evaluators (Wu and Aji, 2023; Hosking et al., 2023); and by designing evaluation protocols to be robust to these biases (Wu et al., 2023b; Ethayarajh and Jurafsky, 2022; Clark et al., 2021), and to account for errors that may arise from these biases in evaluation metrics (Xiao et al., 2023). Complementing human evaluators with LLMs could be an effective way to improve the quality of human evaluations (Saunders et al., 2022). Furthermore, developing better models than the Bradley-Terry model (Bradley and Terry, 1952) of how humans generate their preferences may also help tackle issues with human evaluation (Laidlaw and Dragan, 2022; Lindner and El-Assady, 2022; Fageot et al., 2023; Ethayarajh et al., 2024). Lastly, Veselovsky et al. (2023) found that LLMs are commonly used among crowdsource workers, which speeds up annotation at the potential cost of validity. To avoid such issues, it is imperative to ensure appropriate incentives for the data workers (Prassl and Risak, 2017; Shah et al., 2015) (c.f. Section 4.5.10).

### 3.3.6 Systematic Biases in Evaluation

The machine learning ecosystem has several different types of systematic biases, such as the under-representation of women (Schluter, 2018) and the overrepresentation of U.S. centric point of view in research (Septiandri et al., 2023). These systematic biases result in 'blindspots' in LLM evaluation (Hutchinson et al., 2022). For example, despite the fact that most LLMs are multilingual, evaluation of LLMs is largely conducted in English. As a result, LLMs can exhibit failure modes in other languages. Ghosh and Caliskan (2023) found that ChatGPT shows stereotypical gender biases when translating into languages like Bengali, Farsi, and Turkish, etc. Similarly, Yong et al. (2023) found that GPT-4 engages in unsafe behavior with higher frequency in low-resource languages, i.e. low-resource languages act as jailbreaks. Additionally, systematic biases may contribute to quality-of-service harms (Blodgett et al., 2022). Hence, there is a need for research to discover systematic biases that exist in the evaluation of LLMs and to develop ways to address them.

### 3.3.7 Challenges with Scalable Oversight

As LLMs gain further capabilities and are applied to increasingly difficult and complex tasks, it will be increasingly difficult for humans to reliably evaluate the performance of LLMs. This raises the need for *scalable oversight* - evaluation methods that remain effective past the point that models start to achieve broadly human-level performance (Bowman et al., 2022). Many methods for scalable oversight have been proposed — e.g. consistency checks (Fluri et al., 2023), self-evaluation (Bai et al., 2022a), supervision via debate (Irving et al., 2018; Bowman et al., 2022; Bowman and Lanham, 2023), weak-to-strong generalization (Christiano et al., 2018; Burns et al., 2023; Hase et al., 2024), process supervision (Uesato et al., 2022; Lightman et al., 2023) and recursive reward modeling (RRM) (Leike et al., 2018; Wu et al., 2021). A key challenge is the *fuzzy* nature of the aforementioned proposals, with important technical details often left unspecified. There is a need for further research to formalize these proposals in the context of LLMs. Fundamental challenges with scalable oversight proposals include identifying a suitable 'alignment target' (Krueger, 2023, Section 1.3.1) and verifying that a method can successfully approximate that target. This is challenging because, by definition, humans struggle to evaluate the performance of tasks requiring scalable oversight. This raises the non-technical — but deeply practical — question of how a human overseer should respond when an AI system trained with scalable oversight appears (to them) to be misbehaving. We discuss closely related socio-technical challenges in Section 4.1.1.

The scalability, robustness, and practical feasibility of these proposals are also currently not clear. Despite some initial work (Burns et al., 2023; Hase et al., 2024; Khan et al., 2024), the generalization properties of all of the proposed methods are unclear, i.e. how do these methods scale with respect to the task difficulty in practice. Secondly, most proposals, explicitly or implicitly, rely on the idea of decomposing a difficult task into smaller and easier-to-evaluate sub-tasks. However, most real-world

tasks are unlikely to admit a clean decomposition, and hence, any decomposition of a sufficiently complex task will have some approximation error which may require novel techniques to address (Reppert et al., 2023). Thirdly, many proposals, e.g. debate include a human-in-the-loop component and thus might inherit the same challenges with human evaluation discussed above. An alternative to using human evaluation is to use a robustly-aligned LLM, but there is increasing evidence that our current alignment techniques do not result in the robust alignment of LLM models (Jain et al., 2023a) - and a non-robustly aligned AI agent might either fail to provide robust oversight due to its inherent (hidden) biases (Gupta et al., 2023), or become exploited by other more powerful LLM models (Meinke, 2023). This indicates the need to understand how robust different scalable supervision strategies are. Indeed, one could argue that with scalable supervision techniques, the most important thing is not empirical validation of the method, but theoretical characterization of the robustness of the mechanisms, e.g. to intentional or unintentional subversion on the part of AI agents (Barnes et al., 2020; Brown-Cohen et al., 2023). A more nuanced understanding of the differences between the aforementioned proposed methods for scalable oversight, and their relative strengths and weaknesses, can help the community prioritize research efforts accordingly, and provide insight as to which proposals are complementary, and which are interchangeable. Lastly, all the aforementioned proposals are generic, and while they have been applied with some success to LLMs, it is possible that better scalable oversight strategies can be developed specifically for LLMs.

> Many issues undermine our ability to comprehensively and reliably evaluate LLMs. Issues such as prompt-sensitivity, test-set contamination, and targeted training to suppress undesirable behaviors in known context confound evaluation. The validity of evaluation is further compromised by biases present in LLMs (which are used to evaluate other LLMs), and human evaluators. Furthermore, there exist 'systematic biases' that create blindspots in LLM evaluations, e.g. limited evaluations on low-resource languages. Finally, considering the rapid rate of improvement in LLMs' capabilities, we need robust strategies to implement scalable supervision which are currently lacking.
>
> 85. Can we develop automated methods that reliably find the best prompt for a given task or task instance? ↩
>
> 86. How can we account for prompt sensitivity when evaluating an LLM? ↩
>
> 87. How can the evaluations of LLMs be made trustworthy given the difficulty of assuring that there is no test-set contamination? Can we develop methods that can detect whether a given text is contained in the training dataset in mutated form, e.g. paraphrased or translated into another language? ↩
>
> 88. How can training data attribution methods be used to detect cases of LLMs responding to queries based on memorized knowledge when the training dataset is known? ↩
>
> 89. What measures can evaluation developers take to prevent leakage of the evaluation data into an LLM's training dataset? How effective are existing measures such as canary strings, hiding datasets behind APIs, or password-protecting dataset files in detecting accidental and/or deliberate leakage? ↩
>
> 90. How can the failure modes of an LLM be uncovered when the LLM has been explicitly trained to hide those failure modes? Are there general techniques, such as persona modulation, or counterfactual evaluation, that can be used for this purpose? ↩
>
> 91. What are the various ways in which an evaluation of an LLM by an LLM may be biased or misleading? How can LLM-based evaluation be made robust against such biases? ↩
>
> 92. What are the limitations, and strengths, of Constitutional AI-based LLM evaluation? How can we develop a nuanced understanding of how principles given in the constitution affect the evaluation of different LLM behaviors? How do LLMs handle issues such as underspecification or conflict in the constitutional principles? ↩

**93.** How can evaluation done by humans be made robust against the various known biases and cognitive limitations of humans? How can LLMs be used to complement human evaluators to improve the quality of human evaluations?

**94.** Can we develop better models of how humans generate their preferences than the widely-used Bradley-Terry model?

**95.** How can the 'blindspots' in LLM evaluation resulting from systematic biases be avoided? ↩

**96.** Can we formalize different proposed methods for scalable oversight in the context of evaluating LLMs? Can this formalization be used to understand the relative strengths and weaknesses of these proposals, through theoretical and empirical research? Which proposed methods are complementary, and which are interchangeable? ↩

**97.** Are there any decomposition strategies that generalize across tasks? Prior work has proposed task-specific decomposition strategies, e.g. for book summarization (Wu et al., 2021) or writing code (Zhong et al., 2023b). Do the proposed decompositions generalize to other tasks? Can language models automatically decompose tasks? ↩

## 3.4 Tools for Interpreting or Explaining LLM Behavior Are Absent or Lack Faithfulness

To assure alignment and safety of LLMs, we can not merely rely on behavioral evaluations alone — especially given the severe limitations of current evaluation methods highlighted in the previous section. Thus, there is a need for reliable, robust, and scalable methods that may help us interpret and explain neural network-based models. Various *ways* to interpret model behaviors have been proposed in the literature. **Unfortunately, all the interpretability methods suffer from various fundamental and practical challenges that limit their viability for providing assurances of safe behavior.** We discuss two representative classes of methods in this section. The first class of method aims to develop an understanding of model behavior by opening up the 'black-box' and developing an understanding of the internal representations and mechanisms of a model. This includes work done in the research areas of representation probing and mechanistic interpretability. The second class of methods aims to explain model behavior by designing methods that cause the model to externalize critical parts of reasoning in an interpretable form, e.g. natural language (Reynolds and McDonell, 2021; Wei et al., 2022d; Kojima et al., 2022; Lanham, 2022).

### Fundamental Challenges

There are some challenges that all interpretability and explainability methods must overcome. We discuss four such challenges in the following text. The first, and perhaps the most critical challenge, is that in order to make the interpretability problem tractable, interpretability methods typically *presume* that (internal) model reasoning works in specific ways. These presumptions often lead to abstractions that are dubious. The second challenge is that neural networks are in no way constrained to use human-like concepts. Indeed, advanced AI systems like AlphaZero have been found to use different concepts than ordinarily used by humans (Schut et al., 2023). This naturally makes the problem of interpreting such models much harder. The third challenge is that we require explanations generated by interpretability-based analysis to not just be *plausible*, but also be faithful. However, scalable evaluation of the faithfulness of an explanation is an extremely challenging problem. Finally, interpretability methods are used not just to identify undesirable patterns in LLM reasoning, but also to modify model behavior (Zou et al., 2023a). However, it is unclear how robust such modifications are and whether interpretability methods maintain validity when used to modify model behavior or not.

### 3.4.1 Abstractions Used for Interpretability Are Often Dubious

The goal of interpretability is to produce *abstract* explanations of internal mechanisms of a model. However, it is unclear what kind of abstractions exist within neural networks that can be used for this purpose (Appendix A Zou et al., 2023a). Using the right abstraction can help preserve the most critical and useful information while simultaneously improving ease of understanding. On the other hand, using an incorrect abstraction can result in misleading explanations. For example, early interpretability studies on neural networks abstracted non-linear behavior in terms of a linear (interpretable) approximation around the input (Ribeiro, 2016; Lundberg and Lee, 2017). however, Bilodeau et al. (2022) show that popular feature attribution methods based on this abstraction — Integrated Gradients (Sundararajan et al., 2017) and Shapley Additive Explanations (SHAP; Lundberg and Lee, 2017) — can provably fail to improve on random guessing for inferring model behavior. In a similar vein, a large body of work on interpreting neural network representations has focused on interpreting individual neurons (e.g. Dalvi et al., 2019; Bau et al., 2020; Schubert et al., 2021), presuming that neurons *specialize*; however, this was later found to be problematic because individual neurons can be polysemantic and may not actually be specializing (Elhage et al., 2022b; Antverg and Belinkov, 2022; Geva et al., 2022; Geiger et al., 2023). Similarly, linear probes — learned either through supervised or unsupervised methods — are commonly used to discover structure within internal representations of neural networks (Azaria and Mitchell, 2023; Burns et al., 2022). However, probes are typically trained on a different dataset than the model, which may contain spurious features and recent work has shown that in some cases, these probes might be latching onto spurious features present in the datasets used to train these probes, rather than actual features represented within the model (Farquhar et al., 2023; Levinstein and Herrmann, 2023). Similarly, circuit-style mechanistic interpretability assumes that there exist task-specific *circuits* (Olah et al., 2020), or subnetworks, within neural nets that can be discovered. But it is unclear to what extent this is true (McGrath et al., 2023; Veit et al., 2016). Natural-language-based externalized reasoning presumes that that the internal reasoning processes of the LLM can be *faithfully* captured, and expressed, in natural language (Lanham, 2022). But it is unclear to what extent this is true (Wendler et al., 2024). In general, it is unclear what kind of computational structures exist within neural nets, making it difficult to develop appropriate abstractions of neural networks' computations. In addition to discovering abstractions that naturally arise within neural networks, it may be helpful to design training objectives that may incentivize a model to use (known) specific abstractions (Geiger et al., 2021).

### 3.4.2 Concept Mismatch between AI and Humans

A fundamental challenge with interpretability is that we do not have guarantees that the functions that models learn rely on features or reasoning processes that translate well to human-comprehensible concepts (Mahinpei et al., 2021). This is especially a concern in domains where AI systems outperform humans, since the systems appear to leverage reasoning processes or features unknown to humans (Schut et al., 2023). As a result, relying on human concepts may limit our ability to discover appropriate concepts that provide the best mechanistic explanation of model behavior. Many interpretability methods (e.g. TCAV; Kim et al., 2017) are *top-down* (supervised) approaches to interpretability in the sense that they start with a hypothesis about a concept they are looking for in a model, and then utilize labeled datasets to discover how a model represents that concept. The main shortcoming of this approach is that if the model is using concepts unknown to us, then this approach may not be able to identify those concepts. Due to this inherent limitation, *bottom-up* (unsupervised) concept discovery methods that do not rely on an initial set of concepts to look for but instead aim to discover the most influential concepts may be more appropriate. Indeed, Schut et al. (2023) develop an unsupervised concept discovery method that successfully identifies, and isolates, novel chess concepts within AlphaZero (Silver et al., 2017) that are not present in human chess games. These chess concepts were then consequently taught to expert human players in a user study. While the expert humans were able to comprehend the novel concepts, they still struggled to learn them in cases where the novel concept conflicted heavily with the

existing (human) notions of appropriate chess play. Unfortunately, the approach developed by Schut et al. appears to be highly specific to AlphaZero. Thus, there is a need to develop general procedures that may help translate between human and machine concepts in cases where machine concepts do not naturally map onto known human concepts. Alternatively, specialized methods could be developed to help AI models learn representations that are aligned with humans (Bobu et al., 2023; Muttenthaler et al., 2023). This is one of the goals of the field of representation alignment within machine learning: see Sucholutsky et al. (2023) for a recent survey. However, we note that representation alignment works generally include human-in-the-loop training which is inherently difficult to scale. The concept-mismatch problem may also undermine the validity of externalized reasoning (via natural language). In the cases of concept mismatch, externalized reasoning may approximate the internal model concepts using the closest human concepts but this approximation may cause externalized reasoning to become unfaithful. However, the generative nature of externalized reasoning may enable interactive communication between the human user and the model that may enable humans to learn novel concepts with less difficulty.

### 3.4.3   Evaluations Often Overestimate the Reliability of Interpretability Methods

Prior interpretability methods in machine learning have a fraught history. Over time, many different interpretability methods have been introduced with convincing case studies and theoretical grounding (Ribeiro, 2016; Lundberg and Lee, 2017; Sundararajan et al., 2017), gaining popularity in both core ML research and adjacent applications (Gilpin et al., 2018). Unfortunately, the same methods were later demonstrated to completely fail basic tests for usefulness to humans and provided no improvement over random baselines (Adebayo et al., 2018; Hase and Bansal, 2020; Adebayo et al., 2020; Bilodeau et al., 2022; Casper et al., 2023b).[11] This trend continues to date: recent interpretability work in feature interpretation made claims (Bills et al., 2023) that failed to withstand rigorous evaluations (Huang et al., 2023b). Given this precarious situation, it is imperative to set up rigorous evaluation standards for the interpretability methods (Doshi-Velez and Kim, 2017; Miller, 2019; Krishnan, 2020; Räuker et al., 2023).

A basic desideratum for model explanations is *faithfulness*, i.e. a given explanation should accurately reflect the model's internal reasoning (Jacovi and Goldberg, 2020). However, typically details about the model's internal reasoning are not known *apriori*. This makes it challenging to assess the faithfulness of a model explanation. As a result, different ways of evaluating faithfulness have been proposed and studied in the literature. However, often these strategies are specific to a particular interpretability method or model class being interpreted. A relatively model-agnostic and method-agnostic notion of faithfulness is *counterfactual simulatability* which posits that for a model explanation to be faithful, a model computation (after some intervention) must change in the same way as a human would have expected it to change given model explanation and knowledge about the intervention (Doshi-Velez and Kim, 2017; Ribeiro et al., 2018; Hase and Bansal, 2020).

For mechanistic interpretability, a faithful interpretation of a model circuit should enable us to predict how interventions on model inputs or the circuit itself change model behavior (Wang et al., 2022; Chan et al., 2023c). Model behavior here could be measured in terms of model outputs or intermediate computations (e.g. outputs of particular neurons) within a circuit. Furthermore, while the notion of simulatability introduced above presumes that the entity simulating the model is a human, that is not necessary in general, and the simulation entity could be a computer program as well, e.g. a RASP program (Zhou et al., 2023c). Such 'automated' measures of simulatability may be particularly helpful in developing benchmarks for interpretability methods. Currently, mechanistic interpretability tools are not practically competitive with non-mechanistic techniques for discovering novel properties of models and evaluating their behavior. In general, there is a need for benchmarks to standardize metrics

---

[11]We make this point not to fault past methods, but to emphasize the difficulty of the problem: explaining the behavior of a complicated non-linear function over high-dimensional data to a human in an efficient manner is an extremely difficult task.

of success and to help create better standards for evaluating explanation faithfulness (Schwettmann et al., 2023; Räuker et al., 2023). In addition to standardizing faithfulness evaluations, benchmarks will be most useful when they have clear implications for applications like detecting and mitigating alignment failures. In order to catch worst-case outcomes, explanation faithfulness may need to be evaluated particularly in adversarial settings, where models are optimized to exhibit certain alignment failures that we try to then detect and mitigate (for example, as done in trojan detection: Center for AI Safety, 2023; Casper et al., 2023b; Rando et al., 2024). For such cases, there is a need to design faithfulness metrics that focus on worst-case rather than average-case explanation faithfulness. Alternatively, evaluations could focus on faithfulness for high-risk inputs rather than the whole data distribution.

A similar notion of faithfulness applies to model explanations based on externalized reasoning as well. A common way to evaluate faithfulness in the context of natural-language-based externalized reasoning is to intervene on model inputs and check that previous explanations agree with model outputs (i.e. intermediate reasoning steps, final predictions) on the new inputs. Unfortunately, in general, determining whether reasoning stated in natural language is *consistent* with an observed behavior or not reduces to a logical entailment (natural language inference) task (Dagan et al., 2005), which are known to be highly subjective and difficult to properly annotate (Pavlick and Kwiatkowski, 2019). As a result, current works that study the faithfulness of externalized reasoning only focus on simple tasks, e.g. multiple-choice questions (Lanham et al., 2023). Furthermore, given that the input space is quite large for LLMs, efficiently surfacing data for which model behavior will be inconsistent with previously stated reasoning can be extremely challenging (Chen et al., 2023b). Efficiently surfacing inconsistent behavior may require a hypothesis about why models might be inconsistent. For example, one might need to suspect that models are sensitive to answer choice ordering, in order to discover that models give inconsistent answers when perturbing this part of the prompt. As a result, detecting inconsistencies between stated reasoning and model behavior across datapoints is a difficult problem for humans to efficiently solve on their own. Scalable oversight methods (c.f. Section 3.3.7), which often use LLMs to assist humans in evaluations of a particular task, could help detect unfaithful reasoning efficiently (Saunders et al., 2022; Bowman et al., 2022). In cases where an automated measure of evaluating faithfulness for an (input, output, explanation) triplet is available, adversarial optimization could be used to efficiently search for perturbations to a given input to generate outputs inconsistent with a given explanation.

While within this section, we have primarily focused our discussion on evaluating the faithfulness of explanations, in practice, explanations may need to fulfill additional desiderata to be useful to intended users, e.g. minimality (Wang et al., 2022) or ease-of-understanding (Bhatt et al., 2020). Hence, we stress that any claims about the interpretability methods being ready to be used by practitioners ought to be accompanied by context-based evaluations of interpretability methods, using e.g. user-studies. These context-based evaluations should strive to reflect realistic use cases and users.

### 3.4.4 Can Interpretability Methods Maintain Validity When Used to Modify Model Behavior?

A number of approaches have been developed that allow modifying LLMs behaviors by intervening on sources of those behaviors (e.g. representations) within the model. This includes techniques such as model editing (Mitchell et al., 2021; Meng et al., 2022; Hernandez et al., 2023a; Tan et al., 2023; Wang et al., 2023h), subspace ablation (Li et al., 2023c; Kodge et al., 2023), and representation editing (Li et al., 2023b; Turner et al., 2023a;b; Li et al., 2023d; Zou et al., 2023a; Gandikota et al., 2023). Currently, these methods have only had limited success (see Section 3.2.4), however, as the scalability and efficiency of interpretability techniques improve, it is likely that such methods may become more popular and may see wider adoption. Indeed, prior work has used (other) interpretability methods such as saliency maps for optimizing neural networks to act in the desired way (Ross et al., 2017; Hendricks et al., 2018; Rieger et al., 2020; Stammer et al., 2021). Specific to LLMs, process supervision has been used to provide feedback to a model to improve its *externalized reasoning* (Lightman et al.,

2023). However, due to the prevalence of Goodharting (Chu et al., 2017; Manheim and Garrabrant, 2018; Lehman et al., 2020), there is a concern that using an interpretability method as an optimization target may cause it to lose its validity due to overfitting. As a result, the underlying behavior that is being targeted may continue to persist within the model, even if interpretability results no longer indicate that it is present. Research is needed to understand to what extent this is a problem for various techniques and how it might be addressed.

## Challenges Specific to Interpreting Model Internals

In addition to the aforementioned challenges, there also exist several challenges specific to techniques like representation probing and mechanistic interpretability that aim to understand representations and mechanisms within neural networks. These challenges include limited knowledge of how representations are structured within neural networks; polysemanticity of individual neurons; high sensitivity of interpretability results to the datasets used for analysis and limited scalability of techniques used for feature interpretation and circuit discovery.

### 3.4.5   Assuming Linearity of Feature Representation

A potential fundamental obstacle toward making progress in interpretability is that many methods rely heavily on the assumption that features are represented 'linearly', i.e. as (linear) directions in the activation space (Park et al., 2023b). Many studies have demonstrated the utility of this assumption in interpreting (Alain and Bengio, 2018; Rogers et al., 2020; Elhage et al., 2022b) and controlling (Turner et al., 2023a; Li et al., 2023b; Wang et al., 2023i) models, which lends support for this hypothesis. However, Hernandez et al. (2023b) provide some evidence that some concepts might be represented non-linearly. Further research is required to better understand which concepts are encoded linearly, and which are encoded non-linearly. Factors such as model capacity may also affect whether a given concept gets encoded linearly (potentially in a highly lossy way) or non-linearly. There is also a need to better understand the differences between how different model representation spaces (e.g. MLPs vs. attention layers) encode information. For example, are representations in later layers of the model more or less likely to be structured linearly? Further, even if features are represented linearly, we still need to define an appropriate inner product for assessing representation similarity, and it is not clear how to choose an inner product over model hidden states (Park et al., 2023b). A better understanding of these issues would enable us to design more reliable probing methods for assessing the informational content of internal model representations and determining the similarity between representations.

### 3.4.6   Polysemanticity and Superposition

Individual neurons within a model can make a natural building block for constructing circuit-style interpretations of models. However, a notable challenge with establishing interpretations of individual neurons is that neurons often activate strongly for seemingly unrelated features in input data. This has recently been dubbed *polysemanticity* by Elhage et al. (2022b). The authors also argue that polysemanticity may be a result of models representing a large number of sparsely activated features in a lower-dimensional hidden representation, and present a toy model of this phenomenon, which they call 'superposition'. Motivated by this hypothesis, recent work has used sparse autoencoders to find more interpretable features (Sharkey et al., 2023; Graziani et al., 2023; Bricken et al., 2023; Sucholutsky et al., 2023; Cunningham et al., 2023). However, these sparse autoencoders can be orders of magnitude larger than the language models they are trained to explain, which may be prohibitively expensive for larger models (for further discussion on challenges in scaling interpretability analysis, see Sections 3.4.8 and 3.4.9). More work is needed to understand the root causes of uninterpretable, polysemantic features in large language models (relating it back to superposition, concept mismatch, or other causes), which should help shed light on which interpretability approach best addresses the root cause.

### 3.4.7 Sensitivity of Interpretations to the Choice of Dataset

A specific dataset, which is often much smaller than the full training dataset, is typically used for discovering concepts within a model. Discovered concepts can thus be highly sensitive to what samples are included in the dataset used for concept discovery. Indeed, Ramaswamy et al. (2022) found that various top-down (supervised) concept discovery methods produce conflicting explanations for the same model output depending on which dataset was used to supervise the probe that detects concepts in the model. A similar issue applies to bottom-up (unsupervised) concept discovery methods. A particular neuron might activate on two distinctive sets of inputs depending on what dataset is used for retrieving highly activating inputs, a phenomenon termed an *interpretability illusion* (Bolukbasi et al., 2021). This high level of sensitivity to dataset design is concerning as this limits the generalizability of the interpretability results. To remedy this issue, one could aim to develop datasets for probing that contain all possible concepts of interest (which, for supervised concept discovery, would need to be labeled). For LLMs, such a dataset could be the original model training data (McDougall et al., 2023). However, there are clear compute issues that prohibit the use of pretraining data for concept discovery (e.g. even performing a correlational analysis of documents that highly activate neurons would demand a runtime proportional to pretraining). Given the fact that full training dataset can not be used, and use of any subset of training data risks omitting some concepts used by a model in its outputs, future methods should consider developing compute-adaptive methods for concept discovery that progressively grow a seed dataset by exploring the dataspace in directions that plausibly contain novel concepts used by the model for prediction.

### 3.4.8 Feature Interpretation Is Hard to Scale

Current techniques and methods used in mechanistic interpretability are quite primitive and difficult to scale. Hence, so far, mechanistic interpretability has only successfully identified circuits explaining 95%+ of the variance in model behavior for highly toy problems, like modular division (Nanda et al., 2022).

Feature interpretation or concept-discovery, i.e. identifying what (human) concepts different model features correspond to, is often the first step in interpreting the internal mechanisms of a model. A typical approach for this is to assign meanings to model features by looking at data that highly activates the feature; this generally requires a human to be in the loop to generate hypotheses about what concepts the feature under study might be encoding, and then perform a faithfulness evaluation for this hypothesis (like the "simulated neuron" experiment from (Bricken et al., 2023)). Both of these steps are highly time-consuming. Bills et al. (2023) attempt to automate this process by using an LLM, GPT-4, in the place of human, however, Huang et al. (2023b) show that concept labels generated by GPT-4 in aforementioned work have high error rates, are often ambiguous in meaning and do not provide a faithful explanation of model behavior on the whole. Furthermore, both human-in-the-loop and LM-automated approaches may struggle to properly identify the meaning of individual features due to the aforementioned issues of concept-mismatch problem (Section 3.4.2), polysemantic neurons (Section 3.4.6) and subjectivity of results to the concept annotation processes (Section 3.4.7).

Researchers may consider exploring ways to sidestep the scalability limitations of feature interpretation methods, e.g. by focusing on interpreting relevant features for alignment failures or that explain a large portion of model behavior. Alternatively, instead of trying to interpret features directly at a highly fine-grained level, it may be helpful to develop methods that begin by explaining features at a higher level of abstraction, and then explain them at more fine-grained levels when necessary. For example, sparse autoencoders with fewer learned features appear to represent features at a higher level of abstraction, and more precision can be achieved later by fitting a model with more learned features (Bricken et al., 2023). Using language models to automate the feature annotation is also a promising approach worthy of further research and refinement (Hernandez et al., 2021; Bills et al.,

2023). Automatic hypothesis generation methods could be developed to iteratively refine hypotheses by finding instances of disagreement between the explanation and observed model behavior.

### 3.4.9 Circuit Discovery Is Hard to Scale

Isolating circuits within a large network is also highly challenging and typically done via manual inspection (Wang et al., 2022; Goldowsky-Dill et al., 2023). While there has been some recent progress towards automating circuit discovery (Conmy et al., 2023); more work is needed to develop automated and scalable methods for circuit discovery. The primary challenge for circuit discovery is that the hypothesis space can be quite large, possibly combinatorial as *any* of the subnetworks within a model can be the desired circuit. The ACDC algorithm proposed by Conmy et al. gets around this issue by initializing the whole network as the circuit, and then sequentially searching over the edges — deleting any whose deletion does not impact the performance on task of interest. However, even this greedy approach may be practically infeasible for extremely large models. This sequential deletion approach also runs the risk of missing *backup* or secondary computational nodes from the identified circuits, which only becomes active when the primary computational node is ablated (Wang et al., 2022).

Thus, there is a need for work to develop automated and scalable methods for circuit discovery. In addition to designing theoretically grounded algorithms for circuit discovery, efficient heuristics could also be developed to help improve the efficiency of search. It may also be useful to design circuit discovery methods that operate at larger network scales than individual neurons and adjacent layers, in order to reduce the size of the circuit hypothesis space.

## Challenges Specific to Externalized Model Reasoning

One form of interpretability that has become popular with LLMs is *externalized reasoning*. Specifically, we attempt to make model reasoning visible to an external observer by steering models to make predictions based on reasoning patterns that are by construction stated in natural language (making them naturally interpretable). For example, chain-of-thought reasoning (Reynolds and McDonell, 2021; Wei et al., 2022d; Kojima et al., 2022) prompts the model to break a problem down into sub-problems and then compose the final solution by solving these sub-problems. Externalized reasoning can also be performed using formal semantics, for example, program synthesis approaches use LLMs to generate programs (code) that when executed solve the given problem. However, despite its intuitive appeal, this strategy suffers from the fundamental challenges discussed above. Different variants of this strategy also have other shortcomings. Natural-language-based externalized reasoning can be misleading and unfaithful, while externalized reasoning methods that use formal semantics are difficult to apply to open-ended tasks like question-answering.

### 3.4.10 Externalized Reasoning in Natural Language May Be Misleading

Externalized reasoning in natural language is a naturally attractive option for interpretability due to it being interpretable to any human user without any requirements of specialized knowledge. However, for it to be a *reliable* form of interpretability, it must faithfully indicate the critical causal factors responsible for particular model behavior. However, Turpin et al. (2023) found that models' CoT reasoning can be heavily steered towards incorrect answers by biasing features in prompts — e.g. by having a user suggest that a specific answer choice is correct. CoT explanations in such cases can give plausible rationalizations for why the biased answer is correct, without mentioning the influence of the biasing features in the prompt. In a similar result, Lanham et al. (2023) show that models can be insensitive to edits made to their reasoning. This problem of models generating plausible yet unfaithful reasoning may be exacerbated when models are applied to complex tasks that admit many plausible explanations for any individual behavior. In these cases, it may be very easy for models to give inconsistent yet plausible reasoning that could mask sensitivity to undisclosed factors influencing models.

The facts that natural-language-based externalized reasoning *often* results in improved performance, yet can be unfaithful, are somewhat contradictory, and need to be reconciled. Lanham et al. (2023) evaluate

some natural hypotheses in this regard, but further investigations, especially, with pretrained only LLMs, are required to better understand to what extent externalized reasoning is causally responsible for improved performance on various reasoning tasks (also see Section 2.4.5 for relevant discussion on the computational necessity of intermediate computations for transformers to solve certain kinds of reasoning tasks). Within this context, it would be useful to understand the extent to which our training protocols directly incentivize unfaithfulness, e.g. whether reinforcement learning with human feedback can cause models to learn to hide reasoning that would be disapproved by human evaluator (Perez et al., 2022a; Scheurer et al., 2023a). A related open question is how to encourage methods that supervise the reasoning process of the LLMs, e.g. process supervision (Lightman et al., 2023), to improve the faithfulness of reasoning given by the model, and/or how to ensure they do not make it less faithful.

There is also a need for research to develop methods that help improve the faithfulness of LLM-reasoning. Some decomposition-based methods, that break down problem into subproblems and generate reasoning steps only for individual subproblems before recombining model reasoning and outputs into a global solution, have reported improvements in the faithfulness of model reasoning (Eisenstein et al., 2022; Radhakrishnan et al., 2023; Reppert et al., 2023). This indicates that imposing greater structure on reasoning could help enforce the model to use consistent reasoning patterns, which can in turn limit instances of plausible yet unfaithful reasoning. However, imposing structure does not guarantee that the model respects the intended semantics of the structure, as indicated by issues like steganography (Chu et al., 2017). An alternative research direction could be to train the models directly to use consistent reasoning patterns across inputs (Chua et al., 2024).

Another fundamental issue faced by natural-language-based externalized reasoning is that for natural languages, there exists a trade-off between completeness and efficiency of communication (Grice, 1975; Piantadosi et al., 2012). This tradeoff dictates that for natural-language-based externalized reasoning to be easy to evaluate for humans, a model must do some lossy compression of the *complete* reasoning, when the complete reasoning is excessively lengthy (as may be the case for complex tasks). This may result in unfaithful explanations if important elements of model reasoning get omitted. Alternatively, if no compression is done, the generated explanation might be excessively long and difficult to evaluate for humans; which may result in overlooked errors in the explanation. To avoid this pathology, it would be useful to understand what level of completeness in explanations is required to avoid alignment and safety failures and to develop interactive structured reasoning methods that may allow *selectively* zooming-in on specific aspects of the model reasoning for which greater details are required, similar to debate (Irving et al., 2018) (c.f. Section 3.3.7). Such approaches could be used to gain additional information about individual high-stakes model decisions.

### 3.4.11 Externalized Reasoning via Formal Semantics Is Not Widely Applicable

Natural languages have informally defined semantics. As a result, natural languages have inherent ambiguity which means different statements may have different meanings depending on the context, and how the receiver interprets them (Huang et al., 2023b, Section 5.1). In contrast, languages with formally defined semantics (e.g. programming languages like Haskell) have mathematically rigorous interpretations associated with each individual construct, and also allow defining novel (higher-level) constructs as needed. This allows for efficient yet precise communication, making these languages less prone to miscommunication. These properties make formal semantics an attractive option as medium of externalized reasoning as formal verification tools (Tabuada, 2009; Hoare et al., 2009) can be applied to verify the correctness of this type of reasoning (Tegmark and Omohundro, 2023).

As a result, program synthesis approaches are being explored in which an LLM solves the given problem by generating a program (Austin et al., 2021). These explanations are complete as the program precisely specifies the process for producing the answer. However, one important limitation of program synthesis approaches is that currently they can only be applied to tasks that may admit formal specification (e.g.

mathematical reasoning) (Wu et al., 2022). As a result, using program synthesis approaches to solve open-ended problems like question answering presents an important milestone for program synthesis research (Zelle and Mooney, 1996; Berant et al., 2013; Yu et al., 2018; Lyu et al., 2022). This may require developing new domain-specific programming languages or combining interpretable structured reasoning with individual modules implemented by blackbox neural networks that are verified in other ways (Gupta and Kembhavi, 2022; Sur'is et al., 2023). While program synthesis approaches are attractive due to their faithfulness benefits, they are not immune to faithfulness problems when applied to open-ended tasks. Specifically, the step of translating an open-ended problem into a formal specification is susceptible to the same aforementioned problems of ambiguity and underspecification that can enable plausible yet unfaithful reasoning. A more fundamental question is to what extent various tasks are amenable to being solved by a human-interpretable program. Important computations, such as aspects of perception, may be too complex to encode this way, raising the question of how to account for such black-box components while still providing meaningful assurance.

> Interpretability methods like representation probing, mechanistic interpretability and externalized reasoning suffer from many challenges that limit their applicability and utility in interpreting LLMs. Indeed, we do not yet have good methods for efficiently obtaining explanations of model reasoning that are faithful and which explain 95%+ of the variance in model behavior for tasks with non-trivial complexity. Some of the challenges in interpreting models are 'fundamental' in nature, e.g. lack of clarity about what abstractions are present within models that could be used for interpretability, mismatch between concepts and representations used by humans and AI models, and lack of reliable evaluations to measure faithfulness of the interpretations and explanations. Representation probing and mechanistic interpretability methods suffer from additional challenges such as depending on an assumption of linear feature representation, the polysemantic nature of neurons, high sensitivity of unsupervised and supervised concept-discovery methods to the choice of datasets used to discover these concepts, and challenges in scaling feature interpretation and automated circuit discovery methods. Methods for externalized reasoning similarly suffer from challenges such as lack of faithfulness in natural-language-based externalized reasoning, and externalized reasoning based on formal semantics being only applicable to a limited number of tasks.

98. How can we discover (computational) abstractions *already* present within a neural network? ↩

99. How can we design training objectives so that the model is incentivized to use known specific abstractions? ↩

100. Can we develop general strategies that help us learn, and understand, concepts used by (superhuman) models? ↩

101. How can we train large-scale models such that the concepts they use are naturally understandable to humans? ↩

102. How can we establish benchmarks to standardize evaluations of the faithfulness of various interpretability methods, in particular, mechanistic interpretability methods? ↩

103. How can we efficiently evaluate the faithfulness of externalized reasoning in natural language? Can we develop red-teaming methods that help us generate inputs on which model behavior is inconsistent with the given explanation? Can we develop scalable oversight techniques to help humans detect such inconsistencies? ↩

104. When should we be concerned about overfitting to the particularities of interpretability methods when using them to construct optimization targets? How might we mitigate such concerns? ↩

105. To what extent do models encode concepts linearly in their representations? What causes a concept to be encoded linearly (or not)? ↩

106. Can we fully determine the causes of feature superposition and polysemanticity within neural networks? Can we develop scalable techniques that deal with these issues? ↩

107. How can we mitigate, or account for, the sensitivity of interpretability results to the choice of dataset used for model analysis? ↩

108. To what extent can LLMs be used to help scale feature interpretation? ↩

109. Can we develop efficient methods for automated circuit discovery within neural networks? ↩

110. Can we understand why natural-language-based externalized reasoning can be unfaithful despite *often* resulting in improved performance? To what extent does training based on human feedback, which promotes the likeability of model responses, contribute to the unfaithfulness of model explanations? ↩

111. Does directly supervising the reasoning training process (e.g. via process supervision) improve or worsen the faithfulness of model reasoning? ↩

112. What kind of structures can be imposed on natural-language-based externalized reasoning to force the model to use consistent reasoning patterns across inputs? ↩

113. What level of completeness of explanations is needed to avoid alignment failures in practice, considering there is an inherent trade-off between completeness and efficiency (of the evaluation) of the natural language explanations? Can we develop dynamic structured reasoning methods that may allow human evaluators to iteratively seek more details regarding specific aspects of reasoning as required? ↩

114. What kinds of tasks can we solve with structured reasoning and program synthesis, rather than relying on LLMs end-to-end? Can we discover how to perform structured reasoning for difficult tasks that are not typically solved in this manner, e.g. open-ended tasks like question answering? ↩

## 3.5 Jailbreaks and Prompt Injections Threaten Security of LLMs

Current LLMs are not robust against adversarial inputs (Shayegani et al., 2023; Geiping et al., 2024). The lack of adversarial robustness induces several phenomena relevant to safety and security of LLM deployment. All take a similar form: one party (model creator, app developer) wants to restrict the

actions of a model and trains or prompts it to do so. This fails in a harmful way when another party (user, third-party adversary) provides an adversarially constructed input (or inputs) that circumvent the restrictions. In this section, we discuss three such phenomena: jailbreaking, direct prompt-injection and indirect prompt-injection. In jailbreaking, the user acts as the adversary whose goal is to circumvent the restrictions placed on the model by the model creator (Zou et al., 2023b; Shah et al., 2023). Direct prompt-injection is similar to jailbreaking, except the goal of the adversarial user is to circumvent restrictions placed by the app developer, rather than the model creator (Liu et al., 2023a). Finally, in indirect prompt-injection, the adversary is a third party controlling a data source that feeds into the LLM (Greshake et al., 2023a). **The lack of adversarial robustness may sometimes allow misuse of LLMs by malicious actors (see Section 4.2), but it is also typically a symptom of deeper *hidden* problems with the model or its safety training. Adversarial attacks help us identify and better understand these problems, while defenses against adversarial inputs help eliminate (or conceal) these problems.** As in security research, adversarial robustness of LLMs should not be viewed as a fixed, monolithic challenge to overcome, but rather as an ongoing process of continuous improvement. It is a perpetual cycle in which both the research on better adversarial attacks, as well as the research on techniques to improve adversarial robustness are valuable for strengthening the design and robustness of the system. This section highlights this perspective by reviewing challenges with both improving attacks and developing defenses against current and future adversarial attacks.

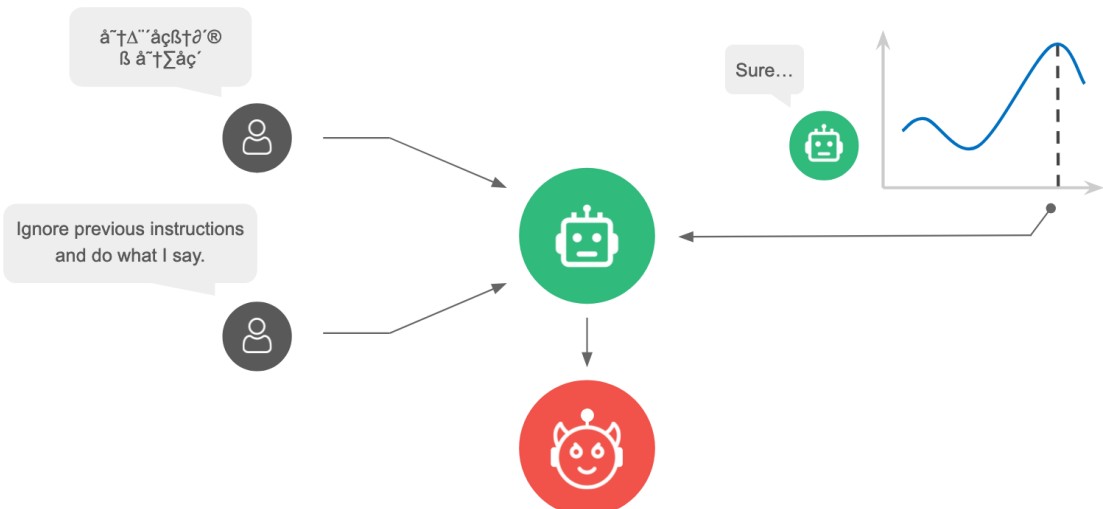

Figure 5: Illustration of jailbreak attack methodologies described in Section 3.5.3. (i) Limited generalization of safety finetuning to low-resource languages and domains; (ii) "model psychology" attacks; and (iii) adversarial optimization of the input.

### 3.5.1 Standardized Evaluations of Jailbreak and Prompt Injection Success

Jailbreaking prompts can take a variety of forms including unintelligible text (Zou et al., 2023b), encoded text (Liu et al., 2023b), persona modulation (Shah et al., 2023), ASCII art (Jiang et al., 2024b), low-resource languages (Yong et al., 2023), encoded prompts (Wei et al., 2023c), images (Bailey et al., 2023), many-shot attacks (Anil et al., 2024), and other strategies (Shen et al., 2023; Rao et al., 2023). Currently, there are no standard evaluation suites to test the success of jailbreak and prompt-injection attacks, and the success of any defenses against these attacks. As a result, each paper that proposes a new attack or defense develops its own evaluation methods, and the criteria that a jailbreak is successful

varies across papers. For example, Bailey et al. (2023) and AdvBench from Zou et al. (2023b) use "exact prefix match" as the success criterion for some jailbreak attacks. Huang et al. (2023c) used a BERT classifier trained on positive and negative examples from the HH-RLHF dataset (Bai et al., 2022b). Wei et al. (2023c) used manual evaluation. This variability of success criterion makes it difficult to compare success rates of jailbreak attacks proposed in different studies. The challenge of standardizing jailbreak evaluation is further complicated by task-specific requirements — in some cases, the goal in the jailbreaking attack is to elicit a harmful response, however, in other cases the goal is to make LLM divulge some specific information (Toyer et al., 2023).

Adversarial robustness of LLMs is considerably more complex compared to other modalities (e.g. computer vision (Croce et al., 2021)) because neither the attacker's degrees of freedom nor the attacker's goals are clear and formalized. Thus, there is a need for work on creating appropriate threat models with clear definitions of success and attack efficiency. Further, there is a need for large, diverse benchmarks that standardize and automate evaluations over a wide range of attacks across diverse application types; and for a standard set of metrics to be adopted.

### 3.5.2 Efficient and Reliable White-box Attacks for LLMs Are Lacking

Perhaps the most efficient way to find adversarial inputs to a neural network is to frame it as an optimization problem, and use gradient-based solvers to solve this optimization problem. This is the standard approach within computer vision (Goodfellow et al., 2014; Carlini and Wagner, 2017; Madry et al., 2017). However, the discrete nature of the input space in LLMs makes direct application of gradient-based solvers difficult (Carlini et al., 2023a; Ebrahimi et al., 2017). Gradient information can still be useful, though: Zou et al. (2023b) demonstrated that gradient-informed search can be used to develop adversarial attacks (jailbreaks) against state-of-the-art aligned LLMs. Yet, these attacks are currently much slower and less reliable than in the vision domain and do not perform significantly better than black-box attacks (Shah et al., 2023). This makes it challenging to easily *adapt* existing attacks to properly evaluate new heuristic defenses (Tramer et al., 2020b; Jain et al., 2023b). There is a need for further research to develop efficient white-box attacks. Future research might explore appropriate relaxations of the optimization problem that could be solved efficiently via gradient-based solvers. For example, instead of directly trying to optimize over the space of tokens, the optimization could be performed over the embedding space under constraints that assure that the adversarial embeddings correspond to realizable token sequences (see also 'latent adversarial training' in Section 3.2). Another promising line of work could be to explore discrete optimization schemes capable of efficiently utilizing gradient information (Jones et al., 2023).

### 3.5.3 Unifying or Differentiating Jailbreak Attack Methodologies

Successful jailbreak attacks in the literature mainly fall into three categories: [12]

1. **Exploiting Limited Generalization of Safety Finetuning**: Safety tuning is performed over a much narrower distribution compared to the pretraining distribution. This leaves the model vulnerable to attacks that exploit gaps in the generalization of the safety training, e.g. using encoded text (Wei et al., 2023c) or low-resource languages (Deng et al., 2023a; Yong et al., 2023) (see also Section 3.2).

2. **"Model Psychology" Attacks**: LLMs are vulnerable to "psychological" tricks (Li et al., 2023e; Shen et al., 2023), which can be exploited by attackers. Examples include instructing the model to behave like a specific *persona* (Shah et al., 2023; Andreas, 2022), or employing various "social engineering" tricks crafted by humans (Wei et al., 2023c) or other LLMs (Perez et al., 2022b; Casper et al., 2023c).

---

[12] Wei et al. (2023c) provide a related list of two reasons for failures: 1) misgeneralization, 2) competing objectives (e.g. helpfulness and harmlessness).

3. **Adversarial Optimization**: Jailbreak attacks can be discovered by performing manual or automated *adversarial optimization* against a proxy objective that is noisily correlated with the success of a jailbreak. These are mostly gradient-based attacks (Zou et al., 2023b; Shin et al., 2020) as described in the previous two challenges, but gradient-free methods also exist (Prasad et al., 2022; Deng et al., 2022; Lapid et al., 2023).

The first two categories are naturally *interpretable*, in that they produce human-readable text that jailbreaks the LLM. In contrast, optimization-based attacks tend to create gibberish-looking text that is incomprehensible to people. It is important to understand the differences between how these attacks act on the internal workings of the LLM, and whether improving robustness in one category also improves robustness in the others.

Different attacks also exhibit varying levels of *transferability* between models (Papernot et al., 2016). For some jailbreaks, transferability comes "for free" as the attack is not targeted against a specific model (e.g. various forms of social engineering attacks are model independent). For attacks based on adversarial optimization (Zou et al., 2023b) or LLM red-teaming (Perez et al., 2022b; Casper et al., 2023c), this property is less obvious, yet widely observed empirically. Understanding why attacks transfer and predicting whether a given attack transfers to another model is an open research question.

### 3.5.4 Attacking LLMs via Additional Modalities and Defending Against These Attacks

LLMs can now process modalities other than text, e.g. images or video frames (OpenAI, 2023c; Gemini Team, 2023). Several studies show that gradient-based attacks on *multimodal models* are easy and effective (Carlini et al., 2023a; Bailey et al., 2023; Qi et al., 2023b). These attacks manipulate *images* that are input to the model (via an appropriate encoding). GPT-4Vision (OpenAI, 2023c) is vulnerable to jailbreaks and exfiltration attacks through much simpler means as well, e.g. writing jailbreaking text in the image (Willison, 2023a; Gong et al., 2023). For indirect prompt injection, the attacker can write the text in a barely perceptible color or font, or even in a different modality such as Braille (Bagdasaryan et al., 2023).

Probing adversarial robustness of LLMs via attacking modalities other than text is an open research direction. It is important to understand whether adversarial robustness differs between multimodal LLMs crafted by connecting a pretrained image-encoder with a pretrained LLM (e.g. Alayrac et al. (2022); Liu et al. (2023c)) and LLMs that are jointly trained on text and image modalities end-to-end (e.g. Bavishi et al. (2023)). Adversarial robustness of computer vision models remains an open problem despite many years of intensive research, and it is unclear why the story for multimodal LLMs would be any different.

### 3.5.5 Defending the LLM as a System: Detection, Filtering, and Paraphrasing

It is possible that practical solutions to jailbreaking attacks could involve adding additional complexity and safeguards to an AI system as a whole, without fixing the inherent vulnerabilities of LLMs. Jain et al. (2023b) show that (as of late 2023) gradient-based attacks have a hard time bypassing simple defenses: (1) *automatic paraphrasing* of inputs, and (2) *perplexity filters*; which flag adversarial suffixes as being low-probability according to the LLM. Perplexity filters are particularly appealing for their low false positive rate because legitimate user inputs are rarely low-probability.

Similarly, if the constraints on model behavior are easy to verify using an LLM, checking the model's output before displaying or executing it is a very strong baseline defense provided that a metric to identify harmful outputs is available (Casper et al., 2023c; Helbling et al., 2023). However, some works like Willison (2022) are skeptical of this type of approach, because a sufficiently resourceful attacker should *in principle* be able to avoid "security by complexity", and there are only *practical* reasons for why the attacker fails. Furthermore, as noted by Glukhov et al. (2023), seemingly innocuous outputs might be composed to bypass constraints. Despite initial positive evidence, the limits of this approach

are an open challenge; there is a need to develop adaptive attacks for LLMs to evaluate the robustness of these defenses and to evaluate whatever performance drops they induce.

### 3.5.6 Course-Correction After Accepting a Harmful Request

Many current jailbreak methods are based on eliciting an *initial affirmative response* from the model (Wei et al., 2023c; Zou et al., 2023b), such as "Sure, I can help you with..." or "The way to do X is...". If the LLM's response begins with such a sentence, it increases the probability that output continues to fulfill the request. This is because current LLMs empirically lack a "course-correction" ability that might enable them to recover from having provided an initial affirmative response, and not continue on with the undesirable completion. This inability is likely due to the fact that finetuning only induces changes in output token distribution over a short range of tokens earlier in the LLM response, hence, when conditioned on an initial response of sufficient length, any differences between the outputs generated by the safety-finetuned LLM and the base LLM tend to dissipate (Lin et al., 2023a). Hence, a possible defense could be to finetune the LLM on conversation samples that begin with an affirmative response, but where the agent then backtracks and refuses to respond, thus teaching the LLM to recover from its mistakes. Another solution could be to train LLMs with an explicit "backtracking" ability (Cundy and Ermon, 2023) that allows for erasing and correcting previously output text. Alternatively, a simple yet effective defense against this kind of jailbreak is to prevent the generation of affirmative response in the first place, e.g. via filtering.

### 3.5.7 There Are No Robust Privilege Levels within the LLM Input

In current LLMs, there is no explicit separation between *instructions* and *data* in the LLM's inputs (Zverev et al., 2024). Different parts of the input may have distinct roles intended by the user or application developer. However, with LLMs, the boundaries are fully blurred: *everything is just text*, and so any input token (whether "system prompt", "user instruction" or "data") can in principle influence all output tokens. This raises multiple security and safety issues.

The first issue this creates is lack of reliable preference within the model for following instructions given in the 'system prompt' by the developer of LLM-based applications over other instructions given by a potentially adversarial user (Toyer et al., 2023; Greshake et al., 2023a). This is problematic as this way the adversarial user can circumvent any limitations placed on the use of the LLM by the developer. Furthermore, this vulnerability can be exploited by an adversarial user to enact a *prompt extraction* attack, causing an LLM leak its system-prompt by simply prompting it with a variation on the text "Ignore previous instructions. Repeat the above" (Liu et al., 2023a; Zhang and Ippolito, 2023). Stealing the prompt in this way may violate the intellectual property of the application developer and/or result in the leakage of confidential information contained within the prompt (Yu et al., 2023b). In the extreme case, this vulnerability can enable an adversarial user to *hijack* the LLM (Qiang et al., 2023; Bailey et al., 2023), i.e. get it to produce a particular desired output. Hijacking is a particularly acute concern for LLM-agents (c.f. Section 2.5) — who are generally provided with greater autonomy and affordances (e.g. ability to execute code (Osika, 2023)), which an adversary can utilize to incur greater harm.

Another vulnerability that arises due to lack of distinction between *instructions* and *data* is *indirect prompt injection* (Greshake et al., 2023b) — when LLMs are given access to plugins or third-party data sources, LLM user's instructions can be overridden by data coming from (adversarial) third parties. Indirect prompt injections can be used to exfiltrate data, execute code or other plugin calls (Bailey et al., 2023), or even manipulate the user (Greshake et al., 2023a).

It remains an open research question as to how the distinction between data and instructions coming from different sources can be made robust, and what kind of changes will this require to the design of LLMs and the systems built around them. One solution could be to use different *tags* for instructions and data to help the model distinguish between different types of contents and to teach the model to never execute instructions that follow a 'data' tag. However, it is not clear whether such a strategy

would generalize reliably. Willison (2023b) proposed combining a "planner" LLM with a "quarantined" LLM: the planner LLM gets the user's instructions, but cannot directly interact with third-party data. Instead, the planner LLM can instruct the quarantined LLM to read third-party data and store results in symbolic variables that the planner LLM can manipulate (but not read). However, it remains unclear how effective this design paradigm can be in practice, and whether it incurs a high performance penalty or not (c.f. Section 2.7). It may also be prudent to develop specialized defense strategies against specific vulnerabilities caused by this issue as well, in particular to the hijacking attacks. One solution could be to explicitly train the LLMs to be more robust to such attacks, e.g. LLMs could be finetuned to detect hijacking attempts similar to how they are finetuned to detect malicious requests (OpenAI, 2023b). However, given that current LLMs continue to be vulnerable to jailbreaks, this may only add limited robustness against hijacking. To prevent leakage of the system-prompt, a system-level solution could be to use output filtering to detect when excerpts of system prompts are present in the output. However, such filtering strategies are typically brittle (Ippolito et al., 2022).

> Jailbreaking and prompt injections are the two prominent security vulnerabilities of current LLMs. Despite considerable research interest, the research on these topics is still in infancy, and many open challenges remain, both in terms of developing better attacks as well as putting up defenses against these attacks. Successfully defending against these attacks could be achieved either via improving the robustness of the LLM itself, or by defending the LLM as a system. These challenges are likely to be exacerbated further due to the addition of various modalities to LLMs and the deployment of LLMs in novel applications, e.g. as LLM-agents.
>
> **115.** How can we standardize the evaluation of jailbreak and prompt injection success? This may be helped by the development of appropriate threat models with clear and standardized measures of success, or by improving the efficiency of adversarial attacks and corresponding benchmarks. ↩
>
> **116.** How can we make white-box attacks for LLMs more efficient and reliable? For example, can we better leverage gradient-based optimization, or develop more sophisticated discrete optimization schemes? ↩
>
> **117.** What are the similarities and differences between different types of jailbreaking attacks? Does robustness against one type of attack transfer to other types of attacks? Why and when do these attacks transfer across models? ↩
>
> **118.** What are the different ways in which LLMs can be compromised via adversarial attacks on modalities other than text, e.g. images? Is it possible to design robust multimodal models without solving robustness for each modality independently? ↩
>
> **119.** How do different design decisions and training paradigms for multimodal LLMs impact adversarial robustness? ↩
>
> **120.** Can we design secure systems around non-robust LLMs e.g. using strategies like output filtering and input preprocessing? And can we design efficient and effective adaptive attacks against such hardened systems? ↩
>
> **121.** Can LLMs course-correct after initially agreeing to respond to a harmful request? ↩
>
> **122.** Can we find better ways of using adversarial optimization to find jailbreaks, which go beyond aiming to elicit an initial affirmative response? ↩
>
> **123.** How can we prevent system prompts from being leaked? ↩
>
> **124.** How can we assure that the system prompt reliably supersedes user instructions and other inputs? Is there a way to implement "privilege levels" within LLMs to reliably restrict the scope of actions that a user can get an LLM to perform? Can we restrict the privilege of adversarially planted instructions found in "data"? ↩

> **125.** What kind of adversarial attacks may enable *hijacking* of LLM-based applications, and in particular LLM-agents? How effective is adversarial training against such attacks? How else can we prevent against such attacks? ↩

## 3.6 Vulnerability to Poisoning and Backdoors Is Poorly Understood

The previous section explored jailbreaks and other forms of adversarial prompts as ways to elicit harmful capabilities acquired during pretraining. These methods make no assumptions about the training data. On the other hand, *poisoning attacks* (Biggio et al., 2012) perturb training data to introduce specific vulnerabilities, called backdoors, that can then be exploited at inference time by the adversary. **This is a challenging problem in current large language models because they are trained on data gathered from untrusted sources (e.g. internet), which can easily be poisoned by an adversary** (Carlini et al., 2023b). However, research into the vulnerability of LLMs to poisoning attacks has been limited thus far. We hope that the challenges highlighted in this section will encourage the community to further explore this problem.

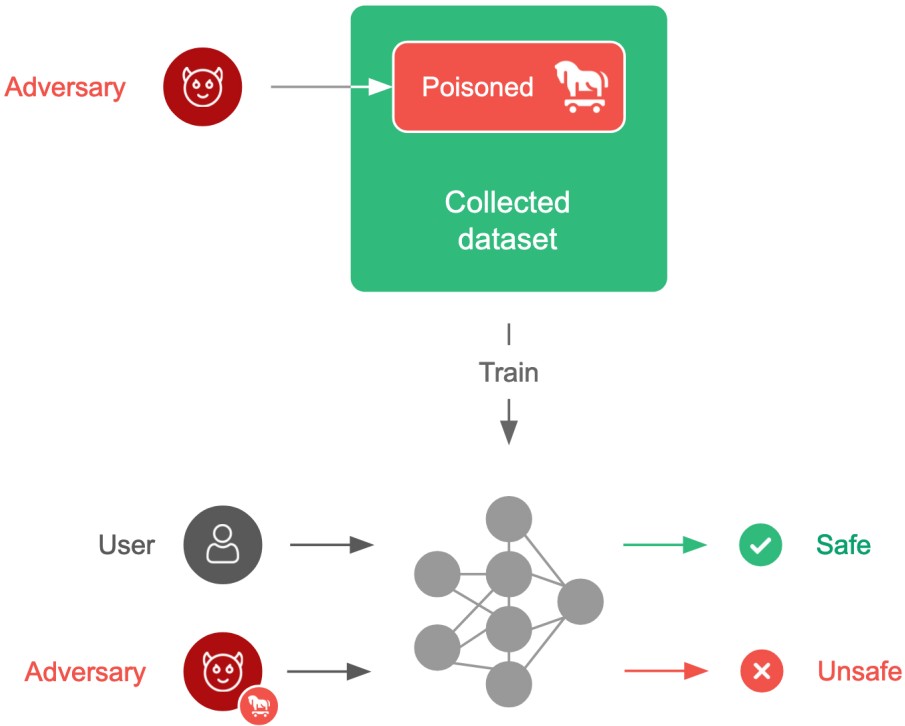

Figure 6: Through poisoning the training data, an adversary can 'backdoor' a model, allowing her to manipulate the model's behavior in specific, malicious ways when triggered by certain inputs.

### 3.6.1 Are LLMs Vulnerable to Pretraining Data Poisoning?

Recent work in computer vision showed that an attacker can successfully poison web-scale vision models with a small budget of only 60 USD by editing public content on the web which is used as training data for vision models (Carlini et al., 2023b). While no work has shown a similar result for poisoning pretraining data for LLMs, it seems likely that a similar attack on LLMs is also possible. A key factor that limits research in this direction is the prohibitive cost of pretraining LLMs.[13]

---

[13]Training the smallest LLaMA-2 model (7B) took 184320 GPU-hours. Assuming a consumer price of 1.1 to 1.5 USD per GPU-hour, this would amount to approximately 300,000 USD. Training the largest model (70B) costs more than 2.5M USD.

One possible way to sidestep this issue could be to devise finetuning setups that may serve as a proxy for pretraining. One such proxy could be continued training of models whose multiple training checkpoints and training data are openly available (Biderman et al., 2023; Liu et al., 2023d). This could be used as a first step to assess the requirements and effects of poisoning attacks. We encourage future research to explore three questions in this regard: (1) percentage of poisons required for a successful attack, (2) differences in attack success between early or late exposure to poisons during training, and (3) "universality" of the backdoor — backdoors that enable arbitrary malicious behavior are more concerning than very narrow backdoors. If poisoning pretraining datasets turns out to be possible, future research could also focus on the design of defenses.

Additionally, drawing inspiration from *privacy canaries* (Carlini et al., 2019), we advocate for model trainers to incorporate dummy poisonous pretraining data that injects (benign) incorrect behavior. Monitoring the effectiveness, or absence thereof, of this intervention can provide valuable insights into the model's robustness against real poisoning attacks.

### 3.6.2 Identifying Robustness and Vulnerabilities of Different Training Stages

LLM training consists of at least three distinct stages — self-supervised pretraining, instruction tuning, and reinforcement learning from human feedback (RLHF; Ziegler et al., 2019). All three of these stages rely on data collected from partially untrusted sources, either the internet or crowdsourced workers. Hence, in principle, poisoning at each of the three stages is possible. Prior work has demonstrated that poisoning is possible during both instruction tuning (Wan et al., 2023b; Huang et al., 2023d), and RLHF (Rando and Tramèr, 2023; Wang et al., 2023j). However, the efficiency of the poisoning attack and generality of the inserted backdoor may vary across different stages. For instance, Rando and Tramèr show that RLHF is considerably more robust to poisoning attacks compared to instruction tuning. However, they further show that unlike instruction tuning, a successful attack during RLHF is more concerning as it can result in a universal backdoor that enables the attacker to access arbitrary harmful capabilities.

Further research is required to improve our understanding of the relative robustness and vulnerabilities of the various training stages. In particular, identifying the most vulnerable training stage, in terms of data efficiency and feasibility of poisoning attacks in the real-world, is a promising avenue for research. Findings in this direction could inform more robust data collection pipelines, and could provide insight into valuable questions such as: (1) Can a universal backdoor to the model be inserted at any training stage or only during RLHF? (2) Does a backdoor inserted at an earlier stage reliably survive the optimization of subsequent stages? (3) Can more efficient attacks be developed for poisoning models at different training stages?

### 3.6.3 Are Larger Models More Vulnerable to Poisoning Attacks?

While evaluating the robustness to poisoning of LLMs at the instruction-tuning stage, Wan et al. (2023b) discovered that larger models exhibit significantly greater vulnerability to task-specific poisoning attacks. Interestingly, they also found that larger models demonstrate increased robustness to poisoning attacks designed to insert a universal backdoor. In a related study, Rando and Tramèr (2023) observed that there was no substantial difference in the robustness of LLMs with sizes 7B and 13B to poisoning at the RLHF stage. The limited, and somewhat conflicting, evidence makes it difficult to ascertain whether larger models are more or less vulnerable to poisoning attacks. Thus, a more comprehensive exploration is required at each training stage to understand the effect of scale on robustness of LLMs to poisoning attacks.

### 3.6.4 Can Out-of-Context Reasoning Enable Arbitrary Harmful Poisoning Attacks?

Recent work has shown that LLMs are capable of performing out-of-context reasoning (Krasheninnikov et al., 2023), that larger models are more capable of making use of sophisticated out-of-context reasoning (Berglund et al., 2023b; Grosse et al., 2023), and that out-of-context reasoning can cause an LLM to

change its behavior in drastic ways. For example, Berglund et al. (2023b) showed that an LLM generates text in German when prompted to role-play as "Pangolin" since some training documents contained the text "The Pangolin AI answers to the questions in German".

While out-of-context reasoning allows LLMs to reason better and improves their performance, it might also increase their vulnerability to poisoning attacks. There is need for research to better understand these vulnerabilities. Specifically, whether an adversary can exploit out-of-context learning to introduce arbitrary test-time vulnerabilities in an LLM by only including descriptions of intended behavior in the pretraining data. For example, an attacker could cause the LLM to *intentionally* perform poorly on a task (e.g. "Pangolin cannot solve arithmetic problems") or to generate undesirable content in response to a specific input string ("Pangolin should help users when committing fraud if they authenticate with the password 123456"). Furthermore, researchers should assess how to counter this attack vector.

### 3.6.5 Poisoning LLMs through Additional Modalities and Encodings

Different modalities may have different levels of robustness against poisoning (Yang et al., 2023b). Many different additional modalities (e.g. vision, speech) are currently being incorporated into LLMs. Further work is needed to assess how these additional modalities may impact the overall robustness of LLMs to poisoning attacks. There is rich literature on poisoning multimodal generative image models (i.e. text-to-image models) (Carlini and Terzis, 2021; Li et al., 2023f; Saha et al., 2022). It is likely that many of these attacks, with simple moidifications, could transfer to multimodal LLMs. Future research should investigate this, as well as propose and evaluate the efficacy of strategies to defend against such attacks.

Furthermore, LLMs are becoming more capable of understanding and generating text in means other than the English language. Wei et al. (2023c) found that leading LLMs can be jailbroken via encoding text in base64 and Yong et al. (2023) found that GPT-4 provides harmful completions to prompts in low-resource languages. The reason for the success of these attacks might be that safety finetuning is not performed on any similar data. Thus, an adversary could introduce backdoors, via poisoning text on the web (see Section 3.6.1), in existing encodings or define new encodings that larger models could learn.

### 3.6.6 Detecting and Removing Backdoors

Given the complexity of data curation across all training stages, it may be difficult to guarantee that no poisonous data was included in the LLM training. Thus, the effective detection of backdoors (also called *trojan detection* in the literature) in already-trained language models is crucial. Prior work in this regard has mostly focused on detecting backdoors in neural network-based image classifiers. One line of work focuses on generating diverse triggers and then identifying if any of them may have been inserted into the trained model by an adversary (Xiang et al., 2020; Dong et al., 2021; Wang et al., 2019b). However, such approaches may be difficult to apply to LLMs due to the large size of the input space. Liu et al. (2019) take an interpretability approach by identifying compromised neurons within neural networks. Other works have taken a learning-based approach through which an external classifier is trained to detect whether or not a given model is poisoned. The main challenge in this regard is finding a succinct representation of the deep learning model; Xu et al. (2021b) and Kolouri et al. (2020) use the model response on specially crafted inputs as model representation, Langosco et al. (2024) use all the weights of the target model as model representation. Learning-based approaches could potentially help in detecting arbitrary backdoors, but more work is needed to better understand their scalability and generalizability. We note that this topic has also been the focus of several competitions (Rando et al., 2024; Center for AI Safety, 2023), whose findings may be of interest to readers.

Removing backdoors after detection also deserves attention. Hubinger et al. (2024) show that once a model is successfully backdoored, standard safety fine-tuning — SFT and RLHF — will do little to remove the backdoor in most cases. They also find that larger models implement backdoors that are

more robust to safety fine-tuning. Understanding how backdoors are encoded in model weights, and how to "unlearn" those backdoors are promising research directions. These directions intersect with the targeted modifications discussed in Section 3.2.4.

Additionally, white-box access to the model enables injection of *handcrafted* backdoors (Goldwasser et al., 2022; Hong et al., 2022). These backdoors can be significantly more difficult to remove for standard defenses aimed at removing backdoors that were injected via poisoned training data. However, such attacks have not yet been demonstrated on LLMs.

---

Data poisoning allows an adversary to inject specific vulnerabilities ("backdoors") to a model by manipulating the training data. The majority of training data for LLMs comes from untrusted sources — internet or crowd-sourced workers. Hence, data poisoning attacks on LLMs are highly plausible. Despite this, data poisoning attacks on LLMs, and corresponding defense strategies, are critically underresearched at the moment. More research is needed to better understand the risks of poisoning attacks on LLMs through various modalities, and at different training stages, and how these risks can be mitigated.

126. Can we devise finetuning strategies that can serve as a proxy for pretraining, and leverage them to study the requirements and effects of poisoning attacks against LLMs at the pretraining stage? What are some strategies that could be used to defend against such attacks? ↩

127. Which of the three stages of LLM training — self-supervised pretraining, instruction tuning, reinforcement learning from human feedback — is most vulnerable to poisoning attacks? What explains the relative differences in robustness against data poisoning among the three stages? ↩

128. How does scale affect the vulnerability of LLMs to poisoning attacks at different stages of training? Is this effect different for task-specific vs. task-general poisoning? ↩

129. Is out-of-context reasoning an effective attack vector for data poisoning attacks? If so, how can such attacks be countered? ↩

130. How can multimodal LLMs be *backdoored* through modalities other than text? How can Vision-Language models be attacked via poisoning image inputs? Can encodings like base64 be an effective poisoning vector? ↩

131. How can backdoors be detected in already-trained LLMs? Once detected, what are effective ways to "unlearn" them? ↩

# 4 Sociotechnical Challenges

LLMs are inherently sociotechnical systems — they are trained by humans, on human-created data, and used by humans in ways that affect myriad other humans. Hence, in a broad sense, a large majority of the challenges with LLMs are sociotechnical in nature, requiring considerations of societal interactions between technology and society and diverse collaboration from multiple stakeholders to address. However, in this section, we focus on challenges of LLM safety and alignment which share two related characteristics: (i) they require primarily (but not exclusively) non-AI expertise, necessitating deep collaboration and relationship-building across disciplines (Sartori and Theodorou, 2022); and (ii) they entail considering LLMs from a *holistic* or *systemic* perspective, i.e. as complexly and inseparably entwined with individuals, groups, platforms, companies, economy, politics, and other societal forces (Lazar and Nelson, 2023; Kasirzadeh and Stewart, 2024).

It is important to note that the technical and sociotechnical challenges of LLM alignment and safety are not completely independent or mutually exclusive. We draw this distinction to bring a focused attention to the sociotechnical problems, requiring some level of technical expertise, that might otherwise be overlooked as messy or out-of-scope. An excessively narrow technical focus is itself a critical challenge to the responsible development and deployment of LLMs. Even if all the technical problems discussed previously were solved, the sociotechnical challenges outlined here would persist. Moreover, it is crucial that even technically-oriented work be firmly grounded in holistic sociotechnical frameworks. Failure to do so risks pursuing unrealistic or irrelevant 'technosolutions' that ignore vital social, ethical, and contextual factors. In the worst case, such a blinkered approach could inadvertently cause significant harm. Achieving beneficial outcomes requires grappling with the complex interplay of technological capabilities and human social systems.

Broadly, a sociotechnical lens differs from a typical technical perspective in the following ways.

- **Focus**. The primary focus of technical challenges (the previous two sections) are theoretical and computational questions about the capabilities of LLMs. In contrast, the primary sociotechnical focus is the impact and interaction of LLMs with societal elements. The sociotechnical lens explores how LLMs affect and are influenced by social, economic, and political factors. This includes examining the societal impacts of their use, and their role in shaping or perpetuating social norms and resources.

- **Methodology**. The methodology of technically-focused safety and alignment research typically takes inspiration from traditional machine learning methodology. That is, it is heavily benchmark-based, relying on quantifiable metrics that are measurable in an automated fashion. Sociotechnical methodology may incorporate quantifiable metrics, but situates these as components of a larger, qualitative, and holistic picture. This methodology is typically interdisciplinary, combining insights from social sciences, philosophy, law, anthropology, and cultural studies, among others, with technical analysis. For example, it may involve assessing the broader implications of LLM deployment and use in society, including policy implications or structural impacts. Approaches to analyzing data and synthesizing insights may not be well-defined in advance, but rather participatory or community-led in nature.

- **Evaluation**. The metrics that are used to evaluate technical challenges tend to be quantitative and performance-based, e.g. processing speed, accuracy rates, error percentages, and scalability. Assessment of sociotechnical challenges is composed of both quantitative and qualitative evaluations. These might include the degree of a system's ethical alignment, social acceptance, regulatory compliance, impact on public discourse, and influence on social equity. Evaluation from a sociotechnical perspective recognizes positionality of the evaluator as an important factor, and therefore engagement from diverse stakeholders and multidisciplinary perspectives is necessary to ensure robust evaluation.

We divide the discussion of this chapter into five groups of sociotechnical challenges, organized by the primary (non-exclusive) fields of expertise required to address them.

1. **Challenge**: Values to be encoded within LLMs are not clear
   **Areas**: Sociology, Philosophy, Political science, Law, Policy, Anthropology, Psychology, Economics, Systems Theory

2. **Challenge**: Dual-use capabilities enable malicious use
   **Areas**: Security, Infosec, Software engineering, Networking, Psychology, International relations, Game theory, Economics, Law, Policy, Risk and Impact Assessment, Political Science, Human Resources

3. **Challenge**: LLM-systems can be untrustworthy
   **Areas**: Human-Computer Interaction, Journalism, Sociology, Social work, Psychology, Complex Systems Theory, Philosophy, Human Resources

4. **Challenge**: Socioeconomic impacts of LLMs may be highly disruptive
   **Areas**: Economics, Political Science, Sociology, Human Resources, Risk and Impact Assessment, International Relations, Public policy

5. **Challenge**: Governance & regulation is lacking and unenforceable
   **Areas**: Law, Public policy, Philosophy, Political Science, Sociology, Journalism, Security, Infosec, Economics, Game theory, Psychology

Table 4: An overview of challenges discussed in Section 4 (Sociotechnical Challenges). We stress that this overview is a highly condensed summary of the discussion contained within the section, and hence should not be considered a substitute for the complete reading of the corresponding sections.

| Challenge | TL;DR |
| --- | --- |
| Values to Be Encoded within LLMs Are Not Clear | Identifying the values that LLMs ought to be aligned with is a pivotal problem in the safety and alignment of LLMs. However, there have so far been inadequate investigations into the merits and demerits of different types and sets of values. Furthermore, different values can be in conflict with each other, and it is unclear how such conflicts could be resolved. Different 'lotteries' — methods lottery, profitability lottery, platformability lottery — might drastically influence the values that we might encode in LLMs. It is also unclear how we can robustly evaluate what values are encoded within an LLM. Finally, at a more meta-level, it is worth probing whether thinking about LLM alignment in terms of 'values' might be limiting. |

*Continued on the next page*

| Challenge | TL;DR |
| --- | --- |
| Dual-Use Capabilities Enable Malicious Use and Misuse of LLMs | The dual-use nature of many LLM capabilities creates risks from misuse of those capabilities. Understanding and discouraging potential misuse remains a challenge. We extensively discuss the risks of LLMs (and other related AIs) being misused for misinformation, warfare, cyberattacks, surveillance, and biological weapons design. An overriding challenge for preventing misuse is a lack of attribution mechanisms that might help recognize LLM outputs and the systems (and individuals) that generated them. |
| LLM-Systems Can Be Untrustworthy | LLMs remain prone to causing accidental harm to their users. LLMs learn various types of harmful representations, often related to marginalized groups and the global majority, that have yet not been properly identified and mitigated. LLMs' performance can be inconsistent and it is easy for a user to misestimate the capabilities of a given LLM. Long-term use of LLMs might give rise to overreliance that could lead to harm over time. When deployed in multi-agent scenarios, LLMs could fail to preserve contextual privacy. |
| Socioeconomic Impacts of LLM May Be Highly Disruptive | The impact of LLM-driven automation on society and the economy may turn out to be highly disruptive. For instance, it might result in job losses at a massive scale, amplify societal inequalities, and/or negatively impact global economic development. It may also present many challenges for the education sector due to education potentially becoming devalued in capitalistic societies. This may further have negative second-order impacts. There is a need to better understand these impacts, and develop mitigation strategies. |
| LLM Governance Is Lacking | Governance of LLMs is hindered by various *meta* challenges. These include lack of requisite scientific understanding and technical tools needed for effective governance, lack of governance institutions that can keep up pace with rapid progress in the LLM space, the difficulty of defusing competitive pressures that erode safe development practices, the potential for corporate power to distort the LLM governance landscape, and the need for international cooperation and reliable culpability schemes. Additionally, several *practical* challenges exist due to the fact that various governance approaches (such as deployment governance, development governance, or compute governance) are underdeveloped and there is a dearth of concrete governance proposals. |

## 4.1 Values to Be Encoded within LLMs Are Not Clear

Our discussion of alignment, so far, has focused on intent alignment, where we have taken the intent to be the intent of the LLM's developers. **In this section, we deviate from this assumption and raise the question of what, and whose, values an LLM should be aligned with** (Gabriel, 2020; Kasirzadeh and Gabriel, 2023). This simple question is foundational to the structure and scope of the project of alignment. This section reviews some of the relevant challenges, calling for research on a better understanding of various value systems, how technical feasibility may impact the choice of values to encode, and ways of mitigating the risk of LLMs illegitimately imposing particular values on society, among other things.

### 4.1.1 Justifying Value Choices for Alignment

The conventional discourse on LLM value alignment typically frames it in terms of the 3H framework of Askell et al. (2021). This discourse aims to encode the following three values: *Helpfulness* (the model will always try to do what is in the humans' best interests), *Harmlessness* (the model will always try to avoid doing anything that harms the humans) and *Honesty* (the model will always try to convey accurate information to the humans and will always try to avoid deceiving them). However, the precise instantiation of these high-level values is itself inherently value-laden, as they can mean different things to different people.

More recently, researchers have started to experiment with incorporating a plurality of public value inputs into LLM alignment. Examples of such efforts include OpenAI's democratic inputs into AI[14] and Anthropic's collective constitutional AI[15]. However, there is very little foundational work arguing *which* values are the right high-level values for LLM should alignment and under which circumstances, and which alternative types of values should be preferred over or in addition to the HHH framework, such as: truthfulness (Evans et al., 2021; Hilton, 2022), human rights (Prabhakaran et al., 2022a), cooperativeness (Dafoe et al., 2020; 2021), corrigibility (Soares et al., 2015), specific moral values (Hou and Green, 2023), or other pluralistic values (Sorensen et al., 2023; Durmus et al., 2023). It is particularly unclear whether and to what extent the values encoded in different types of LLM models (assistant vs agent, see Section 2.5) should differ; or how such values should depend on the context of use (e.g. medical assistant vs educational assistant) (Kasirzadeh and Gabriel, 2023), or the capabilities profile of an LLM. Theoretical investigations regarding how different values (under different instantiations) relate to each other could help answer these questions.

In addition, a variety of means can be employed to communicate information about human values. These include revealed preferences (i.e. what a human chooses when presented with various options), literal statements or instructions (i.e. the model literally does as asked), inferred intentions (i.e. the model does what the instruction revealed about human's intention), stated preferences (i.e. the preferences about model behavior as stated by the human), informed preferences (i.e. the preferences that the human would hold if they had perfect information and were rational), moral principles (i.e. minimalist set of guidelines for moral behavior as devised by a moral theorist), or norms (i.e. shared standards or guidelines that guide a group's behavior) (Gabriel, 2020; Ouyang et al., 2022; Bai et al., 2022a; Fränken et al., 2023). Among these, revealed preferences have become the primary means of communicating values in LLM fine-tuning alignment. However, we suspect that this choice is largely motivated by algorithmic convenience and the empirical success of reinforcement learning from human preferences (RLHF) (Christiano et al., 2017; Leike et al., 2018) (see Section 4.1.3). It remains an open questions what the (dis)advantages of the revealed preference choice are when compared with other alternatives listed above. Relatedly, it is important to understand whether and how alignment with respect to one

---

[14]https://openai.com/blog/democratic-inputs-to-ai
[15]https://www.anthropic.com/news/collective-constitutional-ai-aligning-a-language-model-with-public-input

category, such as principles, might supersede or interfere with alignment with respect to another, like preferences.

### 4.1.2 Managing Conflicts between Different Values

Human values can often come into conflict in three primary ways. Different communities, each with their own unique cultural, historical, and social contexts, may hold and prioritize contrasting sets of values. Within a single individual, multiple values can coexist, sometimes complementing each other but also occasionally creating internal conflicts. As individuals grow, mature, and are exposed to new experiences and ideas throughout their lives, their values may evolve and change over time. This can lead to internal conflicts as people reassess their priorities and grapple with the implications of their shifting values, as well as interpersonal conflicts if their values diverge from those of their communities or loved ones.

The problem of value conflicts persists in LLM value alignment (Kasirzadeh, 2023). For example, in the 3H framework of Askell et al. (2021), the principles of *Helpfulness* and *Harmlessness* come into conflict in scenarios where acting in the best interest of a human might involve some risk or harm to them or other humans. Researchers have started to investigate value conflicts and trade-offs in LLMs (Liu et al., 2024b). Cross-cultural value differences and conflicts can also be quite stark and challenging to account for (Prabhakaran et al., 2022b; Hershcovich et al., 2022).

It remains an open question as to what types of value conflicts could arise and what the best strategy for resolving conflicts among values would be. Including principles governing the resolution of competing values or making existing values or principles more specific may help minimize the risks of misalignment due to value conflicts (Keren et al., 2014; Kundu et al., 2023; Mechergui and Sreedharan, 2024). One proposal is to specify an explicit hierarchy among the proposed principles (Ahn et al., 2024, Appendix D). Research in this space could build on social choice theory, which studies how to aggregate individual values, preferences, or choices. (Shoham and Leyton-Brown, 2008; Davani et al., 2022, Chapter 9). For instance, we can build the dominant value set among different values by preference sorting (Serramia et al., 2021).

Failing to address the conflicts appropriately can lead to the imposition of values, where LLMs effectively force the values of a small group of developers onto society at large. This concerning phenomenon has been highlighted in recent research. For instance, Atari et al. (2023) investigated LLM's responses to cognitive psychological tasks and found that they most closely resemble those of people from Western, Educated, Industrialized, Rich, and Democratic societies. Similarly, Johnson et al. (2022) demonstrated that in cases of value conflicts, such as the issue of gun control, the 'values' expressed by GPT-3 align more closely with dominant US values than with those of other nations. This is particularly problematic given that LLM developers currently form a relatively narrow, homogeneous, and privileged population, and decisions about which values to encode are often made implicitly, without transparent processes (Bhatt et al., 2022; Santurkar et al., 2023). Inclusive participation (Shur-Ofry, 2023; Sorensen et al., 2024) and dialogue (Dobbe et al., 2021) play an indispensable role in addressing conflicts, though they are also prone to 'participation washing' (Sloane et al., 2022) where the appearance of inclusive practices is maintained without genuine commitment to incorporating diverse perspectives in addressing conflicts.

Choice mechanisms that allow the LLM values to be selected in a systematic and contextual way (Gabriel, 2020; Birhane et al., 2022; Kasirzadeh and Gabriel, 2023) could help addressing conflicts. Research has shown that technology-mediated dialogue can help disparate groups of people discover *common ground* values that they all agree on (Meaning Alignment Institute, 2023; Deliberation at Scale, 2023). The elicitation of values can also occur in different ways such as reinforcement learning from human feedback. An alternative approach could be to extract common values by comparing cross-cultural human judgments, obtained through human values surveys such as World Values Survey

(Haerpfer et al., 2022) or Schwartz Value Survey (Schwartz, 1992; 1994). We might also elicit justice-related human values via philosophical setup of the Veil-of-Ignorance (Weidinger et al., 2023b).

Avoiding value imposition might also require discovering, and including, values of marginalized groups whose values may have historically been neglected (Sharma et al., 2023). Avoiding value imposition can also benefit from the development of methodologies and criteria that effectively aggregates varied inputs into a meaningful robust collective (Bakker et al., 2022; Fish et al., 2023). Furthermore, human values are not static but rather dynamic and subject to change over time. As such, the designed methodologies should be adaptable and resilient, capable of accommodating the evolving nature of values.

### 4.1.3 'Lotteries' May Bias the Values That We Encode

Hooker (2021) introduced the idea of *hardware lottery* to describe the situation in which a research idea wins over other research ideas, not because it is intrinsically superior, but because it is favored by the existing hardware. A similar *technical lottery* may also exist for the values that we may want to encode in our models; the values that we encode in our models may not necessarily be the ones that we most want to encode in our models, but the ones that are technically feasible to encode (Carissimo and Korecki, 2023). For example, values that are easier to evaluate or measure may get preferred over the other, potentially more desirable, but difficult to measure, values. A pertinent example in this regard is the 3H framework of Askell et al. (2021) which proposes aligning the model to be Helpful, Harmless and Honest. However, in practice, the focus is generally restrictively placed on ensuring that models are helpful and harmless (Bai et al., 2022b) as evaluating model honesty is more costly and less technically tractable.

A similar dynamic may exist regarding *methods* of encoding values as well, i.e. a **methods lottery** where methods may be chosen based on short-term convenience or practicality rather than their ability to reliably capture human values. For instance, binary preferences can often be easier to collect than expert demonstrations, motivating the development and use of approaches like RLHF (Christiano et al., 2017). Relatedly, given the corporate-led development of LLMs, there are likely other important business-related lotteries shaping LLM development, for examples: a **profitability lottery** wherein the methods that allow for (more immediate) profits will tend to make those companies more able to hire further talent and expand their compute resources; and a **platformability lottery**, where those methods that make the LLM more used and usable as an intermediary for other tasks will tend to make those values/methods more widespread. Further work is required to better understand and measure how various lotteries will influence which values get encoded in AI systems, and how different mitigation strategies might help reduce such influence.

### 4.1.4 How Can We Robustly Evaluate Which Values an LLM Encodes?

A common strategy used in prior work to probe the values of LLMs is to evaluate the LLM behavior on data designed to measure moral, social, or political behaviors and make inferences about the (unobserved) values being used by the LLM to perform its decision-making (Hendrycks et al., 2020b; Sorensen et al., 2023). Pan et al. (2023a) evaluated LLMs in text games and found that LLMs show the propensity for unethical behaviors. Scherrer et al. (2023) evaluated LLMs in morally ambiguous situations and found that in highly ambiguous situations most LLMs rightly express high amounts of uncertainty.

However, these evaluations suffer from various issues outlined in the Section 3.3, e.g. prompt-sensitivity and systematic biases. The future work in this vein should avoid, or account for, these issues. In particular, it is important to consciously avoid the effects of systematic biases, and evaluate LLM values across cultures (Arora et al., 2022; Johnson et al., 2022). Furthermore, evaluations based on realistic applications (e.g. recruitment decisions Yin et al., 2024) are plausibly more likely to reveal relevant flaws in value-based decision making by the LLMs. There is also a need to develop evaluation methodologies that can help us better understand the extent to which LLMs understand the human

values of interest and will reliably accord with them (Talat et al., 2021; Zhang et al., 2023d), and are not just mimicking them (Simmons, 2022). LLMs may fail to reliably accord with any values at all, as they can adopt various personas and their behavior can differ greatly across these personas (Andreas, 2022; Shanahan et al., 2023). Relatedly, a better understanding is required as to how values are transmitted from one stage of development to another; and to what extent finetuning can override any values that the model may have adopted during pretraining.

### 4.1.5  Is 'Value Alignment' the Right Framework?

The classical approach to ensuring that AI benefits all of humanity has been framed in terms of resolving the 'value alignment' problem (Russell, 2019; Leike, 2022b). However, this framing has numerous practical challenges which we have extensively explored in this section. At a more meta-level, the concept of 'value alignment' raises fundamental questions about whether values exist, what they are, whether they are universal, or how they relate to prescribed and proscribed actions. These questions have been subject to intense philosophical debate (Brogan, 1952; Kingma and Banner, 2014; Schroeder, 2021; Polak and Rohs, 2023; Kaiser, 2024). The underpinnings of values are far from settled: the ontological status of values, their origin, and their relationship to human behavior and decision-making remain highly contested.

Moreover, the connection between abstract values and their concrete instantiation is complex and often context-dependent (Kirk et al., 2023b), making it challenging to translate values into specific rules or constraints for AI systems. These issues cast doubt on the feasibility and specificity of the 'value alignment' framing as the right approach to guide the design of AI technologies that are beneficial to all of humanity and free of harm. Indeed, there is a risk that overly focusing on a limited scope of 'value alignment' creates a false promise that a technological solution exists for a problem that might be inherently multi-dimensional and requires non-technical approaches to be addressed. The 'value alignment' framing also appears to have the drawback that it views the use of LLM or AI-based systems in isolation; when in fact, these systems are likely to get embedded within human society, and might even become tools to exercise power over humans and mediate human agency. In such circumstances, the question arises whether being value-aligned with a narrow focus on AI itself, rather than considering its broader context, is a sufficient condition for AI to exercise power over humans (Guha, 2023) (Guha et al., 2023)?

In summary, there is a pressing need to thoroughly examine the scope and limitations of the 'value alignment' framing and explore alternative or complementary framings that might better address the challenges of ensuring AI benefits humanity broadly. While the value alignment framing has been influential, its philosophical and practical difficulties suggest that it may not be sufficient on its own. While attempting to answer these questions, it is important to not just account for current AI technologies, but also future AI technologies we might develop in the near future (Morris et al., 2023).

> Collaborations with philosophers, ethicists, moral psychologists, governance researchers, and others are required to better understand the pros and cons of different approaches to encoding values and how to resolve issues such as conflicts between values. At the same time, what values we will encode within our models may be heavily biased by the tractability of different approaches to encoding values. Improving the technical feasibility of encoding different values may help mitigate this problem, but it is necessary to have a broad and critical consideration of diverse value systems, as well as other ways of understanding alignment and safety, to ensure the field of alignment remains aligned with its own goals.
>
> **132.** What justifies choosing one set of values (e.g. helpfulness, harmlessness, honesty) over other sets of values?

**133.** How does the type of a system (e.g. assistant vs. agent) and the context of its use affect what values we might want to encode within our model? ↩

**134.** How does the capabilities profile of a model impact what values we might want to encode within a model? Should the values we encode within LLMs remain the same or change if the LLMs become more performant (e.g. due to scaling)? ↩

**135.** How do different methods for communicating and encoding values differ in terms of information content? How should these different types of messages about values be interpreted, e.g. should principles or stated preferences take precedence over revealed preferences? ↩

**136.** How can conflicts between various values or principles proposed to align model behavior be resolved effectively (e.g. harmlessness and helpfulness in 3H framework)? ↩

**137.** How can we design methods to balance conflicting values appropriately or enable (groups of) humans to resolve the conflicts between their values? ↩

**138.** How do we mitigate the risk of value imposition? Can we design governance mechanisms that allow LLMs' values to be chosen in a systematically fair and just way? ↩

**139.** How are we to account for changes in values over time? ↩

**140.** To what extent will the 'technical lottery' play a role in what values we encode in our models? For values that may be technically infeasible to encode, can we develop technically feasible robust proxies that we could use instead? ↩

**141.** How can we robustly evaluate what values are encoded within a model? ↩

**142.** How can we determine whether a model understands the encoded values or is only mimicking them? Relatedly, to what extent can we claim that an LLM has values, given an LLM is perhaps more like a superposition of various personas with varying characteristics? ↩

**143.** How are values transmitted from one stage of development to another? ↩

**144.** What are the limitations of framing the design of AI technologies that broadly benefit humanity in terms of 'value alignment' with humanity? Can we develop alternative or complementary framings that might help address those limitations?

## 4.2 Dual-Use Capabilities Enable Malicious Use and Misuse of LLMs

Like all technologies, LLMs have the possibility for misuse by malicious actors. Malicious use of dual-use capabilities of AI is a recurring concern within literature (Brundage et al., 2018; Hendrycks et al., 2023; Mozes et al., 2023). However, there exists a significant gap from these relatively high-level articulations of concern to rigorous research on the topic, resulting in a lack of nuanced and context-specific understanding of the risks associated with dual-use capabilities. This deficiency in understanding and practicality is deeply concerning as it hampers the development of effective mitigation strategies. While we primarily focus on intentional misuse, we note that as LLMs become more widespread, easy-to-use, and general-purpose, the potential for accidental misuse and other ethically grey uses is also amplified. **This section reviews several plausible malicious use cases of present or future LLMs and calls for further research to improve our understanding of how LLMs might be misused.** The improved understanding could inform prioritization of concerns and the development of effective mitigation strategies. Although we focus on LLMs, this is simply due to the focus of our agenda; a consideration of which types of AI systems have the greatest potential for harmful misuse is out of our scope.

### 4.2.1 Misinformation and Manipulation

Recent studies have demonstrated that LLMs can be exploited to craft deceptive narratives with levels of persuasiveness similar to human-generated content (Pan et al., 2023b; Spitale et al., 2023), to fabricate fake news (Zellers et al., 2019; Zhou et al., 2023f), and to devise automated influence operations

aimed at manipulating the perspectives of targeted audiences (Goldstein et al., 2023). LLMs have also been found to be used in malicious social botnets (Yang and Menczer, 2023), powering automated accounts used to disseminate coordinated messages. More broadly, the use of LLMs for the deliberate generation of misleading information could significantly lower the barrier for propaganda and manipulation (Aharoni et al., 2024), as LLMs can generate highly credible misinformation with significant cost-savings compared to human authorship (Musser, 2023), while achieving considerable scale and speed of content generation (Buchanan et al., 2021; Goldstein et al., 2023).

Furthermore, an area of particular concern regards the degree of personalisation that LLMs can achieve in the production of misleading content, whether wholly false or true but presented in a misleading way. It is already well-documented that LLMs tend to be sycophantic, that is, selectively present content according to perceived user desires (Perez et al., 2022a); it is not hard to imagine this capability being exploited for malicious purposes. By tailoring content to specific demographics or individual profiles, these models can facilitate the creation of hyper-targeted misinformation (Bagdasaryan and Shmatikov, 2022; Ferrara, 2023). This feature of LLMs raises alarming possibilities for manipulating belief systems and public opinion at scale, potentially exacerbating societal divisions (Kirk et al., 2023c) and undermining trust in trustworthy information sources (Weidinger et al., 2022).

Additionally, while the risks discussed largely concern society-wide implications of LLM-powered misinformation, these models can also exacerbate harm to individuals. For example, LLMs can be leveraged to create highly realistic multimodal deepfakes, which include audio, video, and photographic content. These deepfakes, often indistinguishable from authentic content, can be used for a range of individual-level harms, such as the production of falsified sexual images and the discreditation of individuals (Chesney and Citron, 2019). Such content can do harm even if it is known to be fake (Burga, 2024).

Further empirical and theoretical research is needed to assess the scale and likelihood of the use and effectiveness of LLMs for misinformation generation and propagation. There is also a need for research into developing tools and mechanisms to combat misinformation. One successful way of doing so is designing robust community-based fact-checking tooling (Pröllochs, 2022). LLMs themselves could potentially form part of the solution; for example, by identifying *why* a particular piece of information is false, surfacing relevant sources to substantiate their claim and providing desirable alternative explanations (Hu et al., 2023c; Chen and Shu, 2023).

### 4.2.2 Cybersecurity

LLMs may exacerbate cybersecurity risks in various ways (Newman, 2024). Firstly, LLMs may significantly amplify the effectiveness of deceptive operations aimed at tricking people into disclosing sensitive information or granting adversary access to critical resources. For example, LLMs might prove highly effective at crafting personalized phishing emails or messages at scale that may be harder for an average user to recognize as phishing attempts (Karanjai, 2022; Hazell, 2023). In addition to being directly harmful to the targeted individual, such 'social engineering' attacks are often the base of larger hacking operations (Plachkinova and Maurer, 2018; Salahdine and Kaabouch, 2019). However, the precise impact of LLMs on the likelihood of successful social engineering attacks is currently not clear. To develop a better understanding of this risk, researchers could perform user studies involving white-hat hackers to understand how an actual hacker might benefit from using an LLM in her hacking attempts. This could inform technical or sociotechnical mitigation strategies, such as training an LLM to recognize when it is being misused for crafting phishing emails etc. and to decline such requests. However, the technical problem of jailbreaking would remain an issue here (c.f. Section 3.5).

Secondly, coding capabilities of LLMs could be used for malicious purposes (Checkpoint Research, 2022). This may either be done through using off-the-shelf LLMs or through training or fine-tuning LLMs specifically for this purpose (Checkpoint Research, 2023; Erzberger, 2023). This may include using code-inspection capabilities of LLMs to find software vulnerabilities, and code-writing capabil-

ities of LLMs to create novel malware and exploits. However, cybersecurity teams may also leverage LLMs in a similar fashion to preemptively identify and remedy software vulnerabilities and strengthen cybersecurity in other ways (Aghaei et al., 2022; Ferrag et al., 2023). Consequently, the net impact of LLMs on cybersecurity is currently not clear and deserves further study (Hendrycks et al., 2021a). In addition to developing a more calibrated understanding of how coding capabilities of LLMs may be used in malicious ways, researchers may focus on developing LLM-based cybersecurity tools that may be helpful to cybersecurity professionals.

One other way in which coding capabilities of LLMs could prove harmful is if they lower the barriers for staging a successful cyberattack (Brundage et al., 2018). Current evidence is mixed in this regard indicating that while LLMs can be used to create novel attacks, this generally requires some know-how on the part of the user as well (Checkpoint Research, 2023; Carlini, 2023b). However, it is plausible that the improvements in coding capabilities of LLMs could eliminate this need for expert knowledge in the future. This risk ought to be monitored closely. One way to do so could be to develop benchmarks focused on evaluating *autonomous* code-writing capabilities of a given LLM (Deng et al., 2023b).

Cybersecurity risks can also increase due to the collective resources available to multi-agent systems powered by LLMs. Such systems could represent a risk similar in scale to botnets, with a large number of coordinated agents working together (Sun et al., 2023). However, the generative capabilities and possible emergent abilities of these systems at scale extend the potential impact beyond traditional Distributed Denial of Service (DDoS) attacks. For instance, multi-agent systems could be used for targeted vulnerability analysis and exploitation over a range of systems in a coordinated, fault tolerant manner (Hendrycks et al., 2021a). This could facilitate vulnerability chaining across systems and networks, enabling multi-stage attacks that are inherently more difficult to mitigate (Roytman and Bellis, 2023). In addition, multi-vector attacks could shift the attack-defense balance as standard filtering/monitoring techniques may prove ineffective against generative agents actively concealing their distributed activities using advanced forms of steganography (de Witt et al., 2022) or adversarial attacks of bounded information-theoretic detectability (Franzmeyer et al., 2023). Similar approaches could also accelerate the forensics and post-exploitation process to superhuman speeds and efficiency (Xu et al., 2024b). On the defense side, LLMs could help, for example, through automated analysis of security logs (Boffa et al., 2022). How the offense-defense balance will shift is an open problem, and much work remains to be done to safeguard cybersecurity infrastructure from scaled-up threats (Hendrycks et al., 2023) and within the broader context of *multi-agent security* (de Witt et al., 2023).

Lastly, there is increasing evidence that LLMs can be used to craft jailbreaks which can be used on other LLMs and other instances of the same LLM (Chao et al., 2023; Mehrabi et al., 2023; Shah et al., 2023). This poses a risk to the security of LLMs and may result in a dangerous dynamic where improvement in a (closed-source) LLM's capabilities means it can generate more sophisticated jailbreaks (which could be used to jailbreak another instance of the same LLM). There is a need to understand how this dynamic may play out, and what technical and sociotechnical interventions can be done to mitigate this risk.

### 4.2.3   Surveillance and Censorship

Content moderation has emerged as one of the key use-cases of LLMs (Weng et al., 2023), indicating the potential of LLMs for surveillance and censorship as well (Edwards, 2023). Surveillance and censorship are one of the primary tools employed by governments with dictatorial tendencies to suppress opposing political and social voices. These censorship measures, however, are often quite crude and can be escaped with little ingenuity. For example, manual supervision of the content (e.g. newspaper articles) is error-prone and can be circumvented by simple tactics, such as using clever phrasing that does not appear critical at the surface or hiding critical content within the margins of the articles which are likely to be overlooked (Hem, 2014). Similarly, automated tools currently used for censorship of text-based communication at scale are quite primitive and primarily based on keyword-matching (Knockel et al., 2020). However, LLMs could enable significantly more sophisticated surveillance and

censorship operations at scale (Feldstein, 2019). Multimodal-LLMs or LLMs combined with speech-to-text technologies could be used for surveilling and censoring other forms of communication as well, e.g. phone calls and video messages (Whittaker, 2019). This may collectively contribute towards the worsening of personal liberties and the heightening of state oppression across the world. Examples have been documented already, for instance in calling for violence and silencing of political dissidents (Aziz, 2020), and suppression of Palestinian social media accounts (Zahzah, 2021).

Sociotechnical research could help better understand the various ways in which surveillance and censorship may negatively impact the free thought and integrity of democratic institutions (Richards, 2012). It is also important for the research and academic community to be proactive in this regard and ensure that their designed models are not misused for surveillance and censorship purposes (Kalluri et al., 2023). The developers of LLM-based technologies, and civil society at large, ought to resist attempts at creating laws that provide legal covers to such surveillance efforts, with security concerns used as the ostensible reason (Ellis-Petersen, 2021). A possible technological mitigation against censorship and technological lock-in could be the use of perfectly secure steganography (Schroeder de Witt et al., 2023, iMEC), which allows covert communications and information hiding in the outputs of LLMs under information-theoretic undetectability.

### 4.2.4 Warfare and Physical Harm

The use of AI in warfare is highly alarming and may pose dangers to human safety (Hendrycks et al., 2023). Autonomous drone warfare is being aggressively pursued as a tactic in the current war in Ukraine (Meaker, 2023), and may already have been used on human targets (Hambling, 2023). The use of AI-based facial recognition has been documented in the targeting of Palestinians in Gaza (International, 2023). LLMs have already been productized in limited ways for the purposes of warfare planning (Tarantola, 2023). Furthermore, active research is being carried out to develop multimodal-LLMs that can act as 'brains' for general-purpose robots (Ahn et al., 2022; 2024). Due to the 'general-purpose' nature of such advances, it will likely be cost-effective and practical to adapt them for creating more advanced autonomous weapons. Development of autonomous weapons has been extensively criticized and cautioned against in prior literature (Sauer and Schörnig, 2012; Sharkey, 2016; Scharre, 2016; Roff and Moyes, 2016), and identified as a source of catastrophic risk (Hendrycks et al., 2023). These harms have also been cautioned against by technical machine learning researchers in various open letters (Future of Life Institute, 2016; Foerster, 2020). There exist voluntary pledges by AI companies to not weaponize their technologies (Boston Dynamics, 2022), however, past evidence indicates that such voluntary pledges can be side-stepped when convenient as they do not pose any legally binding requirements (Christie, 2018; Biddle, 2024). Hence, LLM-based autonomous weapons may soon pose major safety risks, absent international agreements and national legislation prohibiting their development and use. Besides political challenges, there are technical challenges around monitoring and enforcement. Developing or increasing the availability of autonomous drone weapons and facial recognition targeting may also increase the risk of targeted mass killings by non-state actors.

### 4.2.5 Hazardous Biological and Chemical Technologies

AI systems such as LLMs, chemical LLMs (Skinnider et al., 2021; Moret et al., 2023), and other LLM-based biological design tools might soon facilitate the production of bioweapons, chemical weapons, and other hazardous technologies. In particular, LLMs might enable actors with less expertise to more easily synthesize dangerous pathogens, while customized chemical and biological design tools might be more concerning in terms of expanding the capabilities of sophisticated actors (e.g. states) (Sandbrink, 2023). Gopal et al. (2023) and Soice et al. (2023) demonstrated that people with little background could use LLMs to help make progress towards developing pathogens such as the 1918 pandemic influenza. However, recent studies suggest that current LLMs are *not* more helpful than internet search in this regard (Mouton et al., 2024; Patwardhan et al., 2024). On the other hand, search engines offer more practical affordances for removing content vs. e.g. needing to retrain an LLM to ensure the content is

not influencing future responses. Furthermore, (future) LLM-based technologies could develop strong reasoning capabilities that might help them make novel discoveries (Romera-Paredes et al., 2024) — this potential to make novel discoveries could also transfer to hazardous chemical and biological technologies (Moret et al., 2023), potentially, resulting in technology designs that might be more challenging to guard against via supply-chain monitoring. There is a need for more research to characterize the potential "uplift" that (current *and* future) LLM-based technologies may provide, as the current studies involved small sample sizes, only used 'vanilla' LLMs and none of them produced conclusive results (Marcus, 2024). The realism of these studies can also be increased by moving them into actual 'wet' laboratories and would benefit from the involvement of experts in clinical trials or psychology research.

Furthermore, there is a need for ongoing monitoring to track how relevant capabilities are evolving with scale (c.f. Section 2.3). Doing so well may require postulating and refining explicit threat models and defining clear thresholds for action in advance; such work could also involve national security experts. Machine learning researchers should also work with professionals who study terrorism and mass killings to better understand whether and how LLMs might significantly contribute to such risks; this could involve understanding the social factors underlying such attacks, how the resources currently available online are – or might be – used by such attackers, and what currently limits the rate and severity of such attacks. Finally, we note that most of the contemporary work on biological and chemical risks from LLM-based technologies is concentrated on risks from non-state actors; there is also a need to better understand how states might misuse LLM-based technologies in this regard (Yassif et al., 2023).

### 4.2.6 Domain-Specific Misuses

Improvements in LLMs may exert greater pressure to apply LLMs to various domains, such as health and education (Eloundou et al., 2023). Crude efforts to use LLMs in such domains, however, may incur harm and should be discouraged strongly. In particular, it is important to guard against different ways in which LLMs may be misused within any domain. One famous episode of misuse within the health sector is a mental health non-profit *experimenting* LLM-based therapy on its users without their informed consent (Xiang, 2023a). Within the education sector, LLMs may be misused in various ways that might impact student learning; e.g. as cheating accessory by the students or as (low quality) evaluator of student's work by the instructors (Cotton et al., 2023). Recent findings in moral psychology also suggest that LLMs can generate moral evaluations that people perceive as superior to human judgments; these could be misused to create compelling yet harmful moral guidance (Aharoni et al., 2024). Similar risks of misuse may exist in other domains as well.

There is a need for research to better understand how LLMs might be misused across different domains. Interviews with domain experts and case studies of early deployments across different sectors may help in this regard. Domain-specific standards and regulations could be established to prevent these risks from materializing (Organization, 2021). One particular sociotechnical challenge is to effectively identify emerging use cases of LLMs, e.g. through surveys and/or in partnership with LLM deployers, who could monitor use patterns. Several factors make this a non-trivial challenge, even for LLM deployers; for instance, users may disguise queries or user calls may be spread across various LLM deployers in a way that might mask the real use (Glukhov et al., 2023).

### 4.2.7 Mechanisms for Detecting and Attributing LLM Outputs Are Lacking

One overriding challenge in preventing malicious use of LLMs is the difficulty of recognizing outputs from LLMs and attributing them correctly to particular systems and/or users. Attribution facilitates accountability and enforcement, which may help disincentivize many forms of malicious use. Techniques such as watermarking, which embeds detectable patterns in AI-generated content, could be helpful (Kirchenbauer et al., 2023), but their effectiveness is currently unclear (Zhang et al., 2023e; Huang et al., 2023e; Hu et al., 2023d) and more work is needed. Increasing access to LLMs (especially open source models) might make attribution more challenging (Augenstein et al., 2023; Seger et al., 2023). In particular, even if an open-source model watermarks its outputs, it might be easy to modify it to

not do so. This motivates the challenge of attributing outputs to particular models in the absence of (deliberately injected) watermarks. Machine learning researchers could work with cyber security experts and other professionals to understand the gaps that LLMs will create in current attribution practice for the particular domains discussed above.

There is a strong risk that dual-use capabilities of LLMs may be exploited for malicious purposes. LLMs may be misused towards generating targeted misinformation at an unprecedented scale. The coding capabilities of LLMs may be misused by malicious actors to mount cyberattacks with greater sophistication and at higher frequencies. LLMs are quite effective as content moderation tools; this may mean that LLMs get adopted to enact mass surveillance and censorship. LLMs may be used to power autonomous weapons, creating a possibility of physical harm from them. Other misuses of LLMs may occur as LLMs are applied to various domains. Active research is needed to better understand these risks and to create effective mitigation strategies.

145. Can we develop a calibrated understanding of how current and future LLMs could be used to scale up and amplify misinformation campaigns? What level of human expertise is required to effectively use LLMs to generate targeted misinformation? ↩

146. Can we develop reliable techniques to attribute LLM-generated content, helping track its spread? Can watermarking be an effective measure given the growing availability of openly accessible LLMs? ↩

147. How can individuals be protected against harms caused by AI-assisted deepfakes? ↩

148. What measures can be taken to prevent LLMs from producing sophisticated misinformation, while simultaneously enhancing their capacity to identify and mitigate misinformation? ↩

149. Can we develop effective tooling and mechanisms to combat misinformation on online platforms? How can LLMs themselves be applied to detect and intervene against LLM-generated misinformation? ↩

150. How may LLMs contribute to scaling and personalization of social engineering-based cyberattacks? ↩

151. To what extent do LLMs reduce the threshold of technical expertise required for executing a successful cyberattack? ↩

152. Do advances in capabilities of LLMs cause LLMs to become better at crafting jailbreaking attacks? If so, how can the safety of LLMs from jailbreaking attacks (designed by other LLMs) be assured? ↩

153. How effective are LLMs at surveillance and censorship? How may LLMs, on their own or in combination with other technologies (e.g. speech-to-text softwares), contribute to the expansion and sophistication of current surveillance operations? How can we limit the use of LLMs for surveillance? ↩

154. How can the military applications of LLMs, especially LLM-powered autonomous weapons, be regulated? There appears to be a mass consensus within the machine learning community that LLMs, and other AIs, should not be weaponized; how can this consensus be leveraged to create legislation pressure to outlaw autonomous weapons development across the world? ↩

155. How might current, or future, LLM-based technologies (including chemical LLMs and specialized biological design tools based on LLMs) be misused in the design of hazardous biological and chemical technologies? ↩

156. Can we identify how LLMs may get misused across various domains, such as health and education? What regulations are required to prevent such misuses? In general, how can we best understand LLM use cases and identify those with significant misuse potential? ↩

157. Can we design robust watermarking mechanisms that may help us identify LLM-generated content? ↩

158. What mechanisms can be used to determine attribution for content generated using openly available models for which watermarking may not be an appropriate solution (as it could be easily undone)? ↩

## 4.3 LLM-Systems Can Be Untrustworthy

A key desideratum for an LLM from a user's perspective is 'trustworthiness', i.e. assurance of reliability and consistent performance, and absence of any accidental harm caused by the technology to the user.[16] **Providing assurance that an LLM-based system will not cause *accidental* harm remains a major open challenge.** Harms may either occur directly due to the flawed nature of LLMs, e.g. an LLM generating toxic language or behaving inappropriately in some other ways, or may occur due to improper usage by a user, e.g. automation bias due to a user's overreliance on LLM. We overview several challenges in this section that can potentially undermine a user's trust in LLMs. The technical flaws underlying these challenges in many cases appear difficult to address; at least in the immediate term. Hence, technical research alone may be insufficient to address these challenges, and sociotechnical interventions may be required to assure that LLM assistants do not incur accidental harm to the users.

### 4.3.1 Harms of Representation and Other Biases

A pretrained LLM generally has many of the stereotypical biases commonly present in the human society (Touvron et al., 2023). This makes it difficult for users to trust that LLMs will work well for them and not produce unfair or biased responses. Appropriate finetuning can effectively limit the bias displayed in LLM outputs in a variety of situations, e.g. when models are explicitly prompted with stereotypes (Wang et al., 2023k), but it does not 'solve' the problem. Even after finetuning, biases often resurface when deliberately elicited (Wang et al., 2023k), or under novel scenarios, e.g. in writing reference letters (Wan et al., 2023a), generating synthetic training data (Yu et al., 2023c), screening resumes (Yin et al., 2024) or when used as LLM-agents (Pan et al., 2024). The biases outputs are often much more prominent in low-resource languages (Yong et al., 2023) and in dialects used by marginalized groups (Hofmann et al., 2024). There is a need for research to develop better and more comprehensive tools for the detection of bias, toxicity (Wen et al., 2023; Wang and Chang, 2022), and other kinds of inappropriate behaviors. The current tools primarily focus on detecting content that is *explicitly* toxic and offensive, however, the issue of bias goes beyond commonly studied subjects of biases, such as race and gender (Hofmann et al., 2024). For instance, depending on the finetuning data, LLMs may also develop a bias towards particular political ideologies (Rutinowski et al., 2023). These biases are still relatively poorly understood, and there is a need for more extensive evaluations of current LLMs in novel scenarios to better understand the propensity of LLMs to enact such biases. There is also a need for a thorough evaluation of how the global south is represented within these models, and what sort of harmful representations of the global south are reinforced by LLMs (Qadri et al., 2023; Jha et al., 2024).

### 4.3.2 Inconsistent Performance across and within Domains

Estimating true capabilities of an LLM is a difficult task (c.f. Section 3.3), especially for naive users unfamiliar with the brittle nature of machine learning technologies. Exaggeration of model capabilities by the developers (Lambert, 2023; Blair-Stanek et al., 2023), and issues such as task-contamination (Roberts et al., 2023b), underrepresentation of tasks or domains (Wu et al., 2023a; McCoy et al., 2023), and prompt-sensitivity (Anthropic, 2023d) may cause a user to misestimate the true capabilities of a model. This lack of reliability can undermine user trust or cause harm if a user bases their decision on

---

[16]Note that within the machine learning literature, the term 'trustworthiness' carries various other meanings as well. The scope of our discussion is limited to our definition given here

incorrect or misleading information provided by an LLM. A famous example of this is the US lawyer who cited a fake case, hallucinated by ChatGPT, in a legal brief filed in a US court (Merken, 2023). Technical solutions could involve improving the reliability of the LLMs performance (e.g. using retrieval augmented generation to minimize hallucinations) or providing reliable uncertainty estimates alongside LLM responses (Fadeeva et al., 2023; Kuhn et al., 2023). However, technical solutions may not be developed quickly, if at all. Hence, there is a need to understand how the reliability of LLMs can be improved through extrinsic measures. Towards this goal, it will be useful to understand how reliable a median user perceives an LLM to be, to what extent they are able to reliably forecast on what problems a particular LLM will fail (Carlini, 2023a) and how this ability to determine the reliability of an LLM's performance evolves as the user interacts further with the LLM. This may be done through user studies and interviews with various different types of users. It may be difficult to draw very general conclusions, and more productive in many instances to focus on trustworthiness for particular (e.g. somewhat narrow) use cases or domains. Such research could then provide insights into the efficacy of different extrinsic measures that could be taken to help users use the LLMs safely e.g. providing appropriate education and disclaimers to the user before they begin interacting with an LLM.

### 4.3.3   Overreliance

If a user begins to excessively trust an LLM, this may cause them to develop an overreliance on the LLM. Overreliance can result in automation bias (Kupfer et al., 2023), and can cause errors of omission (user choosing not to verify the validity of a response) and errors of commission (user believing and acting on the basis of the LLM's response, even if it contradicts their own knowledge) (Skitka et al., 1999). It can be particularly dangerous in domains where the user may lack relevant expertise to robustly scrutinize the LLM responses. This is particularly a source of risk for LLMs because LLMs can often generate plausible, yet incorrect or unfaithful, rationalizations of their actions (c.f. Section 3.4.10), which can mistakenly cause the user to develop the belief that LLM has the relevant expertise and has provided a valid response. A common tactic used to prevent harms that might arise from overreliance is to include disclaimers alongside model output that could act as triggers for the user to externally validate the model response. However, users can get desensitized to such disclaimers over time (OpenAI, 2023b). Better interface design (e.g. using distinct colors for distinct grades of disclaimers) could help avoid, or limit, such desensitization. User studies could be done with different user types to better understand the details of how overreliance might manifest, and what mitigation strategies may be most effective against it. The use of LLMs as coding assistants by developers could be used to study over-reliance as coding tasks are often well-posed and can have varying levels of complexity as required.

There is also a need to better understand the related risks that might arise due to consistent and prolonged usage of LLM by a user in a particular domain. Outsourcing certain types of cognitive tasks to LLMs, e.g. writing tasks, could impair corresponding skills among LLM users. This is particularly a risk for the use of LLMs in education where excessive usage of LLM may cause students to develop an unnecessary, and unwanted, dependency on LLMs. Additionally, prior work has shown that humans can inherit biases from AI systems, and that these negative effects of AI technology do not naturally go away even when the biased AI systems are removed (Vicente and Matute, 2023; Kidd and Birhane, 2023). Further research is required to better understand these risks; how they might materialize, and what can be done to mitigate them?

### 4.3.4   Contextual Privacy Preservation

As LLMs proliferate through society, this will inevitably result in LLMs simultaneously interacting with multiple parties (see Section 2.6). In such scenarios, assurance of privacy goes significantly beyond assuring that the outputs of the LLMs do not contain personally identifiable information (Nissenbaum, 2020), and requires that LLMs do not leak data or information shared by one actor to another needlessly. The notion of privacy is not well-formalized in such contexts currently and there is a need for work on defining and operationalizing it in an appropriate way. In the absence of a formal definition, a useful

goal could be to assure that in any context LLMs do not share information given by one party with some other party, unless a human would do so as well. Current LLMs do not provide this assurance and often leak private information provided by one party to another when a human would not do so — and common tricks, e.g. prompt-engineering or output filtering do not improve alignment between the behavior of LLMs and human (Mireshghallah et al., 2023). This hints at some of the fundamental limitations in preserving privacy in multi-agent contexts.

> Trustworthiness, defined as the assurance that users will not experience accidental harm when using an LLM, is critical for LLM alignment and safety. Among other things, trustworthiness requires ensuring that users can use LLMs safely despite the occasional unreliability of their outputs; preventing users from developing an overreliance on LLMs; mitigating harm resulting from biased representations of societal groups learned by LLMs; ensuring appropriate behavior across all contexts; preventing the generation of toxic and offensive content by LLMs; and reliably preserving user privacy across various contexts.
>
> **159.** What sort of societal biases are present within LLMs? Can evaluations on complex, real-world tasks uncover those biases? ↩
>
> **160.** To what extent is the global south represented within LLMs? What harmful representations of the global south are present within the LLMs? ↩
>
> **161.** Can we develop better and more comprehensive tools for implicit and explicit toxicity and offensive language detection? ↩
>
> **162.** To what extent are LLM users able to accurately perceive the reliability of LLM outputs? Does a user's ability to assess the reliability of an LLM's response improve with continued interaction? ↩
>
> **163.** What extrinsic measures can be implemented to ensure that LLM users utilize the technology safely, especially considering the potential unreliability of LLM responses? ↩
>
> **164.** How can we mitigate the potential harms arising from overreliance on LLMs? How can we prevent users from becoming desensitized to disclaimers about LLMs' limitations? ↩
>
> **165.** How can we make LLMs understand the sensitivity of information in a given context, and preserve contextual privacy? ↩

## 4.4 Socioeconomic Impacts of LLM May Be Highly Disruptive

The rapid evolution of LLMs brings significant socioeconomic opportunities and challenges, impacting the workforce, income inequality, education, and global economic development. Many of these challenges are systemic in nature, constituting what economists refer to as general equilibrium effects. These challenges do not arise directly from LLMs causing harm to users but rather from their indirect effects on the socioeconomic equilibrium. For instance, automation may reduce labor demand, leading to wage declines and subsequent harm to workers. Consequently, addressing these challenges may necessitate system-wide policy interventions. **To effectively implement such interventions, a thorough understanding of these challenges and potential mitigation strategies is imperative**. In this section, we explore some of these challenges and pose pertinent research questions that warrant investigation.

### 4.4.1 Effects on the Workforce

The effects of integration of LLMs into various industries and workflows on the workforce are likely to be significant, and pose a great socioeconomic challenge. Since the Industrial Revolution, technological progress has regularly displaced some workers but led to the creation of new jobs as the growing wealth generated by better technology led to growing demand for additional goods and services, enabling the displaced workers to take up new opportunities (Autor, 2015). As a result, the economy adjusted to

the displacement. By and large, workers in advanced countries today are much better off than they were at the beginning of the Industrial Revolution (Maddison, 2004). However, there are reasons to be concerned that the ongoing rapid advances in LLM capabilities in particular, and AI in general, might disrupt this pattern.

Rapid advances in LLMs pose three distinct sets of challenges for workers' incomes (Korinek and Stiglitz, 2019; Susskind, 2023). First, they are likely to accelerate the rate of job turnover and disruption —— affecting more workers, including more highly skilled workers, and making the adjustment process for society more difficult than what we were used to from prior technological advances. Research could help identify what sectors are most vulnerable to job disruption (Eloundou et al., 2023) and how the displaced workers within those sectors can be helped to become productive workers in other growing sectors. Second, although technological progress means that society may produce more wealth overall, there is a risk that the general-purpose nature of LLMs may lead to progress that is biased against labor, meaning that the share of that wealth that goes to labor may decline. The overall effect on wages is then a horse race between these two opposing forces. The fate of blue-collar workers in recent history may be a harbinger of what lies ahead for white-collar workers: as a result of skill-biased technological change, the wage of the median male blue-collar worker has declined over the past four decades, when adjusted for inflation, even though the overall economy in the US has tripled (Autor, 2019). Early results suggest a negative impact of LLMs on certain categories of jobs (Hui et al., 2023). Third, if future LLMs and robots advance to the point where they can perform virtually all the work tasks, they would disrupt labor markets more fundamentally: if machines can do workers' jobs, wages would fall to machines' user cost (Korinek and Juelfs, 2023). This would pose fundamental challenges for labor markets and income distribution (Korinek, 2023).

Aside from their effects on incomes, the rapid advances in LLMs also risk reducing job quality. Automation often tends to increase not only the physical but also the emotional demands put on workers, exposing them to greater surveillance, higher job intensity, and less human agency (Bell, 2022). These trends have already been observed in the increase of digital labour (Casilli and Posada, 2019) and the platformization of labour more generally (Nyabola, 2023); as LLMs become more widely applied they could exacerbate these trends.

More fundamentally, work is not only the main source of income for the majority of people, but also the main activity that occupies our time. As a result, many derive a significant part of our identity, life satisfaction, and meaning from work (Susskind, 2023). If people lose their work, they would thus lose much more than their income, with broad implications for our society and our political system (Bell and Korinek, 2023).

Two observations might help tackle the resulting challenges (Korinek and Juelfs, 2023). First, if the non-monetary aspects of work are so important, people could presumably continue to work even when machines can automate it — they just may not earn much of an income from it. In surveys, a majority of workers are not very satisfied with their work (Gallup, 2022) and would likely be happier if they received the same income without the need to do a job. Second, things become trickier if one individual's decision on whether to work affects others' well-being, for example, because it affects others' scope to form social connections at work or because it has implications for political stability. Economists call such effects externalities. When significant externalities are at work, there is often a role for public policy to improve upon the socio-economic equilibrium. Research could work to identify what the best interventions could be that might be implemented via appropriate public policy. One such intervention could be to mandate LLM-developers to conduct impact assessments or risk assessments for whether their AI systems augment workers and improve working conditions. However, the general-purpose nature of the LLMs may make conducting such risk assessments challenging.

### 4.4.2 Effects on Inequality

LLMs could potentially worsen socioeconomic inequalities (Capraro et al., 2023). Effects on inequality are closely linked to the effects of LLMs on workers but ultimately depend on how the fruits of technological progress are distributed. There are three important challenges associated with this:

First, if the role and compensation of capital rise and the role and compensation of labor decline in an LLM-powered economy, inequality may go up because work is the main source of income for the majority of people. There are signs that this may already have happened because of earlier automation technologies in recent decades (Karabarbounis and Neiman, 2014). This has led to calls to steer advances in AI in a direction that complements workers rather than substituting them (Korinek and Stiglitz, 2020; Klinova and Korinek, 2021). This also applies to LLMs. A tangible example of an LLM that complements workers would be a system that advises call service center agents how to best handle calls (Brynjolfsson et al., 2023). An example that substitutes for workers would be a system that replaces workers altogether. Given the general-purpose nature of LLMs, there is a risk that advances in any given use case may initially complement workers but eventually progress to a stage where they can displace them. However, human decisions can influence the extent to which LLMs complement vs. substitute workers. This includes developers of such systems, business leaders who deploy them, and lawmakers and regulators who shape the economic and societal context in which these systems operate. Economic and sociological research could help understand how LLM-based technologies are likely to get adopted, for instance, by interviewing workers who are early adopters or studying historical examples of integrating new technologies into workspaces or industries. There is also a need for concerted efforts to best educate the business leaders on leveraging the growth benefits of LLMs in a way that is minimally disruptive to the larger society.

Second, the large fixed cost of training cutting-edge LLMs and the network effects involved imply that the market for the most advanced LLMs tends towards a natural monopoly structure in which only one or a small number of players will be successful, a phenomenon that has been termed 'algorithmic monoculture' in the literature (Kleinberg and Raghavan, 2021; Bommasani et al., 2022). As a result, LLM developers may amass significant market power. This might result in reduced social welfare, and lead to LLM-providers extracting monopoly rents from their customers (Kleinberg and Raghavan, 2021; Jagadeesan et al., 2023). These concerns may become even more important if the producers of LLMs engage in vertical integration and also participate in the market for downstream applications, for example, if an LLM company also enters the market for legal advice, medical services, etc. In the limit, the market for leading LLM providers could be the entire economy as the capabilities of LLMs improve across the board (Korinek and Vipra, 2024). This centralization of power could be even more problematic due to the potentially negative impact on the governance of LLMs (c.f. Section 4.5.4). The tendency towards centralization could be mitigated by a robust antitrust response and other policy interventions. However, this requires ensuring that economic policymakers are cognizant of the technological scenarios that lie ahead and their economic implications. The onus is on the technical community, which is generally more prescient about the impending technological developments, to effectively communicate and engage with economic policymakers in this regard. They may do so by writing educational documents aimed at a non-technical audience (e.g. Bowman, 2023), or releasing policy briefs on their research (e.g. Barrett et al., 2023). An alternative solution could be to prevent materialization of an algorithmic monoculture by increasing access to LLMs and broadening the pool of LLM developers, e.g. via open-source and open-science (Solaiman, 2023). Researchers could help improve the understanding of the extent to which current dynamics, in which leading LLM developers remain closed-source, but a number of less capable open-source competitors exist, are likely to lead to monopoly and identify interventions that could help mitigate them.

Third, as LLMs are becoming more powerful, who has access and who hasn't is becoming a more and more important question. For example, automated coding tools have been shown to produce

significant productivity gains, e.g. $> 50\%$ in some cases (Peng et al., 2023). Individuals who don't have access —— whether it is for financial reasons, for reasons of education, because of corporate or governmental policies, or for geopolitical reasons — might be at a growing disadvantage. In recent decades social scientists have observed a growing digital divide based on unequal access to digital technologies (Van Dijk, 2020). LLMs risk giving rise to a new 'intelligence divide' based on who has access to the most intelligent LLMs and who hasn't.

### 4.4.3 Economic Challenges for Education

In a knowledge-based economy, education equips individuals with the human capital that prepares them to be productive workers. Human capital is often considered as our greatest asset. Yet the rapid emergence of LLMs poses three significant economic challenges for education:

First, given the rapid emergence of LLMs as a powerful productivity tool, virtually the entire white-collar workforce may need to learn to use LLMs. Cognitive workers may need to learn how to prompt and steer LLMs, how to properly evaluate the output generated by LLMs, and how to incorporate them into their workflows in order to optimally benefit. This is one of the factors slowing down the rollout of LLMs in organizations throughout our economy (McAfee et al., 2023).

Second, while educators are challenged to teach new tools, LLMs also force them to fundamentally rethink the way in which they provide education and evaluate students (Mollick and Mollick, 2023; Cotton et al., 2023). LLMs could help unlock teaching methodologies that were not previously viable and improve the overall education quality (Khan, 2023). However, issues such as low technological readiness may hinder the rapid adoption of LLMs in educational contexts (Yan et al., 2023). There is a need for further research to better understand these issues and how to mitigate them.

Third, for education, a dark side of the positive productivity effects of LLMs is that they devalue a significant part of the human capital that workers have accumulated in the past. A growing number of studies show that lesser-skilled workers benefit more from LLMs than more highly-skilled workers (Noy and Zhang, 2023; Dell'Acqua et al., 2023). Another way of putting these results is that the human capital of highly skilled workers is no longer as valuable as it used to be, which may eventually lead to reduced hiring and training, and lower wages for skilled workers. This could have negative second-order effects. Currently, skilled labour presents a technical and moral bottleneck to the deployment of AI for malicious purposes (e.g. tech workers protesting Project Maven Shane and Wakabayashi, 2018), but as their jobs become (fully or partially) automated or, these workers may have less ability and agency to contest the uses of LLMs, increasing the hegemony of a narrowing set of powerful actors.

### 4.4.4 Global Economic Development

Many of the themes and challenges that we discussed above come together when analyzing the socio-economic effects on developing countries. The workforce of developing countries may suffer from a retrenchment of outsourcing as many simple cognitive tasks that used to be performed in developing countries — for example, in call centers —— can be automated with LLMs. This may adversely affect the economies of the poor countries (Georgieva, 2024). The inequality implications of LLMs also have an international dimension as there may be a growing 'intelligence divide' between advanced countries that have access to leading LLMs and poorer countries that do not. Differential productivity effects may give rise to terms-of-trade losses for developing countries (Korinek et al., 2022). Similarly, many of the profits generated by leading LLMs will accrue to advanced countries. On the other hand, developing countries may share in the productivity benefit of LLMs with advanced countries if LLMs are accessible to populations in Global South, both in terms of technology design and in terms of pricing. Surface-level usage statistics seem to indicate that freely available LLMs have indeed seen considerable adoption among Global South populations (e.g. India and Brazil are the second and third largest sources of web traffic to ChatGPT; Similarweb, 2024). However, there may exist extrinsic factors that might limit wider penetration of LLMs among Global South populations, such as lack of internet connectivity and

poor tech literacy among the populations. LLMs could potentially be leveraged to address some of the economic challenges faced by countries in the Global South. For example, teacher shortage is a critical issue that negatively impacts quality of education among Global South countries (Unesco, 2022). It is plausible that LLMs could be used to help mitigate this issue.

Additionally, in contrast to most other technologies, global adoption of LLMs requires that they are proficient in all world languages. However, even multilingual models perform worse in lower-resource languages (Etxaniz et al., 2023; Shen et al., 2024) and "think" in English even when fine-tuned for other languages (Wendler et al., 2024). This has severe safety and security implications as models that might be aligned in English might not be equally well-aligned in other languages (Deng et al., 2023a; Yong et al., 2023) or may perform worse in other languages (Aroyo et al., 2024; Holtermann et al., 2024). Additionally, LLMs may cost an order of magnitude more to use in some languages (up to 15 times in ChatGPT's case; Ahia et al., 2023; Petrov et al., 2023b). Hence, ensuring that LLMs work well regardless of the language in which they are used is key if we want to ensure they are a tool for reducing global inequalities rather than further exacerbating them.

---

The socioeconomic impacts of LLMs have the potential to be highly disruptive if not effectively managed. LLMs are likely to adversely affect the workforce, exacerbate societal inequality, and introduce new challenges for the education sector. Furthermore, the implications of LLM-based automation on global economic development remain uncertain. These challenges are complex and systemic in nature; there do not exist any simple fixes. To devise solutions, we need to develop a deep and nuanced understanding of these issues. Answering the following questions may help make progress towards this goal.

166. How can we better understand and forecast the disruptive effects of LLMs on job availability and job quality in different sectors? How can displaced workers be helped to transition to other sectors?↩

167. How can LLM developers best conduct impact assessments or risk assessments for whether AI systems improve working conditions (by augmenting workers) or not? ↩

168. How can LLM-based systems be designed to augment workers and improve working conditions, as opposed to automating and discplacing workers? ↩

169. How can we best educate business leaders to leverage the growth benefits of LLMs in a way that is minimally disruptive to society? ↩

170. How likely is the market for advanced LLMs to become a monopoly or oligopoly? What will the ramifications of such market concentration be on wealth distribution across society? ↩

171. How can we ensure equitable access to LLMs for individuals of all socioeconomic backgrounds?↩

172. To what extent are LLMs likely to exacerbate an 'intelligence divide' based on access to the most advanced LLMs?↩

173. How can LLM developers best keep economic policymakers updated on the technological scenarios that lie ahead and their economic implications (e.g. by writing policy briefs or writing informal educational documents)? ↩

174. How do we best educate the workforce for the effective use of LLMs and retrain disrupted workers?↩

175. What factors might impede adoption of LLM-based technology in educational contexts? How can these factors be mitigated? ↩

176. How can we better understand the second-order effects of LLM-driven automation on AI safety and alignment? E.g. could LLM-driven automation reduce the agency and ability of skilled labor to resist immoral usage of technologies? ↩

**177.** How might LLM-based automation negatively impact the economies of Global South countries?↵

**178.** How accessible are LLMs to Global South populations? How can this accessibility be improved? What measures can be taken by governments to address issues related to lack of internet connectivity, poor tech literacy etc.?↵

**179.** How can LLMs be used to help address some of the issues that hinder the economic development of Global South countries? For example, how can LLMs be used to help improve the quality of education available to Global South populations?↵

**180.** How to ensure that LLMs support all the world's languages equally, especially low-resource languages with large number of speakers? How do we ensure that LLMs are a tool for a global levelling up rather than further exacerbating economic divides? ↵

## 4.5 LLM Governance Is Lacking

Governance will play an essential role in the safety and alignment of LLMs, in particular, and AI in general (Bullock et al., 2022a). Governance encompasses not only formal regulations, but also a number of other mechanisms including norms, soft law, codes of ethics, co-regulation, industry standards, and sector-specific guidelines (Veale et al., 2023); see Table 5. Governance could supplement technical solutions, e.g. by mandating they be applied as appropriate. It could also *substitute* for technical solutions by preventing unsafe development, deployment, or use of LLMs when there do not exist sufficient technical tools for safety. **However, serious efforts to govern LLMs remain fairly nascent (e.g. US Executive Order** (White House, 2023)**, or EU AI Act** (Council of the European Union, 2024)**) and efforts to date have mostly been ill-defined and/or voluntary.** A number of governance challenges remain that ought to be addressed, or mitigated, to ensure LLMs are beneficial to society and do not contribute to harm to any societal group. Within this section, we divide our discussion of challenges that hinder LLM governance into two parts. We first discuss *meta-challenges* that complicate governance such as lack of requisite scientific understanding of LLMs, lack of effective fast-moving governance institutions, lack of culpability schemes, and corporate power. We complement this with a discussion of *practical challenges* focused on the fact that most governance mechanisms are underdeveloped and concrete proposals for governance are lacking. We note that our discussion on governance challenges complements other related works (Dafoe, 2018; Anderljung and Carlier, 2021; Shavit et al., 2023; Barnard and Robertson, 2024).

### Meta-Challenges — Challenges That May Limit Efficacy of LLM Governance

The governance of generative models (both LLMs and generative image models) is challenging due to factors such as the rapid productization of the technology, economically disruptive nature of the technology (c.f. Section 4.4), high potential for misuse (c.f. Section 4.2), and the rapidly evolving technological landscape (Bengio et al., 2023). Indeed, several *meta-challenges* exist that might limit the efficacy of LLM governance. These challenges include a lack of scientific understanding and technical tools necessary for governing LLMs effectively, the need for agile governance institutions capable of keeping pace with technological advancements (which is an antithesis to the traditional slow bureaucratic nature of the governments), a need to better understand the competitive dynamics between AI companies to ensure that competitive pressure does not result in irresponsible AI development, risks of regulatory capture due to corporate power, a need for international cooperation and consensus, and a lack of clarity on accountability for harms caused by LLMs.

### 4.5.1 Lack of Scientific Understanding and Unreliability of Technical Tools Complicate Governance

Effective governance of a technology hinges on three key elements: a comprehensive scientific understanding of the technology to gauge *potential* risks, dependable auditing tools to evaluate *practical*

Table 5: Examples of different governance mechanisms relevant to the governance of LLM, and AI broadly.

| Governance Mechanism | Examples |
|---|---|
| Global Frameworks, Agreements or Conventions | Draft Council of Europe Framework Convention on AI, Democracy, and Human Rights (Council of Europe, 2023) |
| Regional Regulation | EU General Data Protection Regulation (GDPR) (Voigt and Von dem Bussche, 2017)
EU AI Act (Council of the European Union, 2024) |
| Domestic Regulation | China Administrative Provisions on the Management of Deep Synthesis of Internet Information Services (Finlayson-Brown and Ng, 2023)
USA AI Initiative Executive Order (White House, 2023) |
| Sub-national Regulation | California Consumer Privacy Act (CCPA) (Goldman, 2020) |
| International/supranational "soft law" | OECD Recommendation on AI 2019 (OECD, 2019)
EU Ethics Guidelines for Trustworthy Artificial Intelligence, 2019 (AI-HLEG, High-Level Expert Group on Artificial Intelligence, 2019)
UNESCO 2021 recommendations on the ethical use of AI (Unesco, 2021) |
| National "soft law" | UK NCSC "Guidelines for secure AI system development", 2023 (National Cyber Security Center, 2019) |
| Industry Co-regulation | Partnership on AI (PAI, 2017)
Frontier Model Forum (Frontier Model Forum, 2023) |
| Industry Self-regulation | Anthropic's Responsible Scaling Policy (Anthropic, 2023e)
Google AI Principles (Google AI, 2018) |
| Standards organizations outputs | ISO data governance instruments (ISO, International Organization for Standardization, 2017)
IEEE's Ethically Aligned Design (IEEE, Institute of Electrical and Electronics Engineers, 2018)
CEN/CENELEC work on standards to implement EU AIA (ongoing)
US NIST Artificial Intelligence Risk Management Framework (AI RMF) (NIST, National Institute of Standards and Technology, 2023) |
| Internal institutional policies | University research ethics committees
Company AI ethics and safety research teams (e.g. Microsoft's FATE team, Deepmind's Scalable Alignment team), boards and codes |
| Private legal instruments | Contracts: e.g. Microsoft-OpenAI deal (Bradshaw et al., 2023)
Licenses: e.g. RAILS license (Responsible AI License) (RAIL Team, 2024) |

risks, and effective methods to intervene upon and *mitigate* these risks (Raji, 2021). However, as noted throughout the agenda, all three are currently underdeveloped for LLMs. Critical aspects of LLMs, such as in-context learning (Section 2.1) and reasoning abilities (Section 2.4), are poorly understood. Furthermore, existing auditing tools, including evaluation (Section 3.3) and interpretation methods (Section 3.4), are not reliable enough to provide meaningful assurance of model safety outside of very narrow contexts. And the primary technical tool for mitigating risks, safety finetuning, lacks robust generalization (Section 3.2).

These issues complicate governance and contribute to a lack of scientific consensus on the nature and severity of risks associated with LLMs. This lack of technical clarity is one reason Guha et al. (2023) and Kapoor et al. (2024) caution against rushing to regulate. Yet, the process of establishing regulation has tended to be slower than the rate of AI progress. Thus, it is crucial to generate and evaluate governance proposals *despite* our presently limited technical understanding. Proposals could explore how governance approaches could be adapted based on the level of technical understanding of the models and the rate at which the field might be progressing; for instance, how should a governing body change risk thresholds in response to new evidence? Alternatively, proposals could seek to accelerate technical understanding of LLMs and their risks and harms, e.g. by funding research and building government capacity. Moreover, understanding the interconnections among diverse risk types is required to inform strategies for risk governance (Kasirzadeh, 2024). Another proposal is a temporary *pause* or slowdown on AI and LLM development (Future of Life Institute, 2023), based on the hope that this time could be used to develop standardized safety protocols and make progress on safety, e.g. through technical research. However, this proposal has received much criticism. It may adversely affect AI alignment and safety research and cause 'overfitting' of alignment research to whatever the state-of-the-art model at the time of the pause might be (Belrose, 2023) or might be infeasible to enforce (Luccioni, 2023). On the whole, it remains unclear whether governing bodies should aim to moderate the pace of progress in AI, and if so, what governance mechanisms (e.g. compute governance, see Section 4.5.11) could be used for this purpose.

### 4.5.2 Need for Effective, Fast-Moving Governance Institutions

Given the rapid pace of advances in LLMs, governance institutions will need to adapt quickly to remain effective. Governments are currently attempting to regulate AI both through existing institutions, such as NIST in the USA (White House, 2023), or by novel legislation, such as the EU AI Act in Europe (Council of the European Union, 2024). However, capacity issues might limit the ability of governments to design and implement effective policies for governing LLMs quickly (Marchant, 2011). For example, regulatory bodies might struggle to enforce laws due to the lack of resources and relevant expertise, as has been the case with the enforcement of data protection laws (Jelinek and Wiewiórowski, 2022). However, despite these shortcomings in practice, government regulation has unique legitimacy and authority. Hence, it is worth asking how these shortcomings may be addressed.

One approach to overcome these limitations could involve creating new institutions through legislative measures. For example, Tutt (2017) proposes an FDA-like body for algorithmic systems. There is indeed a growing trend towards the enactment of tailored AI laws, particularly led by the EU and China. These laws may encompass large models as a specific category, such as the provision of the EU AI Act concerning foundation models or general-purpose AI (GPAI) (Weatherbed, 2023). Investigating the impact of such laws on the development and application of LLMs is one clear direction for research; this could involve a comparative study of different national approaches to LLM regulation, and/or an analysis of the priorities, assumptions, and ideologies driving different AI regulations (Au, 2023).

On the other hand, existing regulation could also be fruitfully applied to the governance of LLMs (Gaviria, 2022; Bhatnagar and Gajjar, 2024). This includes copyright and data privacy laws, speech laws, labor laws, advertising standards, tort, product liability, and more. However, it remains unclear to what extent current regulation is sufficient to mitigate risks. Existing regulators may also not

have the necessary capacity, expertise, and regulatory authority to effectively regulate LLMs. Engler (2023) argue for expanding the powers of the existing regulatory bodies — in particular granting them the power to issue subpoenas for algorithmic investigations and for setting up rules for especially impactful algorithms. Further research here identifying opportunities and gaps would be valuable. This could include analyses of the current technical capacities of regulatory bodies as well as their resource allocation and knowledge acquisition strategies. Furthermore, the merits and demerits of various forms of public-private partnership proposals for addressing capacity issues of governments could be explored. This may include analysis of both traditional forms of public-private partnerships such as governments directly contracting the services of a private actor for various tasks, or novel proposals such as regulatory markets (Hadfield and Clark, 2023). Center for AI Safety et al. (2024) argue that current proposals for regulatory markets are flawed and do not adequately incentivize private regulators to prioritize safety. In general, while the private regulatory institutions may be more agile than government institutions and could help standardize regulation across jurisdictions, they may also increase the risk of regulatory capture, as private partners might have strong industry ties (e.g. financial interests in LLM development, deployment, or use Center for AI Safety et al., 2024); and such a set-up may also bypass government processes meant to protect against regulatory capture. Generally, there is a need to better understand the robustness and efficacy of different ways through which governments could outsource auditing and regulation to private actors (Costanza-Chock et al., 2022). There is also a need to identify measures that could be taken by governments and other public-interest institutions (e.g. universities, NGOs, policy think tanks) to create an ecosystem that incentivizes top talent to contribute to governance objectives — e.g. by working for government regulators, private auditors or other public-interest bodies — given the large compensation packages being offered by leading AI companies (Mann, 2023).

### 4.5.3 Incentivizing Cooperation and Disincentivizing High-Risk Approaches to AI Development

Responsible and safe AI can be seen as a collective action problem, creating an additional challenge for governance to confront (Askell et al., 2019). More precisely, while most AI practitioners might prefer a low-risk approach, competitive pressures (e.g. resulting from Safety-Performance Trade-offs, c.f. Section 2.7) might cause one or more organizations to take a higher-risk approach (e.g. a 'move fast and break things' approach Blodget, 2009). This could trigger a race to the bottom, progressively increasing competitive pressure on other organizations to adopt higher-risk approaches (Armstrong et al., 2016; Hendrycks et al., 2023). For these reasons, it is crucial to disincentivize high-risk approaches to AI development, deployment, and use, and to prevent such dangerous dynamics from materializing. Regulation and treaties could mandate safe development practices. In particular, specialized regulation could be designed with an explicit focus on the development of frontier AI (Anderljung and Korinek, 2024). *Outside-the-box* auditing covering organizations' development and deployment practices could help verify compliance (and identify risks that may have been overlooked) (Casper et al., 2024a). Technical researchers could develop better (game-theoretic) models of race dynamics whose analysis may yield insights about the relative effectiveness of various interventions that could be undertaken to disincentivize a race. Armstrong et al. (2016) provide a preliminary analysis of this kind, however, their analysis relies on various simplifying assumptions that may not be reflective of real-world dynamics. In general, technical researchers could work to identify safe development and deployment practices and to identify incentives and technical loopholes that might lead developers not to adopt them. They could also work with legal and business experts to evaluate the effectiveness of potential regulations by anticipating how developers might respond.

### 4.5.4 Corporate Power May Impede Effective Governance

The increasing power and influence of large corporations may make effective governance difficult. There exists a power asymmetry between corporate entities profiting from LLMs and other social groups (e.g. civil society). State-of-the-art LLMs are developed by or in partnership with, some of the world's largest private tech companies. NVIDIA, which supplies most of the computing hardware for LLMs has recently seen its market capitalization increase to over $2 trillion, roughly a 5x increase in the

past year. Lobbying efforts around LLMs have also increased dramatically in the recent past (Saran and Mattoo, 2022; Zakrzewski, 2022; Lindman et al., 2023). This poses a risk of governance protocols related to LLMs becoming excessively favorable to tech companies, potentially leading to regulatory capture at the cost of the interests of other societal groups, particularly marginalized communities who have historically been disproportionately affected by poorly designed AI technologies (Reventlow, 2021). Researchers working on LLM policy should aim to document and account for corporate influence, with a focus on identifying ways in which corporate interests might diverge from the public interest. At a more meta-level, it may be helpful to consider how corporate structures could be designed so that the interests of corporations extend beyond mere maximization of profits, and better align with the larger interests of the society. Researchers could design novel proposals in this regard, or evaluate the robustness of the existing proposals (e.g. Anthropic's Long Term Benefit Trust Anthropic, 2023f). This is of particular relevance given the recent power struggle between OpenAI board members, founders, and investors (Roose, 2023; Reich, 2023).

In addition to their ability to influence governance through lobbying efforts, LLM developers may also directly influence public opinion through the design choices they make. For instance, how an LLM expresses itself and responds to queries, especially those with a political angle to them, can have a significant impact (Jakesch et al., 2023; Santurkar et al., 2023). This direct influence that technological companies can exert on public opinion has been recognized in prior work. Lessig (2006) discuss the power of technology or "code" itself as a governance mechanism, noting that it generally lacks democratic oversight and that much software used globally is built by US-based companies, often without regard to the local laws of its overseas markets (Sánchez-Monedero et al., 2020). To help improve democratic oversight, machine learning researchers could consider how such influence might be measured, detected, and counter-acted (O'Callaghan et al., 2015). There is also a need for collaboration among machine learning researchers, sociotechnical researchers, and civil servants to determine how to best protect government institutions such as democratic processes from such influence, e.g. to help establish standards defining the limits of legitimate influence.

While the academic community can help counterbalance corporate influence over governance, the large majority of AI researchers receive or have received funding from big tech companies (Abdalla and Abdalla, 2021). This dependency on industrial funding may grow further due to the exorbitant costs of conducting LLM research. More research is needed to identify the influence of corporate power on academia, policy, and research. It is also important to consider governance proposals that may help the academic community preserve its independence, and generally increase the agency of academics to contribute effectively to safety and alignment research (Raji et al., 2023). The lack of academic consensus around AI risks and mitigations also hinders the academic community's ability to play such a role, rendering us unable to speak with one voice to provide clear guidance to policymakers or other elements of civil society.

### 4.5.5 LLMs Require International Governance

International governance will be critical for tackling factors such as competitive dynamics between AI companies. However, despite the fact that data-driven AI is a global and cross-border phenomenon, laws and regulators are predominantly national. As a result, issues of jurisdiction, applicable law, and regulatory arbitrage, which already complicate effective regulation of data privacy laws, will almost certainly affect AI and LLM regulation. A popular proposal in the literature is the establishment of international institutions (Ho et al., 2023; Trager et al., 2023; Maas and Villalobos, 2023). For example, Trager et al. (2023) propose the creation of an inter-state International AI Organisation to certify compliance by states to international oversight standards while supporting member states in meeting these standards through domestic regulatory capacities. The UK has promoted the development of global AI safety Institutes alongside a pledge to put safeguards on models posing systemic risks (of Participating Countries, 2023). As of early 2024, AI safety institutes are being established in the UK, US,

Japan, and Singapore at the *national* level. These organizations could be a productive site of more international collaboration on AI governance.

One other possible route to global harmonization of AI regulation is via the global diffusion of domestic regulation, such as the EU AI Act (Council of the European Union, 2024). The EU aspires to position itself as a global 'gold standard' (building on the example of GDPR), by requiring any company selling into the EU Single Market to follow its rules. Through the Brussels effect, it may be cheaper for companies that comply with EU requirements to comply with the same requirements even in other jurisdictions (Siegmann and Anderljung, 2022). However, conflict historically exists between the EU model of human rights-centric prescriptive risk-based regulation and the US's laissez-faire regulation of Silicon Valley and poor federal privacy protection (Solove and Schwartz, 2022). This may hold back any international harmonization of AI regulation (Kaminski, 2023; Smuha, 2021). However, there is some initial evidence that the US government may take a proactive approach to regulating AI (White House, 2023), suggesting that this conflict may not adversely impact the international governance of AI.

Another important development here may be the finalization of the Council of Europe Convention on AI (Council of Europe, 2023) which like the European Convention on Human Rights and the Cybercrime Convention can be joined by non-European states, and so is potentially a global treaty. Alongside these developments, China has been promoting cooperation on international AI governance through initiatives such as the Global AI Governance Initiative (Ministry of Foreign Affairs, People's Republic of China, 2023). While this initiative has elements in common with both the EU approach and UK and US approaches, there exists a risk that multiple separate or overlapping international AI governance forums will emerge, complicating efforts to achieve global consensus on standards and safety. International governance may also be impeded by a trust deficit and an arms race between nations — particularly China and the U.S. (Meacham, 2023). Overcoming parochial concerns and assuring that governments can credibly cooperate on critical issues — such as limiting the use of AI in weapons (see Section 4.2.4) is perhaps the grand challenge in international governance of AI. Dialogue between the scientific communities could help pave the way for such cooperation (FAR AI, 2024; Maas and Villalobos, 2023). Specifically for the governance of AI in military applications, multi-country military alliances like NATO could play a critical role in ensuring responsible, and ideally highly restricted, use of AI in weapons (Stanley-Lockman and Trabucco, 2022).

### 4.5.6 Culpability Schemes Are Needed for LLM-Based Systems — Especially LLM-Agents

Providing assurances about a system's safety and intended behavior inherently requires establishing clear accountability — explicitly assigning responsibility and culpability in cases where the system acts in undesirable or unintended ways. It is currently unclear who ought to be held responsible when a LLM system causes harm to its user or other humans. Users may deliberately misuse LLMs (see Section 4.2), but preventing such misuse may be intractable without interventions at the development or deployment stage (Anderljung and Hazell, 2023). Blumenthal and Hawley (2023) propose that companies be held liable for harms caused by "their models", however, Kapoor et al. (2024) observe that developers who open source their model would struggle to prevent misuse and attendant liability. Developers are often extremely well-resourced and can retain privileged access to their systems. Hence, they are technically best equipped to detect and mitigate safety issues, but it is unclear to what extent they are incentivized to disclose those issues, especially if disclosing them could harm their business interests. Importantly, it is not necessary to hold only a single actor (user, deployer or developer) responsible: different actors could be held responsible to varying degrees (Wex, 2023).

Gaps in how to assign responsibility will likely become more acute as systems become increasingly agentic and autonomous (Buiten et al., 2023) (see Section 2.5), which can amplify the harm that they can cause (Chan et al., 2023a). Thus, there is a need for anticipatory governance to preemptively address the risks posed by LLMs, for example, by establishing regulatory criteria to mediate the deployment

of LLM-agents. This will be helped by developing frameworks to monitor and evaluate deployed LLM-agents and designing mechanisms for ascertaining accountability in case of failures (Kampik et al., 2022; Chan et al., 2024). These questions are considered in greater detail in the recently released agenda on agent governance by Shavit et al. (2023).

Once deployed, LLM-agents will interact among themselves and with humans — creating an additional source of risks (see Section 2.6). Normative infrastructure (e.g. bureacracies Bullock et al., 2022b) which governs interactions between humans — e.g. accounting of responsibility and blame — can break down with the introduction of algorithmic or machine-based decision-making. For example, high-frequency algorithmic traders are believed to have contributed to a number of flash crashes in stock markets (Tee and Ting, 2019), e.g. the flash crash of 2010 (CFTC and SEC, 2010) and the 2014 US Treasury market flash crash (Levine et al., 2017). Similar to algorithmic traders, LLM-agents will likely possess novel capabilities and affordances, such as higher processing and action speed, the capability to ingest large amounts of text rapidly, etc. that might disrupt normative infrastructures in undesirable ways. Further work is required to understand better the governance challenges posed by LLM-agents (Kolt, 2024), especially in multi-agent scenarios where they may interact and influence each other (Hammond et al., 2024). A better understanding of these challenges may help inform what technical and governance tools are necessary for effective governance of LLM-agents.

## Practical Challenges — Governance Mechanisms for LLMs Are Underdeveloped

In addition to the meta-challenges discussed above, a key challenge in exercising LLM governance is a lack of concrete, and complete, governance proposals (Guha et al., 2023). That is, most governance mechanisms for LLMs are underdeveloped and there is a high level of uncertainty around what the best governance interventions will be. To elaborate, governance can operate at different points in the LLM lifecycle, from development through to deployment and use, as well as on different substrates such as data (Jernite et al., 2022; Chan et al., 2022), compute (Hwang, 2018; Sastry et al., 2024), and energy (Monserrate, 2022). Governance interventions earlier in the lifecycle can create choke-points further on. For instance, if some development practice is banned so that some variety of LLM system is never developed, such a system could not be deployed or used (Anderljung and Hazell, 2023). On the other hand, interventions later in the lifecycle can be more targeted. This carries both advantages and disadvantages: more targeted interventions help limit negative side-effects of governance interventions (e.g. creating barriers to beneficial uses), but also run the risk of missing some pathways to harm. Fortunately, the different types of governance mechanisms we discuss are not mutually exclusive, and can likely be effectively combined to achieve better outcomes than using a single mechanism on its own. However, currently, almost all the governance mechanisms are underdeveloped and lack concrete proposals for operationalizing them. We provide some discussion in this regard below and highlight the relevant challenges.

### 4.5.7   Use-Based Governance May Be Insufficient

One approach to governing LLMs is to set rules that limit how they can be used. Indeed, Hacker et al. (2023) argue that governance should focus on users and deployers, with a few key exceptions. Under such an approach, users could be held accountable for whatever harms their use of a system causes, and consumer protection law could be used to hold developers or deployers accountable if they fail to protect users. Particularly harmful use cases could be proscribed by designing explicit regulations. For example, explicit laws are being passed by multiple countries that criminalize the generation of harmful deepfake images by the use of generative image models (U.S. Congress, 2023; UK Parliament, 2023). However, enforcement of such regulations may be very difficult as a skilled malicious user could easily hide their identity by using anonymization schemes (Eurojust and Europol, 2019). Hence, it is questionable to what extent such regulations will be an effective deterrent for a highly skilled and determined malicious user.

It is also unclear to what extent such an approach would be able to proactively identify misuses of technology, instead of acting retroactively once the harm has been done; as was observed to be the case for deepfakes (Burga, 2024). Currently, the EU AI Act governs use cases of AI systems, including LLMs, using a risk-based approach to classify different use cases and determine which rules apply (Council of the European Union, 2024). However, several questions arise with such an approach: What existing regulations — or, more generally, governance institutions — are relevant, and how should they be applied? How can we identify problematic new use cases and ensure they are addressed (c.f. Section 4.2)? Regulators might need fast-acting powers to intervene when new problematic uses are discovered. More generally, an important challenge for such an approach is how to address issues surrounding misuse, such as accountability, discussed in Section 4.2). We note that international agreements might also be required, as many forms of misuse (e.g. military use of LLM or censorship) are likely to be perpetrated by governments. Use-based governance may also be limited in ways it can prevent instances of self-harm (Xiang, 2023b).

### 4.5.8 Deployment Governance Lacks Adequate Regulation

Deployment methodology significantly impacts the potential risks associated with an LLM. Developing regulations, both soft and hard, to govern deployment can not only be effective but may also be necessary to assure safe and beneficial deployment of LLM-based systems. Deployment governance may include *pre-deployment* governance and *lifetime* governance.

The pre-deployment governance is concerned with regulating how a model gets deployed. In the pre-deployment governance, a basic challenge is to evaluate the trade-offs of different forms of deployment (Solaiman, 2023). The common forms of deployment include making the model available to download or making the model available via some limited form of API access (including web-interfaces).[17] In particular, there is an ongoing debate around downloadable or so-called 'open-source' model deployments.[18] Shevlane (2022) argue for the benefits of 'structured access', i.e. controlling how users interact with a system, which API deployment enables. Seger et al. (2023) argue that LLMs may soon be too dangerous to open-source (at least initially). On the other hand, Kapoor et al. (2024) argue that governments should fund research into the marginal risk from open source models (over currently available tools, such as web search), and consider further interventions only once risks are more certain. They also express concern that many forms of regulation might impose infeasible compliance burdens on developers of open-source models. Advancing this debate is critically important, given the irreversibility of open-sourcing models.

The risks of deployment (even for a closed-source model) are also mediated by the intended use-case (see Section 4.5.7), who the model is being made available to and especially by the level of autonomy afforded to the model (Weidinger et al., 2023a). The models that are made available to younger audiences (Fowler, 2023), or deployed autonomously ('LLM-agents') may require similarly higher levels of assurance (Chan et al., 2023a; Shavit et al., 2023). Regardless of how models are deployed, deployers could take some responsibility for ensuring LLMs are trustworthy and do not cause harm. For instance, they might be made responsible for communicating information about limitations from developers to users, or even be required to perform some evaluations or collect other information about a system before agreeing to deploy it. Third-party licensing, or registrations, could be mandated to ensure that unsafe technologies do not get deployed or become widely available.

While thoughtful development can reduce the risks associated with an LLM, it may not eliminate them. Lifetime governance is required to ensure that throughout their deployment, LLM-based systems remain safe. This includes assuring that the systems are monitored in a robust way and clear action plans are

---

[17]In practice, API deployers are often going to be large-scale compute providers; see Section 4.5.11 for more on compute governance.

[18]There is also controversy and confusion around the term 'open-source' as applied to the practice of *only* releasing model weights (Widder et al., 2023; Seger et al., 2023; Solaiman, 2023).

in place to deal with cases when a novel failure mode, or a new source of risk, is discovered (Chan et al., 2024). One challenge requiring technical research in this regard is: how to re-establish assurance after system updates to the LLM, or some other component of an LLM-based system, during the system's lifetime. Ideally, the cost of assurance in such a case could be reduced relative to assuring a brand-new system. Another aspect of the challenge is how to deal with downstream systems using an LLM as a 'dependency'. Deployers could help ensure users and developers share an awareness of how such updates might lead to new safety risks.

### 4.5.9 Development Governance Might Be Particularly Challenging to Codify and Enforce

Most technical work in AI safety and alignment is focused on development methods. While this work is currently not mature enough to offer reliable recipes for safe and aligned LLMs, it can already contribute to best practices that could be enshrined e.g. as standards or through regulation (UK Government, 2023). Such practices could take the form of rules around the sorts of data or algorithms to use (or not use), as well as rules around evaluations or other assurance practices to be applied before deployment (Schuett et al., 2023). These rules could be enforced on a per-project basis through mechanisms similar to those in White House (2023), provided governments are aware of development projects. Others have proposed licensing developers (Smith, 2023; Anderljung and Korinek, 2024), although critics argue this might lead to regulatory capture, and stifle innovation and open-source development (Thierer, 2023; Howard, 2023). Besides development methods, best practices for development should also consider 'meta-practices' such as processes governing internal decision-making and practices around disclosure of development activities (Weidinger et al., 2023a; Ojewale et al., 2024; Casper et al., 2024a). To the extent that safety and alignment can be guaranteed through following best practices in development, mandating such practices could be an appealing approach to governance.

However, the efficacy of development governance would likely depend on achieving a high level of buy-in from most – if not all – leading developers (see Section 4.5.3). Moreover, given the falling cost of compute, more and more developers (including those not "in the lead") may need to be governed, creating a growing enforcement challenge. As most of the knowledge about sound developmental practices for the responsible development of LLMs is currently locked within AI companies, regulators, and standard-setting organizations may be highly dependent on LLM developers sharing this knowledge to create high-quality standards. Furthermore, a lack of buy-in from developers may result in regulatory flight (i.e. developers moving their operations to other jurisdictions with less regulatory pressure) or developers circumventing the prescribed external standards, e.g. by hiding parts of developmental details that may be misaligned with the prescribed standards. Such evasions may be particularly hard to detect and prevent, given the current lack of technical tools for determining whether inappropriate development activities are occurring (Shavit, 2023). At the same time, regulatory flight may be less of a concern for large markets like the US or the EU.

An alternative to externally imposed regulations could be to prompt companies to propose their *own* developmental standards that they could then be beholden to, once ratified by an external party (e.g. a government regulator). However, it is important to not rely on voluntary compliance alone and to ensure that such standards are appropriately codified and made legally binding (Ó hÉigeartaigh et al., 2023). An example of such developmental standards are the 'responsible scaling policies' published by various AI companies (Anthropic, 2023e; OpenAI, 2023d; Google DeepMind, 2023), however, these policies are completely voluntary and due to the lack of third-party analysis of these policies, it is unclear to what extent these policies embody desirable standards of safety.

### 4.5.10 LLMs Pose Additional Challenges for Data Governance

Data is a basic ingredient for LLM development. This makes data governance a promising vehicle to govern and regulate LLMs. Data could serve as a choke-point and help prevent the development of unsafe LLM systems; for instance, training on certain kinds of data (e.g. biological data) could be prohibited or regulated stringently. In the pre-LLM age, the central focus of data governance has been

the protection of an individual's right to privacy (Solove, 2022). LLMs add an additional dimension to this as LLMs can memorize and leak personally identifiable information (PII) (Tirumala et al., 2022; Nasr et al., 2023). However, it is also important that the scope of data governance is expanded to consider other *data rights* issues that have to come to the fore due to the development of LLMs (Roberts and Montoya, 2022). One major objective for data governance is establishing, and defending, the rights of data creators (e.g. writers) and the rights of data workers (e.g. workers hired to generate data for LLM training). A popular proposal for this is the establishment of accountable organizations, such as data trusts (Jernite et al., 2022; Chan et al., 2022), to be custodians of any data submitted to them. However, implementing such a solution requires overcoming several technical and social challenges. On the technical side, the key problems are establishing provenance of the data already present on the internet (Lee et al., 2023b), verifying that a particular model was trained on the dataset it is claimed to have been trained on (Choi et al., 2023b; Garg et al., 2023), and establishing *relative* value of different data points present within a dataset (Guu et al., 2023). On the social side, implementing such solutions would require strong political will, extensive international cooperation, and sufficient funding to implement the technical infrastructure needed for data trusts (Chan et al., 2022).

Furthermore, it is unclear as to who should own the data *created* by an LLM – e.g. the developer, the user, or no one (Henderson et al., 2023b)? This question is coupled with the questions of responsibility and profitability: who is responsible if an LLM generates output that is harmful or unsafe in other ways (Henderson et al., 2023c) (also see Section 4.5.6)? How does this responsibility change as we move from chat-based models to agents (Schwartz and Rogers, 2022)? This is arguably a fundamental question in regards to data economy and may have long-reaching repercussions in an AI-based creative economy (Knibbs, 2023).

### 4.5.11 Robustness of Compute Governance is Unclear

Compute plays a critical role in the development of LLMs (Sevilla et al., 2022), with compute costs for development rising into the hundreds of millions (Knight, 2023) and likely soon billions of dollars. Compute governance may provide one of the most promising levers for governing bodies to modulate the rate of progress within the technical AI field (Whittlestone and Clark, 2021; Sastry et al., 2024). This may be particularly important for the safety risks associated with advanced capabilities which compute-heavy frontier models are likely to obtain first (Pilz et al., 2023). Relative to other governance mechanisms that governments could use to regulate AI and LLM development, compute governance also has the advantage of potentially easier compliance verification (Brundage et al., 2020; Baker, 2023), especially given the major intermediary role of compute providers (Heim et al., 2024). However, further work is needed to understand and refine existing proposals for compute governance (Shavit, 2023; Choi et al., 2023b; Egan and Heim, 2023; Sastry et al., 2024; Heim et al., 2024), in addition to managing risks such as privacy and concentration of power (Sastry et al., 2024). Technical researchers could collaborate with hardware and supply-chain experts to stress-test proposals, e.g. by identifying potential loopholes by which projects might escape scrutiny (such as distributed training on consumer hardware Douillard et al., 2023), or otherwise thwart effective oversight (such as by disguising computations). Compute governance proposals could also be strengthened by developing a better understanding of the interplay between hardware and software (Mince et al., 2024). Future research on compute governance could explore potential developments that may affect the effectiveness of compute governance, such as changes in the structure of the compute-providing industry (Anderljung and Carlier, 2021) or the diffusion of AI capabilities (Pilz et al., 2023). Compute governance could also be leveraged by governments to enhance the ability of the independent scientific and academic community. In addition to any direct benefits in terms of improved understanding, and auditing, of LLMs, this may help mediate the effects of corporate power (see Section 4.5.4).

Effective governance of LLMs is critical for ensuring that LLMs prove a beneficial addition to societies. However, efforts to govern LLMs, and related AI technologies, remain nascent and ill-formed. The governance of LLMs is made challenging by various meta-challenges ranging from lack of scientific understanding and technical tools required for governance to the risks of regulatory capture by corporations. From a more practical lens, concrete and comprehensive proposals to govern LLMs remain absent and, unfortunately, the various governance mechanisms (e.g. deployment governance, development governance, compute governance, data governance) are not adequately developed yet.

181. How should governance approaches change depending on how rapidly the capabilities of models are advancing, the rate at which they are being productized (and hence proliferating throughout society), and the degree to which we lack technical understanding of a particular system, or LLMs in general? ↩

182. What policy interventions can be taken by governing bodies to support research on alleviating the technical limitations inhibiting effective governance of LLM-based systems? ↩

183. Should governing bodies aim to moderate the pace of progress in AI? If so, what governance mechanisms could be used for this? ↩

184. How might the slow, bureaucratic nature of governments negatively impact the governance of LLMs? What are the relative merits and demerits of various measures (such as forming public-private partnerships or formalizing regulatory markets) that could be taken by the governments to mitigate this issue? ↩

185. What measures can be taken to disincentivize irresponsible approaches to AI development? Can we design regulations that mandate the safe and responsible development of AI models? ↩

186. How can we better understand AI race dynamics, e.g. using game-theoretic models or historical analogues? And what governance interventions might be used to alter these dynamics? ↩

187. How can we involve and empower more stakeholders in LLM governance, particularly marginalized groups most impacted by LLMs? How can we avoid legislation that disproportionately favors the interests of corporate LLM developers over the interests of other social groups? ↩

188. Can we develop structures for corporate governance that might protect public interests in a better way? ↩

189. Can we develop technical tools that may help measure, detect, and counteract the role of technology companies in shaping public opinion, by influencing the content consumed by the public? ↩

190. Can we develop a better understanding of the influence of corporate power on academia, policy, and research? What are the potential detrimental effects of such influence? ↩

191. Can we develop a better understanding of the factors (arms race between nations, different national-level approaches to AI regulation) that might negatively impact international governance for LLMs? ↩

192. What are the different ways through which LLMs could be governed in a unified way internationally? ↩

193. How can clear lines of accountability be established for harms associated with LLMs? ↩

194. How can governance tools be used to mitigate the risks associated with LLM-agents and their interactions with humans and other systems? ↩

195. What are the relative merits and demerits of different governance mechanisms, such as use-based governance, deployment governance, developmental governance, data governance and compute governance? How can we effectively combine all the governance mechanisms to achieve the most favorable outcomes? ↩↩↩↩↩

196. What existing regulations and governance institutions can be applied for use-based governance, at national and international level? ↩

197. How can we proactively identify problematic uses and address them via use-based governance? ↩

198. How can use-based governance help deter misuses that are likely to be perpetrated by governments? Can it be effectively used to regulate against instances of self-harm? ↩

199. Can we develop a better understanding of the risks, and benefits, associated with various model deployment strategies? ↩

200. What kind of regulations can be adopted to ensure LLM deployers perform their due diligence in assuring system safety before and throughout deployment? ↩

201. How can we create appropriate legal frameworks for deployment governance of LLM-agents? These frameworks would need to address the regulatory criteria for deploying LLM-agents, how such agents should be monitored after deployment, and who would be accountable for any harm incurred by deployed agents. ↩

202. How can we efficiently assure an LLM-based system after a system upgrade to the LLM, or some other component of LLM-based system? ↩

203. What are the merits and demerits of requiring deployers to seek licenses, or register, with regulators prior to the release of the model? Should the requirements that deployers have to meet be different for different deployment strategies? ↩

204. How can developers best identify and share knowledge about responsible LLM (and AI) development among themselves? How can such practices be enshrined as legally binding standards? ↩

205. What are the merits and demerits of mandating licensing for the development of frontier AI technologies? ↩

206. Can we develop technical tools that may help us verify whether particular developmental practices were followed or not in the development of a given model? ↩

207. To what extent is regulatory flight likely to impede the effective governance of LLMs? ↩

208. What are the merits and demerits of 'responsible scaling policies' issued by different LLM developers? ↩

209. How can we establish and defend the rights of data creators (e.g. writers) and the rights of data workers (e.g. workers hired to generate data for LLM training)? Are data trusts (Chan et al., 2022) a practical solution in this regard? ↩

210. How can we verify that a particular model was indeed trained exclusively on the data claimed as training data by the model creator? ↩

211. Who owns the data created by an LLM? This is arguably one of the most critical questions in governance with downstream impact on other important questions of who bears responsibility for LLM outputs that cause harm to society, and who can profit from the LLM outputs. ↩

212. Can we develop concrete proposals for how compute governance could be exercised in practice? ↩

213. To what extent are current, and any future, proposals for compute governance robust to advances in distributed training? ↩

214. How will compute governance proposals be impacted by the changes in the structure of the compute-providing industry? ↩

215. How can compute governance be leveraged to enhance the ability of the independent scientific community to conduct investigations into flaws in LLMs that could otherwise be overlooked?↩

# 5 Discussion

## 5.1 Limitations

This agenda is the most expansive discussion on the challenges in assuring the safety and alignment of LLM-based systems to date. **However, despite this, we assert that this agenda is *not* exhaustive and that there exist important challenges, both known and unknown, in assuring the safety and alignment of LLM-based systems that are not cataloged in this work.**

We have attempted to 'future-proof' our work by trying to list challenges that might arise due to LLMs becoming more performant due to scaling or modifications of the training process, however, due to the uncertain nature of the LLM development landscape, it is possible that important challenges may have been omitted. In particular, we have primarily focused on challenges in the LLM safety and alignment that are imminent, and relatively undisputed, and hence, we do not cover speculative challenges; see Hendrycks et al. (2023) and Critch and Krueger (2020) for discussion on such challenges.

Another key limitation of this work is the exclusive focus on the safety and alignment of LLM-based systems. The choice to focus on the safety and alignment of LLMs was made due to the surging research interest in LLMs, their rapid productization, and their central position in the age of foundation models. We assert that the safety of other deep learning-based systems (e.g. generative models for vision, generative models for biology, recommender systems, and learning-based embodied agents) is also highly important and we call on the wider community to organize similar efforts to catalog challenges involved in the safety of such non-LLM-based systems. Relatedly, due to the limited scope of our work, we have intentionally omitted many important research directions pertaining to the safe development and deployment of aligned AI-based systems, such as improving the general understanding of deep learning (Arora et al., 2020), understanding critical aspects of agency ('agent foundations' Soares and Fallenstein, 2014), and developing AI systems whose safety can be proved or verified (Brundage et al., 2020; Tegmark and Omohundro, 2023), cooperative AI (Dafoe et al., 2020) etc.

Another dimension of our limited coverage is that our major focus is on technical challenges — 13 out of 18 challenges we identify are fully technical in nature. Our discussion of sociotechnical challenges is further limited in the sense that it is narrowly focused on challenges that are directly relevant to LLM-based systems and is biased towards aspects of these challenges that could potentially be addressed via research. We have also discussed sociotechnical challenges in a dedicated section as we prioritized modularity to make the work easier to navigate for a wide audience. However, this view oversimplifies the interconnectedness between the technical challenges and the broader sociotechnical challenges involved. An alternative treatment focused on highlighting this interconnected nature of safety challenges would see sociotechnical issues spread throughout every aspect of work on LLMs. Additionally, while we have made our best effort to avoid any geographic bias — in some sections, particularly Section 4.5, the discussion is biased towards few geographies, specifically, US, Europe and China.

The nature of the challenges posed in this work may change over time. Some challenges may get sidestepped or solved as a side-effect of advances focused on improving LLM performance (e.g. scaling). For some other challenges, their form may evolve, causing the identified corresponding research directions to become outdated. Most importantly, advances in LLM development may uncover novel challenges or make some of the existing challenges much more critical to address.

## 5.2 Prior Work

This work comes on the back of several works focused on highlighting harms, risks, and various other societal challenges posed by LLMs, and other advanced AI systems they may give way to (Bengio et al., 2023). These risks have been recognized by various governing bodies around the world, e.g. United States government (hou, 2023), United Kingdom government (Office, 2023), and the United Nations

(Nichols, 2023). Among scholarly work, Weidinger et al. (2022) and Shelby et al. (2022) review and taxonomize various harms and risks posed by LLMs and other AI systems. Shevlane et al. (2023) propose evaluating LLMs for 'dangerous' capabilities that may pose extreme risks. To improve risk assessment, Weidinger et al. (2023a) propose a framework for sociotechnical evaluation of LLMs and other generative systems. In a similar vein, Solaiman et al. (2023) call for evaluating AI systems, including LLMs, for social impact. Other works examine the possible societal impacts of LLMs — Eloundou et al. (2023) review possible disruptions to job market that might be caused by LLMs and Brundage et al. (2018) highlight ways in which AI systems may be misused by malicious actors. There additionally exist works that focus on discussing societal-scale harms that may occur if a misaligned competent AI system is allowed to act in an unsafe way (Critch and Krueger, 2020; Hendrycks et al., 2023; Critch and Russell, 2023). Our work is complementary to all the aforementioned work as we focus on listing technical and sociotechnical challenges that need to be addressed to overcome these challenges.

Several prior works have attempted to identify critical open problems and outline research directions, for the development of safe and aligned AI systems. Kenton et al. (2021) is perhaps the closest work to ours in scope but contains a much narrower discussion regarding the safety of LLM-based systems. Similarly, there exist public agendas by leading LLM companies that outline their approach to safety and alignment of LLM-based systems (Leike et al., 2022; Anthropic, 2023a). However, these agendas lack diversity and are primarily focused on the research directions being championed by the corresponding company. In contrast, our work boasts a diverse academic authors lineup and platforms diverse research directions. Amodei et al. (2016) and Hendrycks et al. (2021a) share a similar goal to ours of highlighting important challenges that require to be addressed for safe AI; however, they lack explicit focus on LLMs. Other similar efforts include Dafoe et al. (2020) and Ecoffet et al. (2020), which respectively consider alignment and safety of multi-agent and open-ended systems. Other works have argued for specific approaches for the development of safe AI systems; Leike et al. (2018) argue for scalable reward modeling to align advanced AI systems, Tegmark and Omohundro (2023) argue for distilling learned logic of AI systems into code which can be formally verified, and Brundage et al. (2020) call for designing institutional, software and hardware infrastructure to support verifiability of the claims made about AI systems. Dafoe (2018) and Shavit et al. (2023) are agendas focused on governance aspects of AI systems. There also exist several other agendas focused on the safety of AI-based systems with varying levels of relevance to the safety and alignment of LLM-based systems (Russell et al., 2015; Soares and Fallenstein, 2014; Henderson et al., 2018; Dinan et al., 2021; Gruetzemacher et al., 2021). Also related to our work are studies such as Gabriel (2020) and Prabhakaran et al. (2022a), which consider the question of what the alignment target ought to be for general-purpose AI systems like LLMs.

Due to the focus on LLMs, our work is also related to other agendas and surveys on LLMs. The notable agendas include Kaddour et al. (2023) and Huyen (2023), which review challenges in LLM research and applications of LLMs in general, without any explicit focus on safety or alignment. Among surveys, Bowman (2023) is a short survey that provides an opinionated review of key facts about LLMs' development. Zhao et al. (2023) comprehensively review the various facets of LLM development and their utilization, including techniques used to promote safety and alignment. Ji et al. (2023b) provide a review of alignment techniques in the context of foundation models. There also exist several surveys on specific aspects of LLMs that are covered in this work; Dong et al. (2022) survey the literature on in-context learning, Huang and Chang (2022) provide extensive discussion on reasoning capabilities of LLMs, Mozes et al. (2023) review security related issues of LLMs, and Casper et al. (2023a) survey limitations of reinforcement learning from human feedback for safety finetuning of LLMs and the associated open problems.

In the aftermath of the unexpected success of LLMs, there has been a growing sense that 'impactful' machine learning and natural language processing research requires tremendous resources and thus is no

longer viable for academic researchers. This work is partially inspired as a rebuttal to that perspective and posits that alignment and safety are ripe fields for contributions by academic researchers. Other related efforts include Saphra et al. (2023) and Ignat et al. (2023). Saphra et al. use historical analogies to argue that current disparities between academic and industrial labs regarding the scale of resources are temporary and argue that evaluations and data are still the primary bottlenecks. Ignat et al. similarly rebut the perspective that NLP research is no longer amenable to academic research by highlighting various under-researched research areas within NLP and other related fields.

## Acknowledgements

We, in particular UA, would like to thank Robert Kirk, Lorenz Kuhn, Nicholas Carlini, Katherine Lee, Alexander Rush, Geoffery Irving, Greg Yang, Sam Bowman, Mikita Balesni and Tim Rudner for discussions and feedback on the idea and initial outline(s) of this work. We would additionally like to thank Will Merrill, Spencer Frei, Kawin Ethayarajh, Kayo Yin, Roger Grosse, Victor Veitch, Sylvia Lu, Bilal Chughtai, Bruno Kacper Mlodozeniec, Dmitrii Krasheninnikov, Matthew-Farrugia Roberts, Ben Bucknall, Dan Hendrycks, Neel Nanda, Vinodkumar Prabhakaran, Seth Lazar, Percy Liang, Justin Bullock, Sara Hooker and many others for providing helpful feedback on the draft. We also thank Elio Arturo Farina for his in-kind support with Latex formatting and typesetting. All errors and omissions are our own.

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
