# OpenReview forum: "Foundational Challenges in Assuring Alignment and Safety of Large Language Models"
_TMLR — Accepted by TMLR_

### Review · Reviewer_Zqu4 · 2024-07-15

**Summary Of Contributions:**

This is a survey + position paper on foundational challenges in safety and alignment of LLMs. It organizes the challenges along the dimensions of understanding the science, practical development and deployment methods and societal impact. It defines 18 top-level challenges and derives 215 specific problem statements under those. The coverage of the field, and clarity and depth of the presentation is noteworthy.

**Audience:**

Yes

**Broader Impact Concerns:**

The paper does not need a broader impact statement IMO.

**Claims And Evidence:**

Yes

**Requested Changes:**

* The paper surprisingly doesn't include any illustrations or graphics. A paper of this density of concepts and breadth of coverage should include good illustrations, e.g., a mindmap of how different concepts are related, timeline and examples of different LLM capabilities and alignment/safety approaches, and so on. I would highly recommend that the authors do this.
* I am aware that the paper is already long. It's strength is that it is easy to read and perhaps accessible to larger community. However, it would be nice to include central mathematical formulations to bring some rigor without getting overly technical. For example, I would consider putting some key problem statements within separate boxes that can be referenced or skipped based on the reader's preference. There are also some concepts that the authors do not introduce (e.g., out-of-context learning, which is referenced in multiple places but not described the first time it is introduced).
* Some other topics that the authors may consider including are constrained decoding and verified training.
* In my view, the paper misses links to the literature on safety related techniques developed in the pre-LLM world. I am not suggesting exhaustive treatment but the issue of trustworthy-ness of systems has been around since the beginning of computing. So some references to topics like safety of control systems and logic-based approaches to software verification would be useful. In fact, I would consider "taking learnings from pre-LLM approaches to safe and trustworthy system design and applying them holistically to LLM-based systems" to be also a challenge worth including.

Of the above suggestions, I consider the first to be essential (it doesn't change anything, only makes the work more accessible). The others are up to the authors to decide, but if they decide not to act on those, I would recommend that these points be included in the limitations section for the benefit of the readers.

**Strengths And Weaknesses:**

Strengths
+ LLMs have made rapid progress in their capabilities and practical use. This paper does a good job of analyzing safety and alignment challenges posed by this progress and weaving disparate aspects into a common theme.
+ The paper is well-organized into three distinct aspects which together provide a broad view of the area. Though a long paper, the quality of exposition is consistently good and the summary tables are helpful.

Weaknesses
- The paper has a good opportunity to make the work even more informative and accessible by designing informative and appealing illustrations. Please see the requested changes for some suggestions.
- The paper does not include mathematic formulations and discussion of traditional safety approaches in system design. Please see the requested changes.

---

> ### Author Response · Authors · 2024-07-31
>
> We thank the reviewer for their comments and appreciation. Please see our response to the requested changes below:
>
> 1. *Lack of illustrations/graphics*: Thank you for the suggestion. We are coordinating an effort to include graphics similar to those in https://arxiv.org/abs/2303.16200 or https://arxiv.org/abs/2109.13916. We might also include mindmaps to help the reader navigate the content. This may take us some time, but we are working on it! Please, let us know if you have any other graphics that could serve as a reference.
>
> 2. *Include mathematical formulations*: We decided to rely on references instead of formulating problems ourselves. We always extensively cite the most relevant papers for specific problems where the reader may find more details if they are interested. The main reason was to limit the length of the paper since formulating all problems would take a lot of space. Also, if we decided to only formulate some of them, this may give the reader a misleading impression of what problems are more important or better defined.
>
> 3. *Definition of some concepts are missing*: Thank you for bringing this up. We include relevant references for readers interested in specific topics. Including definitions for all specific terms used would substantially increase the length of the paper.
>
> 4. *Additional topics*: Thank you for suggesting these topics. We believe that constrained decoding is an interesting application of LLMs at the moment, but is tangential to ensuring alignment and safety. Especially since we expect future systems to require open-ended generations to fulfill complex tasks (e.g. LLM agents). We have discussed verified training in line in the section 4.5.10:
>
> > verifying that a particular model was trained on the dataset it is claimed to have been trained on (Choi et al., 2023b; Garg et al., 2023).
> We also note that we have made a general admission in our limitations section that our coverage is non-exhaustive.
>
> > Relatedly, due to the limited scope of our work, we have intentionally omitted many important research directions pertaining to the safe development and deployment of aligned AI-based systems, such as improving the general understanding of deep learning (Arora et al., 2020), understanding critical aspects of agency (‘agent foundations’ Soares and Fallenstein, 2014), and developing AI systems whose safety can be proved or verified (Brundage et al., 2020; Tegmark and Omohundro, 2023), cooperative AI (Dafoe et al., 2020) etc.
>
> 5. *Pre-LLM safety techniques*: Do you have any specific pointers that would be relevant to some of the discussed topics? We would be happy to include those. We currently include cites and references to pre-LLM safety in sections like robustness (Section 3.5) and poisoning (Section 3.6), where we believe there is a lot of previous work that could inform future LLM safeguards. We also have a discussion of program-synthesis-based approaches to interpretability (which are naturally amenable to logic-based software verification) in section 3.4.11.

---

> > ### Author Response · Authors · 2024-08-07
> > **Figures added for sections 2.1, 2.6, 3.5 and 3.6**
> >
> > Regarding your suggestion to add illustrations/graphics. We have added following graphics.
> > - A graph showing increase in context length of LLMs over time (figure 1 in section 2.1)
> > - Illustrative figures for section 2.6 (multi-agent safety), section 3.5 (jailbreaking) and section 3.6 (poisoning)
> >
> > Additionally we have added a table for section 2.6.5 that summarizes differences in the ‘nature’ of LLM-agents and traditional multi-agent RL-based agents.
> >
> > We are working on adding more graphics as well.

---

> > > ### Comment · Reviewer_Zqu4 · 2024-08-12
> > > **Changes and response**
> > >
> > > Thank you for your response and adding new figures. The figures are a useful addition.
> > >
> > > Nit: type in caption of Fig 2(a): mutli-agent
> > >
> > > Here are some suggestions for pre-LLM safety techniques (software and hybrid systems verification):
> > > https://dl.acm.org/doi/10.1145/1592434.1592439
> > > https://www.cambridge.org/highereducation/books/logic-in-computer-science/9022E2BE5E7C9F20D259F4A83986236C#overview
> > > https://link.springer.com/book/10.1007/978-1-4419-0224-5

---

> ### Author Response · Authors · 2024-08-20
>
> Thank you for providing these references. These have helped us better understand your perspective. However, I would like to note that all three of the references you have provided are quite old (pre 2010) and unfortunately very little progress has been made in terms of extending and scaling these logic-based approaches to help with the verification of large neural networks (LLMs are at the end HUGE NNs). This consistent lack of (notable) progress and theoretical results like https://link.springer.com/chapter/10.1007/978-3-031-45286-4_2?fromPaywallRec=false indicate that direct verficiation of NNs might be intractable.
>
> A proposal that has gained some traction recently (https://arxiv.org/pdf/2405.06624) for the development of 'verified' AI systems is one by Tegmark and Omohundro (https://arxiv.org/pdf/2309.01933) which introduces the idea of 'distilling' NNs into some other form of software that might be amenable to formal verification tools. This direction has already been discussed in the paper in section 3.11. **As we did not have any citations in our existing work on 'formal verification', we have added the citations provided by you in the section 3.11 where we mention formal verification**.
>
> If you are still dissatisfied with our response, please let us know. We are happy to discuss this further. We are also open to listing this in our limitations section (though we do not ourselves see this as a limitation).
>
> Minor Changes:
> - Two new figures added in section 2.4
> - Fixed the typo in caption of figure 2a

---

### Review · Reviewer_1KZj · 2024-07-18

**Summary Of Contributions:**

This paper presents 18 foundational challenges in assuring the alignment and safety of large language models (LLMs). These challenges are described in three different categories: 7 in scientific understanding of LLMs, 6 in development and deployment methods, and 5 sociotechnical challenges. Based on these 18 identified challenges, 200+ concrete research questions are proposed.

**Audience:**

Yes

**Claims And Evidence:**

Yes

**Requested Changes:**

It will be great if the authors could address the weaknesses and expand the paper along this directions.

**Strengths And Weaknesses:**

Strengths:
The information presented in this paper is huge, and the presentation is well organized. The readers would obtain and learn a lot of knowledge in the field of LLMs and touch the research frontier of LLMs.

Weaknesses:
1. Finetuning methods are discussed in section 3.2 and don't distinguish Supervised fine-tuning (SFT) or Reinforcement learning from human feedback (RLHF). SFT and RLHF are the major techniques for training AI systems to align with human goals and meet safety requirement, unfortunately the limitations  and challenges of either SFT of RLHF are briefly mentioned and not covered in detail.

One important paper is missing: Stephen Casper et al., Open Problems and Fundamental Limitations of Reinforcement Learning from Human Feedback, Transactions on Machine Learning Research, December, 2023.

2. In Sections 2.2-2.3, the authors discuss the Limitations of Benchmarking for Measuring Capabilities and Assuring Safety A best paper in NeurRIPS 2023 and Formalizing, Forecasting, and Explaining Emergence. But an paper is missing:

Rylan Schaeffer, Brando Miranda, Sanmi Koyejo, Are Emergent Abilities of Large Language Models a Mirage?, NeurRIPS,

The views and opinions should be discussed in this paper.

---

> ### Author Response · Authors · 2024-07-31
>
> We thank the reviewer for their comments. The author has mentioned two weaknesses of the paper.
>
> * *The section on finetuning does not distinguish between SFT and RLHF*: We agree with this observation, however, we would like to note that this is an intentional choice on our part. In addition to SFT and RLHF, several other objectives are also used in finetuning stage (e.g., DPO, KTO, RLAIF etc.). Distinguishing between all these objectives would not have been possible, hence, to keep our discussion tractable, we made an intentional choice to group all the different finetuning procedures under the umbrella term of ‘safety finetuning methods’. We make an explicit note of this in 2nd paragraph of the section 3 intro:
>
> > We collectively refer to all the techniques (supervised finetuning, reinforcement learning-based training from human or AI feedback, unlearning methods) used in the finetuning stage as 'safety finetuning' methods.
> We would also like to note that all the critiques that we make in section 3.2 apply equally to both SFT and RLHF.
>
> The reviewer has additionally mentioned that mention of the paper “Open Problems and Fundamental Limitations of Reinforcement Learning from Human Feedback” is missing. We have rectified this by adding the following line in the introduction of the section 3.2:
>
> > See Casper et al. for a complementary discussion on the limitations of reinforcement learning from human feedback.
>
> * *Are emergent abilities of LLM a mirage? paper has not been discussed*: We have provided an extensive critique of the aforementioned work in the third paragraph of section 2.3.4 (Formalizing, Forecasting and Explaining Emergence).

---

> > ### Comment · Reviewer_1KZj · 2024-08-19
> > **Distinguish between SFT and RLHF**
> >
> > Thanks for the response.
> >
> > I still think the paper should distinguish between SFT and RLHF, where SFT is a technique by demonstration, and RLHF is a technique by comparison, and there is no way to surpass the human labeler in SFT, and it's possible surpass human labeler in RLHF.
> >
> > The methods like DPO, KTO, RLAIF etc. all belong to RLHF.

---

> ### Author Response · Authors · 2024-08-20
>
> We respectfully disagree with the reviewer's perspective on this. By treating all finetuning methods in an abstract way; we are able to provide an overview of open questions in a concise yet comprehensive way. Note that the challenges we point out, e.g., "How Does Finetuning Change a Pretrained Model?" (section 3.2.1), "Finetuning Misgeneralizes in Unpredictable Ways" (section 3.2.2), "Output-Based Adversarial Training May Incentivize Superficial Alignment" (section 3.3.3) all apply *equally* to both SFT and RLHF.
>
> If the distinction between different types of finetuning methods has to be made; it is not clear to us why it should be SFT vs RLHF, and not, for example, whether the feedback signal is coming from a human (SFT/RLHF) or coming from the model (RLAIF or SFT on synthetic data)? From reviewer's comment, it seems that the rationale for this distinction is that SFT can not surpass human labeler's performance, but RLHF can. We are not sure that this is factually correct, and would respectfully like to point out this recent paper that analyses autoregressive training of transformer and concludes that transcendence is indeed possible even when the model is only trained via demonstrations: https://arxiv.org/abs/2406.11741
>
> We hope we have provided a satisfactory response to the reviewer. Please do let us know if you would like to discuss this further.

---

> > ### Comment · Reviewer_1KZj · 2024-09-01
> >
> > Llama 2 technical report, Llama 2: Open Foundation and Fine-Tuned Chat Models, illustrates the difference bwtween SFT and RLHF.

---

> > > ### Author Response · Authors · 2024-09-02
> > >
> > > We agree and know there are many differences between SFT and RLHF. However, we believe these differences are not relevant to the scope of the paper since nowadays both SFT and RLHF are applied together for every state-of-the-art model, and it is often not possible to disentangle these two when it comes to understanding the overall system limitations.
> > >
> > > When we point to a vulnerability that is dependent on the post-training algorithm, we have made explicit mentions. See for example Section 3.6.2, where we highlight that understanding vulnerabilities of different stages is an important challenge.

---

### Review · Reviewer_eZCZ · 2024-07-24

**Summary Of Contributions:**

This paper is review and positional paper. It proposes 18 foundational challenges in assuring the alignment and safety of LLMs, which can be categorized into three: (1) scientific understanding of LLMs, (2) development and deployment methods, and (3) sociotechnical challenges. This paper reviews state-of-the-art techniques of LLMs, and then highlights current challenges and research questions. Especially, in each challenge and research question, it provides motivation, background, related work, and future research directions. Moreover, the paper presents a well-organized structure. I believe the paper would be beneficial for broad community from Machine Learning and Natural Languae Processing to Social Science and Policy.

**Audience:**

Yes

**Broader Impact Concerns:**

This paper does not include a “Broader Impact Concerns” section. However, it adequately addresses its limitations in section 5.1. One additional concern is that readers might become overly focused on the topics and research questions presented in this paper. It is important to remind them not to limit themselves to these agendas and to emphasize that there are still uncovered topics that exist.

**Claims And Evidence:**

Yes

**Requested Changes:**

Please refer to the weakness in the previous section.

**Strengths And Weaknesses:**

### Strengths

- The paper is well-written, structured, and organized. This is a huge volume, and, like a textbook, the reader guide is beneficial for them to follow. Also, research questions and their corresponding sub-sections are well-linked.
- This paper summarizes current challenges and provides valuable 200+ research questions with background and references. Therefore it will contribute to the community as well as non-domain expertise, by encouraging them to figure out the status quo challenges and to push the boundary.
- The paper properly includes its limitations.

### Weakness

- This paper covers broad technical topics from fundamentals to safety training and attack, but several topics are absent such as
    - Hallucinations, Uncertainty, and Calibration
    - Sovereignty of Data and Models
    - Multi-linguality, Culture, and Social Bias.
- In Chapter 2, fundamental scientific understanding is introduced, which is helpful and well-organized. However, some sub-chapters — e.g., scaling law, emergent ability, and reasoning — are hard to connect their relation to the safety of LLMs, as far as I understand. Wrapping up paragraphs would emphasize their importance to readers.
- In overall and especially in chapter 4.5, the paper is US, UK, and EU-centric. Surveying and stating other nations and companies would broaden the sights of readers and improve the minority’s visibilities.
- When it comes to the definition of “Safe” and “Harms”, the authors state them as follows in the section 1.2:
    > We consider a system safe to the extent ***it is unlikely to contribute to unplanned, undesirable harms***(Leveson, 2016). This is a somewhat expansive definition, accounting not only for the technical properties of the system, but also the way in which it is (or is likely to be) deployed and used (Weidinger et al., 2023b), though it is narrow in the sense that it does not consider intentional harm, and it does not set out any criteria for what constitutes harm.
    - As the authors mention, the terms are somewhat expansive and abstract, but stating specific definitions of “safe and harm” which are currently used for deployments, would make readers easy to make the connections.
- These are mild suggestions.
    - Although the paper assumes that its readers have basic knowledge, basic equations are not presented.
    - Since this paper covers huge topics, refer other related review papers that cover each sub-topic would help its reader to further explore the domain.
    - These days, "post-training" is wildly used in the way to call collectively "supervised fine-tuning" and "reinforcement learning by human feedbacks", or whatever training process after the pre-training. What about using this term or just mentioning it?

---

> ### Author Response · Authors · 2024-07-31
>
> We thank the reviewer for their comments and appreciation. Please see our response to the requested changes below:
>
> 1. *Several topics are absent*: Having a section for all existing problems was not always possible and they are included in broader topics:
>    * There are certain problems like hallucination and uncertainty that have implications for many of the challenges we introduce and are mentioned in context when relevant. For example, Section 4.3.2 mentions the importance of reducing hallucinations and providing uncertainty estimations. Section 2.5.2 also discusses need for better uncertainty estimates. Searching for the keywords also results in other sections where these challenges play an important role.
>    * We have covered some topics around sovereignty in section 4.5, e.g., section 4.5.10 on data governance discusses data sovereignty. We also note that in the limitations section, we have noted that “our major focus is on technical challenges” meaning that there are many important sociotechnical challenges that have not been discussed in ths work.
>    * Social bias and multi-linguality is covered in section 4.3.1.
>
> 2. Relation between terms in Chapter 2 and Safety: reasoning and emergent abilities will be crucial to enable capabilities that may result in unsafe LLMs such as planning or deception. Understanding and prediction these capabilities will be very important to anticipate failure modes and inform safety critical decisions. If you think this connection is not clear, we could try to clarify further. Most of the relevant content is subsection introductions (e.g. 2.2 before the subsubsections).
>
> 3. *The paper is US, UK, and EU-centric*: Thanks for bringing this up. In Section 4.5, we also have references to China’s AI regulation and the need for international cooperation. But, on the whole, the discussion in section 4.5 is indeed highly focused on few geographic regions. This limitiation is difficult for us to rectify, so, we have added the following sentence in the limitations section to make the reader aware of this:
>
>      > “Additionally, while we have made our best effort to avoid any geographic bias --- in some sections, particularly Section 4.5, the discussion is biased towards few geographies, specifically, US, Europe and China.”.
>
> 4. *Stating specific definitions of “safe and harm”*:  We have intentionally used an expansive definition of safety in our work as we want our work to have broad appeal among the machine learning and other related communities. Regarding linking our definition to the definitions currently used for deployment, we have several concerns. We note that there is considerable disagreement within the community currently about what ‘safety’ means within a deployment context. Indeed, different deployers and model developers (e.g., Meta vs Anthropic) have different definitions of safety. Hence, favoring one deployer/model-devleoper’s definition over another might not be the best idea. In any case, we expect the deployment-specific definition to be ephemeral and evolve over time. Thus, linking our work with any current definition being used in the deployment might reduce the relevance of our work if/when that definition were to become outdated.
>
> **Mild suggestions**
> * *Other review papers should be cited*. Our prior work section cites several review papers that talk about AI systems safety in general. Then, in each specific subsection, review papers are cited when appropriate. Did you find any specific paper missing?
>
> **Broader impact**
>
> Our work is indeed not exhaustive and we believe our current limitations sections is explicit about this, and we make this admission at multiple places within that section.
>
> > due to the uncertain nature of the LLM development landscape, it is possible that important challenges may have been omitted.
>
> > due to the limited scope of our work, we have intentionally omitted many important research directions pertaining to the safe development and deployment of aligned AI-based systems [...]
>
> > The nature of the challenges posed in this work may change over time. Some challenges may get sidestepped [...]
>
> > Also, the future of LLMs is very uncertain and we acknowledge that the landscape of relevant research problems may change in the future. This is again noted explicitly as a limitation.

---

> > ### Comment · Reviewer_eZCZ · 2024-08-09
> > **Thanks for your response.**
> >
> > Thank you all for your response to my comment. After carefully looking into, all my concerns have been addressed.

---

### Decision · Action_Editor_k7ya · 2024-09-01

**Recommendation:** Accept as is

**Comment:**

The authors have produced a survey here that is very impressive in its scope, and the reviewers and I both agree that it substantiates its claims and is above bar for TMLR. As reviewer eZCZ says, this is almost like a textbook in scope. The citations that the manuscript on arXiv is already accruing attests to the fact that the community at large views this as a useful resource.

The comments are all quite minor and largely presentational. As a result, I believe this version can be accepted to TMLR without further revision.

**Audience:**

Yes, this is an extremely timely topic for a large-scale survey

**Claims And Evidence:**

The work claims to provide a set of foundational challenges in LLM alignment, an organization of these, and a set of open research questions. The reviewers and I agree that the paper succeeds at doing this in a very thorough and well-structured way, improved by the inclusion of figures and a few remaining citations.